# TFvelo: gene regulation inspired RNA velocity estimation

Jiachen Li [1], Xiaoyong Pan [1], Ye Yuan [1]✉ & Hong-Bin Shen [1]✉

RNA velocity is closely related with cell fate and is an important indicator for the prediction of cell states with elegant physical explanation derived from single-cell RNA-seq data. Most existing RNA velocity models aim to extract dynamics from the phase delay between unspliced and spliced mRNA for each individual gene. However, unspliced/spliced mRNA abundance may not provide sufficient signal for dynamic modeling, leading to poor fit in phase portraits. Motivated by the idea that RNA velocity could be driven by the transcriptional regulation, we propose TFvelo, which expands RNA velocity concept to various single-cell datasets without relying on splicing information, by introducing gene regulatory information. Our experiments on synthetic data and multiple scRNA-Seq datasets show that TFvelo can accurately fit genes dynamics on phase portraits, and effectively infer cell pseudo-time and trajectory from RNA abundance data. TFvelo opens a robust and accurate avenue for modeling RNA velocity for single cell data.

Single-cell RNA sequencing (scRNA-seq)[1–3] provides a wealth of information about the gene expression profile of individual cells. To expand the scope of scRNA-seq analysis beyond a static snapshot and capture cellular dynamics without tracking alive individual cells over time, many trajectory inference (TI) approaches, like PAGA[4], Monocle[5], Slingshot[6] and Palantir[7] have been developed. While these TI methods enable pseudotime analysis at both the cell and gene levels, they typically require the annotation of initial cells[8]. In recent years, RNA velocity[9] theory was proposed, which describes the time derivative of gene abundance in a physical-informed approach, by modeling the relationship between the unspliced (immature) and spliced (mature) mRNAs. Combining velocities across multiple genes, velocity-based pseudotime and cell fate can be inferred from the transition probability between cells. To estimate the RNA velocity, velocyto[9] is introduced as the vanilla approach with a steady state assumption. ScVelo[10] models the dynamics without the steady state assumption and employs an Expectation-Maximization (EM) approach for better reconstructing the underlying kinetics.

More recently, several approaches have been developed to improve the estimation of RNA velocity. For instance, VeloAE[11] utilized an auto-encoder and low-dimensional space to smooth RNA velocity. Still in the unspliced/spliced space, DeepCycle[12] used an autoencoder

to map cell cycle-related genes. UniTVelo[13] directly designed a function of time to model the spliced RNA level. In contrast to deterministic models, VeloVI[8], Pyrovelocity[14] and veloVAE[15] estimated RNA velocity using Bayesian inference frameworks. LatentVelo[16] embeds the unspliced and spliced expression into the latent space with a variational auto-encoder, so that a low-dimensional representation of RNA velocity could be obtained. CellDancer[17] models the transcription, splicing and degradation rate of a gene as a function of its unspliced and spliced counts with a neural network. Furthermore, apart from the unspliced/spliced data, dynamics can also be modeled with additional information. For instance, Dynamo[18] improve the RNA velocity model by using the new/total labeled RNA-seq[19]. Taking the advantage of single-cell multi-omics technology[20,21], RNA velocity analysis can be expanded to protein abundances (protaccel[22]) and single-cell ATAC-seq (MultiVelo[23]) datasets. However, these additional omics data or labeling may be cost or even not available in most single cell datasets.

While RNA velocity theory has improved the inference of single-cell trajectories, pseudo-time, and gene regulation inference in numerous studies[24], current RNA velocity models still face severe challenges. Firstly, RNA velocity models always treat each gene independently and do not take the underlying regulation into consideration[25], although it is expected to be helpful for cell fate

[1]Institute of Image Processing and Pattern Recognition, Shanghai Jiao Tong University, and Key Laboratory of System Control and Information Processing, Ministry of Education of China, Shanghai 200240, China. ✉e-mail: yuanye_auto@sjtu.edu.cn; hbshen@sjtu.edu.cn

inference by integrating gene regulatory mechanism analysis. Secondly, these models can not fit the gene dynamics well on most genes, which might be because that the transcriptional dynamics of mRNA splicing may not provide sufficient signal in single cell resolution[25]. Most conventional approaches assume that the joint plot between unspliced and spliced expression should form a clockwise curve on the phase portrait because of the phase delay within them[9,10,13,26], which is, nevertheless, rarely observed from the data. The high sparsity and noise nature of unspliced mRNA counts[27,28] could be one significant reason. The short time scale of splicing process could also make it hard to extract dynamics from the delay between unspliced and spliced RNA. Last but not least, the RNA velocity models can only be applied to scRNA-seq datasets with unspliced/spliced or new/total labels. However, the abundance of unspliced/spliced data may not be available in certain datasets, such as those obtained through Fluorescence In Situ Hybridization (FISH) technologies[29], and some human sequencing data due to privacy restrictions. These challenges motivate us to construct the gene dynamics based on regulation, instead of only relying on the unspliced/spliced counts.

The gene regulation has been explored a lot in the field of single cell research[30–34]. In this study, we aim at inferring the gene dynamics and cell fate based on the underlying regulatory among genes. To this end, we investigate the relationship between gene regulatory patterns and RNA velocity, which reveals that the RNA velocity can be approximately estimated as a linear combination of the expression levels of transcription factors (TFs). Notably, the TFs with the non-zero weights are significantly enriched with TF set functionally linked to the target gene. In addition, we find that the joint distribution of expressions between a TF and its target could form a clockwise curve on the co-expression plot[35], indicating the phase delay between them. Those findings imply that the abundance of TFs could also be used to construct dynamics model of RNA velocity.

In this study, we propose TFvelo to model the RNA velocity based on the expression levels of TFs, where the velocity refers to the changing rate of RNA abundance. The computational framework of TFvelo relies on a generalized EM algorithm, which iteratively updates the learned representation of TFs, the latent time of cells, and the parameters in the dynamic equation. TFvelo can accomplish analysis performed by previous RNA velocity studies, such as gene-specific phase portrait fitting, velocity modeling, inferring pseudo-time without annotation of initial states, and visualizing cell fate with the velocity-based transition stream plot. Unlike methods that depend on splicing information, TFvelo can robustly work on genes with sparse unspliced counts and even datasets without splicing information. Using both synthetic data and multiple real scRNA-seq datasets, we show that the TFvelo model can capture the dynamics on phase portraits of TFs-target, accurately infer the pseudo-time and the developmental trajectories of cells and help explore biological findings.

## Results

### Findings: RNA velocity can be approximately estimated with TFs' abundance

The fact that RNA velocity can be modelled by the transcriptional regulation is supported by the following two findings. We find that the RNA velocity model by previous approach can be approximately estimated as a linear combination of TFs' abundance. On the scRNA-Seq pancreas dataset[9,36], given a target gene, Least Absolute Shrinkage and Selection Operator (LASSO) regression is applied to predict RNA velocity modelled by scVelo, based on the expression level of TFs and the target gene itself (Fig. S1a). Here we run scVelo pipeline on pancreas dataset with the default parameters to get the velocities, and only apply LASSO regression to those genes which are modelled by scVelo. We feed all TFs[37] into LASSO, and get a sparse weights-vector where only some TFs have non-zero weights. As a result, the RNA velocity of

gene $g$ can be estimated as:

$$v_g = \sum_{TF_i \in S_g} w_{g,TF_i} e_{TF_i} + w_{g,g} e_g \qquad (1)$$

where $S_g$ represents the set of TFs, $e_{TF_i}$ and $e_g$ are the expression level of $TF_i$ and gene $g$ respectively, $w_{g,TF_i}$ and $w_{g,g}$ are the weights of $TF_i$ and the target gene itself.

Cells are separated into training and test sets to evaluate the LASSO model. The regression results on the test set shows a high correlation between the predicted and labeled velocity, as demonstrated in Fig. S1b,c, which indicates that velocity of the target gene can be predicted based on the expressions of its TFs. In addition, as shown in Table S1, the TFs with non-zero weights are significantly enriched in TFs set functionally linked to the target gene (according to the ENCODE TF-target database[38]) with $p$ value of 6.66e-06 (one-sided $t$-test), further suggesting that velocity could be modelled by TFs. Secondly, due to regulatory relationship between TF and target, the expression levels of them change asynchronously, which could result in a clockwise curve on the phase portraits. And as expected, we find the desired clockwise curve on the co-expression plots between some TF-target pairs[35] (Fig. S2), which is similar to the theoretical prediction of existing velocity models on unspliced-spliced space. These findings suggest that it is possible to construct a novel RNA velocity model based on the expression of TFs, instead of using unspliced/spliced RNA counts or any additional experimental omics data and labels.

### Estimate RNA velocity with TFvelo

Here we report TFvelo to estimate the RNA velocity of a target gene using the abundance of the target gene itself and its potential TFs. Figure 1a provides an example of phase portrait fitting on a gene from pancreas dataset, to explain how TFvelo outperforms splicing-based methods. Previous methods focus on modeling the dynamics between the spliced and unspliced RNAs, while transcriptional dynamics of RNA splicing may not provide sufficient signal[25], and the 2D unspliced-spliced RNA level could be too noisy or sparse to fit. By comparison, TFvelo considers the relationship between the target gene and multiple TFs, by learning a representation of regulation, allowing for more robust and accurate modeling based on gene regulatory network.

For a target gene $g$, TFvelo explores the relationships between its expression level $y_g$ and the regulation of TFs on $g$. In TFvelo, the RNA velocity $\frac{dy_g(t)}{dt}$, which is defined as the time derivative of RNA abundance, is determined by the abundance of involved TFs $\mathbf{X}_g$ and itself,

$$\frac{dy_g(t)}{dt} = h\left(\mathbf{X}_g(t); \Psi_g\right) \qquad (2)$$

Using a top-down strategy, which can relax the gene dynamics to more flexible profiles[13], TFvelo directly designs a profile function of target gene's expression level,

$$y_g(t) = f\left(t; \Phi_g\right) \qquad (3)$$

where $\Psi_g$ and $\Phi_g$ are two sets of gene-specific time-invariant parameters, which describe the influence of TFs on the target gene, and the shape of phase portraits, respectively. Considering that linear models have been employed to represent the gene regulatory in previous studies[34,39–41], $h(\mathbf{X}_g(t), y_g(t); \Psi_g)$ is implemented with a linear model $\frac{dy_g(t)}{dt} = \mathbf{W}_g \mathbf{X}_g(t) - \gamma_g y_g(t)$. The profile function of $y_g(t)$ can be chosen flexibly from a series of second-order differentiable functions[13], which is designed as a

function in implementation (Methods). This is because sine functions can model the nonmonotonic dynamics start from both up-regulation and down-regulation (e.g., gene MGST3 in Fig. S8), which is

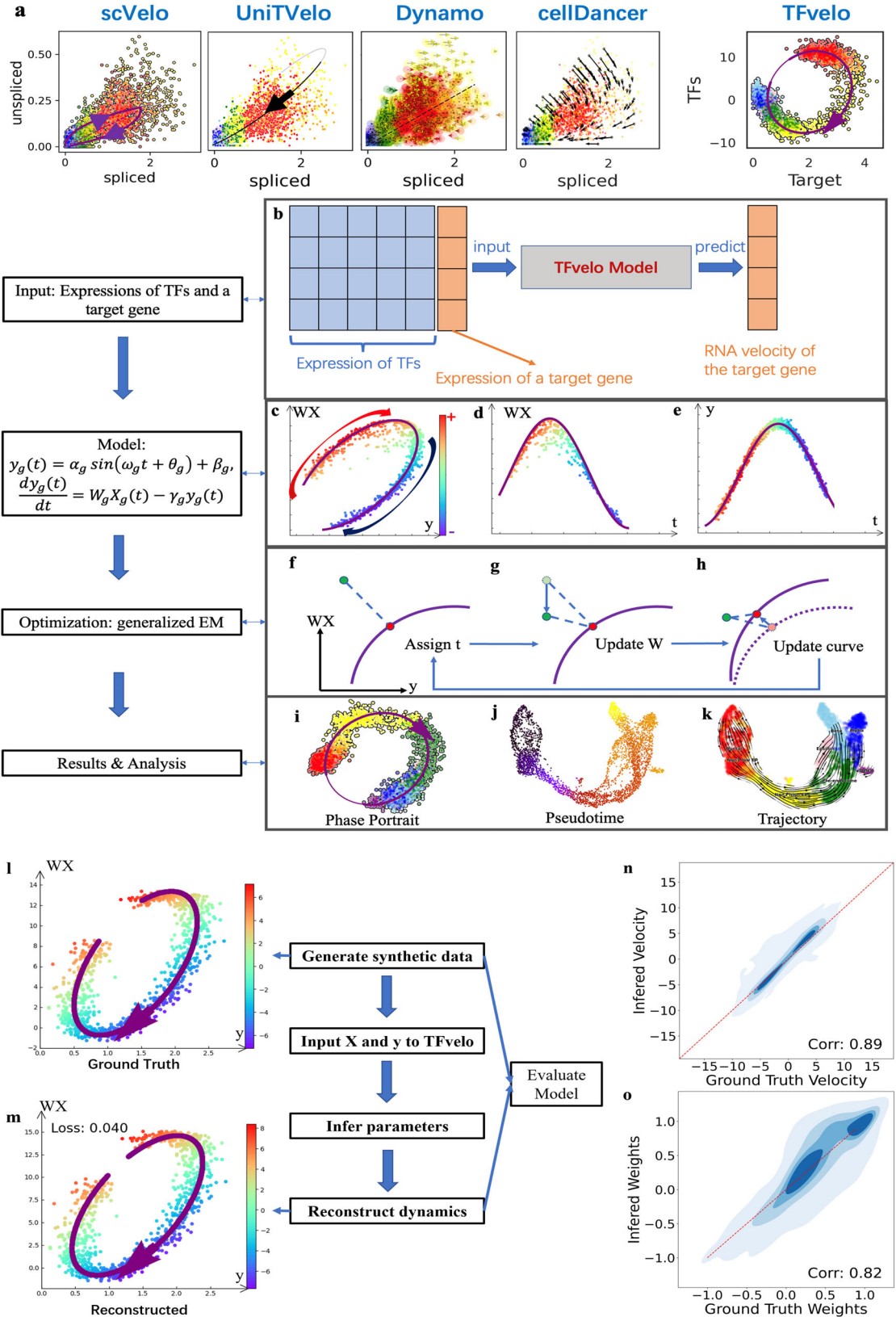

not feasible for those methods relying on one switching time point[9,10,18].

For optimization, all learnable parameters are divided into three groups, which are cell-specific latent time $t$, TFs' weights $\mathbf{W}_g$ and the shape parameters of phase portraits $\psi_g$, in which $n_{cell}$ and $n_{TF}$ represent number of cells and number of genes, respectively.

A generalized EM[42,43] algorithm is adopted to minimize the loss function in each iteration, by updating the three groups of parameters alternately (Fig. 1f–h). Based on the optimized models on all genes, we can obtain a transition matrix and calculate the pseudotime and velocity stream for cell trajectory analysis. Please find the details in Methods.

**Fig. 1 | The overview of TFvelo. a** The comparison among constructed dynamics on LITAF gene in Pancreas dataset by TFvelo and previous approaches. Cells are colored in the same way as those in Fig. 2a. **b** The workflow of TFvelo. **c–e** The phase portrait plot on WX-y space, the plot of $WX$ with respect to pseudotime, and the plot of $y$ with respect to pseudotime. The color bar indicating the RNA velocity is share by the three panels. For simplification, $W_g, X_g, y_g$ and $t_g$, are written as W,X,y and $t$, respectively. **f–h** The steps within each iteration of the generalized EM algorithm. Here the purple line represents the current prediction by the model. **f** Assignment of latent time to each cell (in green) to find the corresponding target point (shown in red), which is the nearest point to the cell on the purple line. **g** Optimization of the weights $W$ to move all cells along the $WX$ axis, minimizing the

mean loss over all cells. **h** Optimization of the parameters in the dynamical function, to move the target point on the curve closer to each cell. **i–k** Visualization and analysis of TFvelo's results, including phase portrait fitting pseudo-time inference, and cell trajectory prediction. **l–o** Simulation on synthetic dataset. **l** An example of the generated synthetic dynamics. **m** Reconstructed dynamics by TFvelo of the sample shown in (**l**) with MSE loss. **n** The joint plot of ground truth velocity and inferred velocity with their spearman correlation. The red reference line refers to that the ground truth is equal to the inferred value. **o** The joint plot of ground truth weights and inferred weights with their spearman correlation. Source data are provided as a Source Data file.

## TFvelo can reconstruct the dynamic model and detect the regulation relationship on synthetic datasets

To validate that TFvelo can detect the underlying dynamic from data, we firstly test it on a synthetic dataset (see Methods), where a target gene is regulated by 10 TFs, and the corresponding weights are either positive or negative, representing that TF may up-regulate or down-regulate the target gene, respectively. We randomly generate 200 synthetic gene dynamics on 1000 cells with different ground truth parameters. For each of them, the TFvelo reconstructs the dynamic function and estimates the weight of each TF, based on the simulated expressions of TFs and target.

The performance of TFvelo was evaluated using the spearman correlation between the ground truth and the reconstructed values. Under 20 iterations for optimization, the spearman correlation coefficient between the ground truth weights and inferred weights of TFs is 0.823, and that for velocity estimation is 0.894 (Fig. 1n, o). The high consistence shows that TFvelo can effectively reconstruct the underlying dynamics. In addition, the high F1 score (0.944) and high ratio of area under the receiver operating characteristic (ROC) curve (0.962) also demonstrate that TFvelo can correctly recognize whether the weight is positive or negative. In addition, we employ a vanilla EM approach as baseline here, by removing the optimization step on TF weight, as well as the strategy of multiple points initialization. After 20 optimization iterations, the performance of this baseline is much poorer than that of TFvelo (Fig. S4). Please see Supplementary Information Section 3 for details on the synthetic dataset.

## TFvelo can model cell differentiation process on Pancreas dataset

To evaluate TFvelo on real scRNA-seq data, we first apply it to the dataset of E15.5 mouse pancreas[9], which has been widely adopted in previous RNA velocity studies. The pseudo-time and trajectory predicted by TFvelo can identify the differentiation process (Fig. 2a, b). Compared with previous methods, TFvelo shows advantage on phase portrait fitting. Example genes are shown in Figs. 2c, S7. Figure 2c compares the phase portrait fitting of different methods, and Fig. S7 provides the gene expression dynamics resolved along gene-specific latent time obtained by scVelo and TFvelo. The expression of H19, which plays an important role in early development[44], is found to increase at the beginning and then decrease at the stage of Ngn3 high EP by TFvelo. For MAML3, only TFvelo can correctly detect the process that starts with up-regulation then turns to down-regulation. Other methods can not correctly detect the order between pre-endocrine cells (in green) and those endocrine populations, including alpha, beta, delta and epsilon cells (in blue, light blue, purple and pink).

To quantitatively evaluate the phase portrait fitting, we propose three metrics, including: (1) The intra-class distance on phase portrait, which reflects how cells within the same type gather on the phase portrait (the lower the better). (2) The inter-class distance on phase portrait, which reflects how cells from different types are separated well on the phase portrait (the higher the better). (3) The fitting error on phase portrait, which measures the normalized distance between each cell to the constructed model on the phase portrait. We calculate

these metrics based on the genes that can be commonly modeled by all methods. For the "intra-class distance" and "inter class distance", we take the Paired Samples T Test between the value obtained on the same gene by TFvelo's phase portrait and the un/spliced phase portrait, which is shared by all baseline approaches, including scVelo, Dynamo, UniTVelo and cellDancer. The TFvelo results (Fig. 2d, e) show advantage with extreme high significance ($p = 4.23\text{e-}51$ and $p = 5.65\text{e-}117$). Compared with those un/spliced based approaches, TFvelo can achieve lower intra-class distance, higher inter-class distance and lower fitting error (Fig. S5h). To draw a conclusion, TFvelo can address the current issue caused by noisy un/spliced data for RNA velocity modeling, by using the learned feature with transcription regulation.

To quantitatively evaluate the learned streams, we employ two metrics adopted in VeloAE[11] and UniTVelo[13], which are "Cross Boundary Direction Correctness (CBDir)" and "In Cluster Coherence (ICCoh)". CBDir assesses the correctness of transitions from one cell type to the next, utilizing boundary cells with ground truth annotations. ICCoh is computed using cosine similarity among cells within the same cluster, measuring the smoothness of velocity in high-dimensional space within clusters. As shown in Fig. 2f, S5i, TFvelo achieves similar median value compared to UniTVelo, and significantly outperforms the other methods. Also TFvelo can even achieve a high velocity consistency (Fig. S5j), which indicates a higher consistency of RNA velocities across neighboring cells in the UMAP space[10]. The direct reason for the higher consistency scores achieved by TFvelo and UniTVelo is that they can generate smoother velocity streams across neighboring cells (Fig. S5b, d). This might be potentially attributed to the similarity in the models of TFvelo and UniTVelo. Both methods directly model gene abundance as high-order differentiable functions with respect to time, without assuming a switching point that could disrupt the smoothness of the model. In addition, to show how the velocity inferred by TFvelo and scVelo correlated with each other, we calculate the cosine similarity of velocity vectors obtained through both methods, using the commonly modeled genes. The distribution of cosine similarity on each cell type is shown in Fig. 2g.

Then we conduct KEGG pathways enrichment analysis based on the best fitted genes, and observe a strong enrichment associated with insulin secretion and the glucagon signaling pathway (Fig. 3a). Additionally, we find that REST and HMGN3 consistently exhibit high absolute weights on modeling most target genes (Fig. 3b). When examining the UMAP distribution, it is clear that REST expression decreases during differentiation, while HMGN3 expression increases (Fig. 3c). Previous research has established REST as a key negative regulator of endocrine differentiation during pancreas organogenesis[45,46]. Earlier studies have also report HMGN3 to be a regulator for insulin secretion in pancreatic cells[46]. We further analyze the weights of these two TFs on modeling genes within the insulin secretion pathway. REST consistently has a negative weight, while HMGN3 consistently exhibits a positive weight (Fig. 3d), which is consistent with the previous findings.

The comparison of phase between TFs and target are shown in Fig. 3e, which shows clear phase delay between the learned

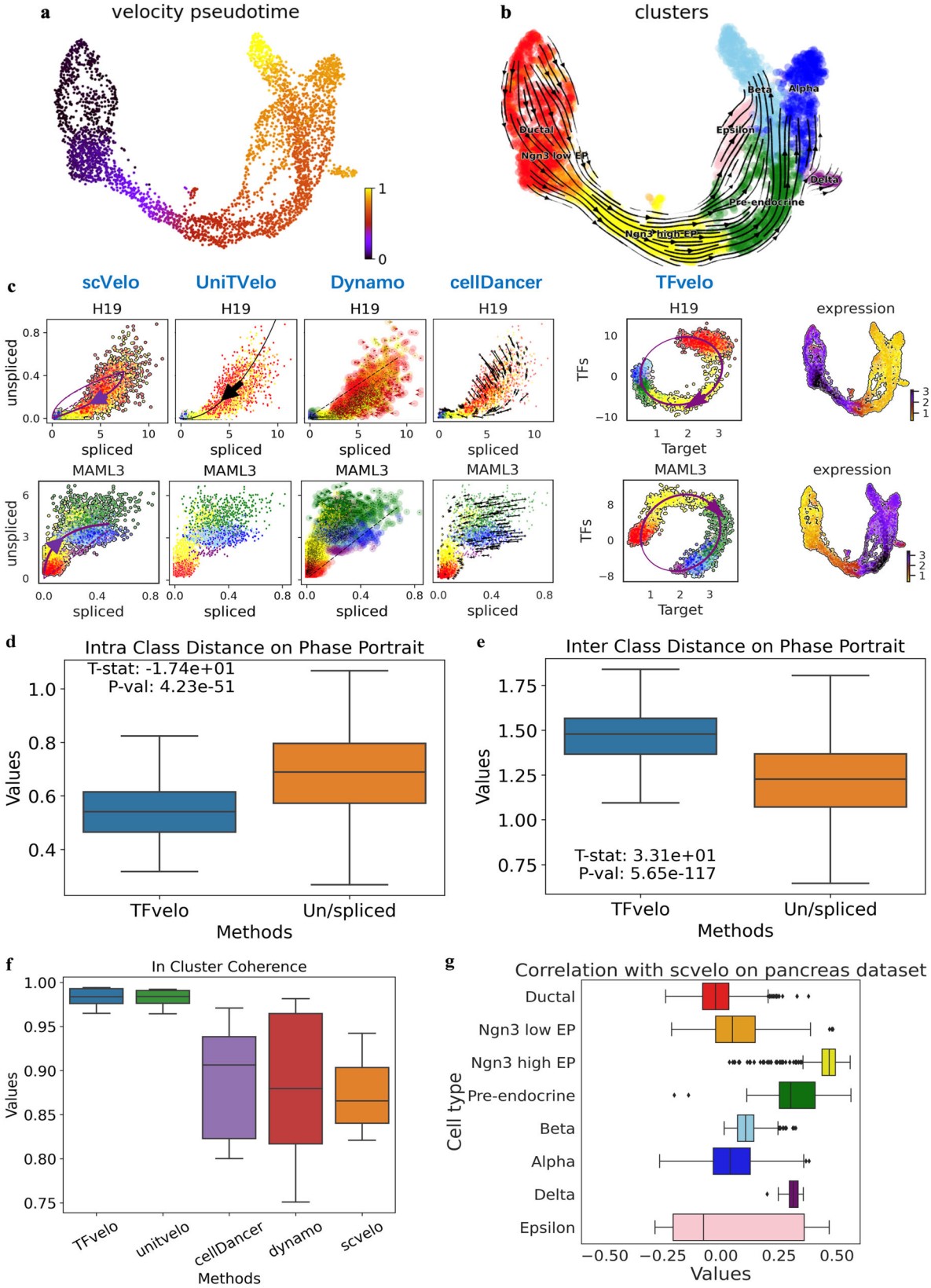

representation of TFs (**WX**) and target gene's expression. We also analyze the phase delay between a TF and the individual target. Among the two genes, HMGN3 has a positive weight on modeling SURF, where a phase delay between them can be observed. On the contrary, REST has a negative one on modeling ECE1, where their changing shows negative correlation.

In addition to the sparsity and noise in unspliced data (Fig. S6), the time scale of splicing dynamics might also make it hard to extract dynamics from the delay between unspliced and spliced RNA. It has been reported that splicing can be accomplished within 30 s[47], a duration much shorter than the timescale of the entire differentiation process described in scRNA-seq datasets. Consequently, the phase

**Fig. 2 | Results of TFvelo on pancreas dataset. a** Pseudo-time inferred by TFvelo in the UMAP-based embedding space. **b** The stream plot projected to UMAP. The Ngn3 low/high EP denotes the neurogenin3 low/high epithelial cells. **c** Two example genes for illustrating the dynamics fitting in phase portrait. The cells are colored in the same way as panel (**b**). **d** Comparison on the intra class distance in phase portrait. Two-sided *t*-test is applied without adjustment. **e** Comparison on the inter class distance in phase portrait. Two-sided *t*-test is applied without adjustment. **f** Comparison on the inter-cluster coherence. **g** The cosine similarity between the velocity vectors obtained by scVelo and TFvelo. For the boxplots in panels (**d**, **e**, **f**, **h**) the down, central and up hinges correspond to the first quartile, median value and third quartile, respectively. The whiskers extend to 1.5× the interquartile range of the distribution from the hinge. The number of samples is 405 for boxplots in panel d, e and f. The number of samples for each boxplot (from top to down) in panel g is 916, 262, 642, 592, 591, 481, 70 and 142, respectively. Source data are provided as a Source Data file.

delay between the unspliced and spliced mRNA might be too brief to be captured in the phase portrait. This might be one reason why the theoretical curve cannot be observed in the unspliced-spliced phase portrait (Fig. 3f, g). Figure 3f provide simulations to illustrate that, even with the same level of noise, a shorter phase delay between variables could make fitting the phase portrait more challenging. We also observed phenomena from the scRNA-seq dataset supporting our hypothesis. Figure 3g shows that the spliced and unspliced changes almost synchronously on some genes, while TFvelo could detect a clearer phase delay for dynamic modeling.

### TFvelo can model cell differentiation process on gastrulation erythroid dataset

The Gastrulation Erythroid dataset is selected from the transcriptional profile for mouse embryos[48], and has be adopted in RNA velocity studies[13]. Using TFvelo, the pseudo-time and cell development flow visualized by streamlines are consistent with the development in the erythroid lineage, from blood progenitors to erythroid cells (Figs. 4a, b). By comparison, several of the previous approaches cannot capture such developmental dynamics, as shown in Fig. 4d.

The sparsity in data is a common challenge for scRNA-seq studies. Considering that only about 20% of reads contained unspliced intronic sequences[9], the sparse unspliced abundance is a large obstacle for dynamics fitting in phase portrait. Our quantitative analysis of the sparsity in unspliced, spliced and total mRNA counts is shown in Fig. 4c, which verifies the high sparsity in unspliced counts. Figure 4e shows the comparison on two genes with very sparse unspliced counts to illustrate the advantage of TFvelo for addressing the issue of sparsity. Although these genes can pass the filtering and selection during preprocessing, they are still too sparse to provide sufficient information for fitting the spiced-based models well. Figure 4f provides more example genes of the comparison between TFvelo and baselines. TACC1 is annotated as a gene involved in promoting cell division prior to the formation of differentiated tissues[49]. In our experiment, the expression of TACC1 initially increases, reaching its highest value in blood progenitors 2, which is just before differentiation into erythroid cells. Subsequently, TACC1's expression decreases with pseudo-time from erythroid cells 1 to erythroid cells 3. By comparison, previous approaches fail to detect a dynamic starting from blood progenitors 1 (in red). As for HSP90AB1, only TFvelo can capture the correct dynamics from the learned phase portrait.

We next take the GO term enrichment analysis based on the best fitted genes, and find that the most significant GO terms include porphyrin-containing compound biosynthetic process (GO:0006779) and heme biosynthetic process (GO:0006783), which are directly related to erythroid development (Fig. 4g). In addition, the TFs having high weights on most target genes are GATA1, GATA2 and LMO2 (Fig. 4h), all of which has been reported to be involved in erythroid differentiation. GATA1 plays a significant role in regulating the transcriptional aspects of erythroid maturation and function[50]. GATA2 is verified to regulates hematopoietic stem cell proliferation and differentiation[51]. LMO2 is a bridging molecule assembling an erythroid, DNA-binding complex[52]. These findings demonstrate the potential of TFvelo in detecting important regulatory genes from single cell data.

### TFvelo can achieve competitive performance compared to Multivelo using only scRNA-seq data

Next we apply TFvelo to a 10x multi-omics embryonic mouse brain dataset, where both Assay for Transposase-Accessible Chromatin with sequencing (ATAC-Seq)[53] and scRNA-seq are available. This dataset was employed in Multivelo study, which is an RNA velocity model designed for multi-omics datasets, capturing dynamics between chromatin accessibility and unspliced mRNA[12]. On this dataset, as described in the Multivelo study, Radial glia (RG) cells in the outer subventricular zone can generate neurons, astrocytes, and oligodendrocytes. The development of cortical layers follows an inside-out pattern during neuronal migration, with younger cells ascending to upper layers while older cells remain in deeper layers. RG cells can divide, producing intermediate progenitor cells (IPC) that act as neural stem cells and give rise to various mature excitatory neurons in cortical layers.

The pseudotime inferred by TFvelo closely resemble Multivelo's results (Fig. 5a, b), and both methods correctly identify the differentiation direction in most cells. Notably, TFvelo achieves this performance without using ATAC-seq data. Figure 5c shows the phase portrait fitting of both methods, where Multivelo could show an additional c-u phase portrait, reflecting the joint plot between ATAC-seq and unspliced mRNA.

Regarding the AHI1 gene, Multivelo mistakenly constructs a cyclic dynamic, in contrast, TFvelo correctly captures the dynamic, with expression increasing consistently from V-SVZ cells (in green). For the NTRK2 gene, TFvelo captures such dynamics correctly, while Multivelo misses the decreasing process at the beginning. On the GRIN2B gene, both TFvelo and Multivelo detect the same dynamics, with expression increasing from the beginning to the terminal stage. These results suggest that by extracting features from multiple TFs, TFvelo can achieve a better fit for the dynamics of individual genes than Multivelo, without using ATAC-seq.

Although ATAC-seq could provide additional information for RNA dynamics modeling, it could be challenging due to multiple issues. For instance, the high sparse and near binary ATAC-seq data[54] could lead to the mixing of cells from different types, posing a challenge for phase portrait fitting. As shown in the first column at Fig. 5c, even the ATAC/unspliced/spliced 3D phase portrait can not provide enough information for dynamic modeling. Specifically, the c-axis (ATAC) could not help separate cells of different types (second column in Fig. 5c). Additionally, peaks obtained by ATAC are more challenging to directly link to a specific gene. By comparison, TFvelo can directly link multiple TFs into modeling a target gene. As a result, TFvelo could extract dynamics that align with the differentiation process well in TFs-target phase portraits (fourth column in Fig. 5c).

### TFvelo can predict the cell fate on dataset without splicing information

Next, TFvelo was applied to another single-cell RNA-seq dataset comprising 1,529 cells obtained from 88 human preimplantation embryos ranging from embryonic day 3 to 7[55]. Since the raw sequencing file is not provided on the online dataset server (see Data Availability Statement), those RNA velocity methods relying on the spliced/unspliced information are not feasible. By contrast, TFvelo still

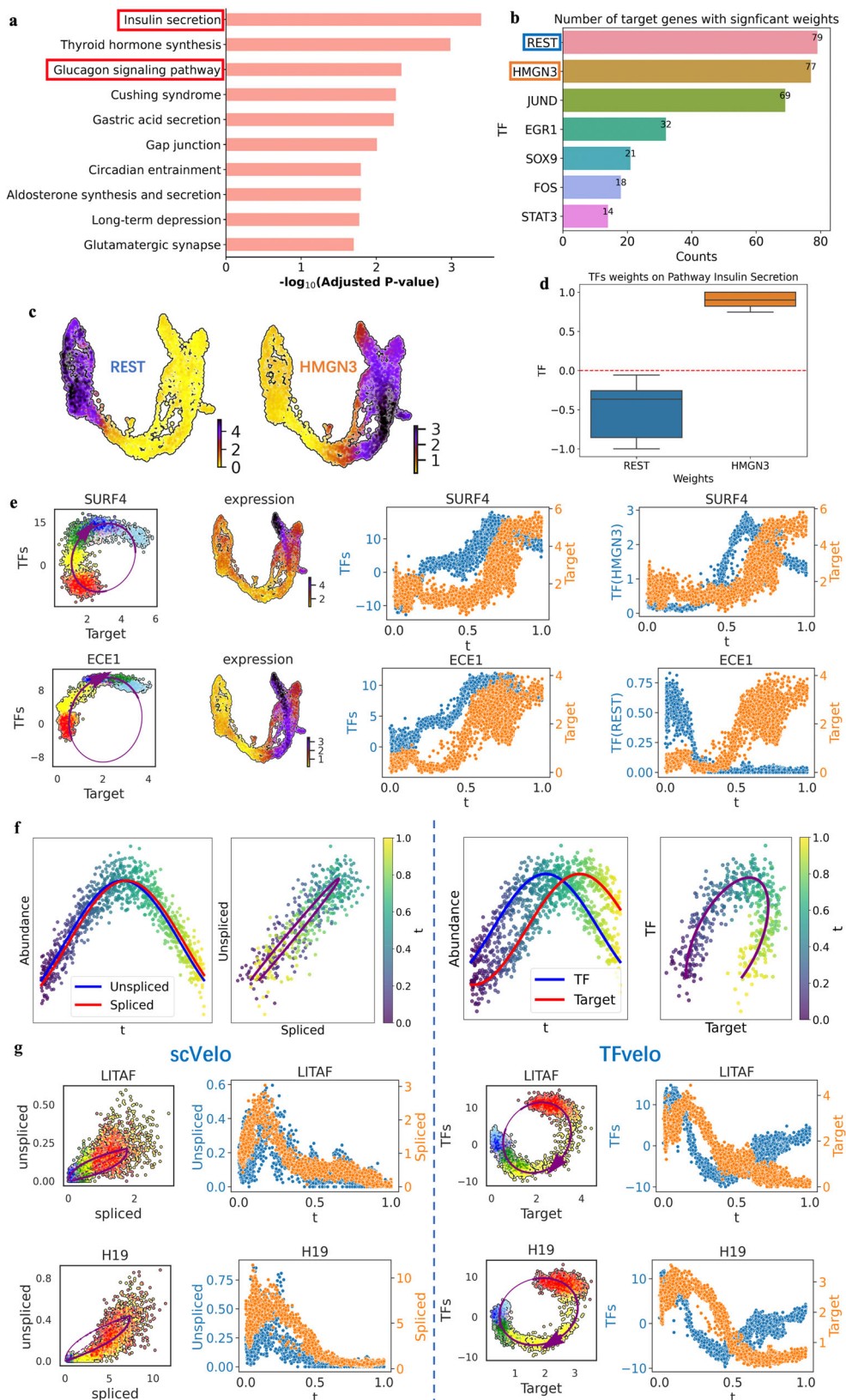

successfully identifies the pseudo-time and developmental trajectory of the cells, as depicted in Fig. 6a, b. Additionally, dynamic models successfully fit the expression patterns of target genes, which are illustrated in Fig. 6d as examples. In details, the dynamics predicted on PPP1R14A reveals a clockwise curve on TFs-target plot, and finds the changing point between the increasing and decreasing of expression

level at day 5 post-fertilization. Consistent with the TFvelo results, it has been reported that the expression of PPP1R14A can be detected with high dynamics that it increases at the beginning and declines throughout the development in the early stage of zebrafish embryos[56]. TFvelo finds that the expression of RGS2 is the highest at the beginning, and decreases with the pseudo-time, which can be supported by

**Fig. 3 | Biological analysis based on the results of TFvelo on pancreas dataset.**
**a** The KEGG pathway enrichment from the genes best fitted by TFvelo on pancreas dataset. The adjusted *p*-value is computed using the Fisher exact test. **b** List of TFs having high weights on multiple target genes. The counts mean the number of target genes where the TF have a normalized weight whose absolute value is larger than 0.5. **c** The distribution of REST and HMGN3 on UMAP. **d** The weights distribution of TFs REST and HMGN3 on modeling in the genes that in the insulin secretion pathway. The down, central and up hinges correspond to the first quartile, median value and third quartile, respectively. The whiskers extend to 1.5× the interquartile range of the distribution from the hinge. The numbers of samples for

the boxplots corresponding to REST and HMGN3 are 7 and 5, respectively. **e** Phase analysis on two example genes. On each row, from the left to the right: the phase portrait fitting, the distribution of abundance on UMAP, the value of learned TFs representation (WX) and the target gene along with pseudotime, and the abundance of a TF and the target gene along with pseudotime. **f** In simulation, the visualization of two variables where the time delay between them is very short or relatively long. The colorbar reflects pseudotime. **g** Comparison of the phase portrait fitting and expression level along pseudotime obtained by scVelo and TFvelo on two example genes. Source data are provided as a Source Data file.

the early findings that RGS2 plays a critical role in early embryo development[57].

## Discussion

The recent development in RNA velocity approaches provide new insights for temporal analysis of single cell data. However, the existing methods often fails to fit the cell dynamics well in the unspliced/ spliced space, which is partially because of the insufficient signal and high sparsity and noise. Motivated by the insight that the phase delay between TFs and target can also provide temporal information, we propose TFvelo to estimate the gene dynamics based on TFs abundance. Different from previous approaches which require the splicing information from the data, TFvelo models the dynamics of a target gene with the expression level of TFs and the target gene itself. In the computation framework, a generalized EM algorithm is adopted, so that the latent-time, the weight of TFs and those learnable parameters in the dynamic function can be updated alternately in each iteration.

Experiments on a synthetic dataset validate that TFvelo could detect the underlying dynamics from the data. Results on various scRNA-seq datasets and a 10× multi-omics dataset further demonstrate that TFvelo can fit the gene dynamics well, showing a desired clockwise curve on the phase portrait between the TFs representation and the target. Furthermore, TFvelo can be used to infer the pseudo time, cell trajectory and detect key TFs which play important role in the differentiation. In addition, compared with the previous RNA velocity model relying on splicing information, TFvelo can achieve better performance on those scRNA-seq datasets, and be flexibly applied to more categories of datasets without splicing information.

Due to the requirement of learning a representation of TFs, the time efficiency of TFvelo is lower than baseline approaches in the same framework, e.g., scVelo. This could be a weakness when being applied to the large-scale datasets. Performing TFvelo with more CPUs in parallel could reduce the running time. Meanwhile, TFvelo still cannot fit the dynamics of some genes well (Fig. S14). Improving the modeling for more genes remains a challenging task that requires further exploration. In the future, more complex models may be explored to replace the current linear model in TFvelo for estimating the weight of each TFs. The employed sine function for describing expression dynamics could also be replaced by other functional forms flexibly. In addition, those recently proposed ideas for improving unspliced/ spliced-based RNA velocity models can also be adopted in the further development of TFvelo, including stochastics modeling with Bayesian inference[14], multi-omics integration[22] and universe time inference[13].

## Methods
### Data preprocessing
We utilize the data preparation procedure with the default setting in scVelo[10], where the difference is that TFvelo is applied to the total mRNA counts, which is the sum of unspliced and spliced counts. Additionally, after filtering out those genes that can be detected from fewer than 2% of cells, and selecting the top 2000 highly variable genes (HVGs), we normalize the expression profile of these HVGs by the total count in each cell. A nearest-neighbor graph (with 30 neighbors by default) was calculated based on Euclidean distances in principal

component analysis space (with 30 principal components by default) on log-transformed spliced counts. After that, first-order and second-order moments (means and uncentered variances) are computed for each cell across its nearest neighbors. These steps are implemented in the same way as scv.pp.filter_and_normalize() and scv.pp.moments().

Additionally, to find potential transcriptional relationship within the selected 2000 highly variable genes, we refer to the TF-target pairs annotated in ENCODE TF-target dataset[38] and ChEA TF-target dataset[58]. If the regulatory relationship between a TF-target pair is labeled in at least one of these two datasets, the TF will be included in the involved TFs set when modeling the dynamic of the target gene.

### The computational framework of TFvelo
Given a gene $g$, TFvelo assumes that the RNA velocity $\frac{dy_g(t)}{dt}$ is determined by the expression level of involved TFs $\mathbf{X}_g$ and the target gene itself $y_g$, which is $\frac{dy_g(t)}{dt} = h(\mathbf{X}_g(t), y_g(t); \Psi_g)$. Using a top-down strategy[13], TFvelo directly designs a profile function of target gene's expression level to relax the gene dynamics to more flexible profiles, that is $y_g(t) = f(t; \Phi_g)$. $\Psi_g$ and $\Phi_g$ are two sets of gene-specific, time-invariant parameters, which control the influence of TFs on the target gene, and the shape of target gene's phase portraits, respectively. Our findings and early studies[39] have shown that it is feasible to infer the regulation relationship according to the learned parameter in linear regression models based on expression data. Additionally, linear models always have high interpretability, and rarely suffer from over-fitting. As a result, to show the capability of modeling RNA velocity based on gene regulatory relationship, we simply adopt a linear model as $h(\mathbf{X}_g(t), y_g(t); \Psi_g)$, which can map the expression levels of multiple TFs $\mathbf{X}_g \in R^{n_{TF}}$ from $n_{TF}$-dimensional space to 1-dimensional representation $\mathbf{W}_g \mathbf{X}_g$ with a weight vector $\mathbf{W}_g$, where $n_{TF}$ represents the number of involved TFs. As a result, the dynamic equation between TFs and the target gene $g$ can be written as

$$\frac{dy_g(t)}{dt} = h\left(\mathbf{X}_g(t), y_g(t); \Psi_g\right) = \mathbf{W}_g \mathbf{X}_g(t) - \gamma_g y_g(t), \tag{4}$$

where $t \in [0,1)$ is the latent time assigned to each cell, $\gamma_g$ is the degradation rate.

Besides, The profile function of $y_g(t)$ can be chosen flexibly from those second-order differentiable functions[13]. For simplification, $y_g(t)$ is designed with a sine function in our implementation, which is high-order differentiable and suitable for capture the curves on phase portrait. In addition, different with those models which assume the gene expression should follow a pattern of initial improvement followed by a decrease (or part of this whole process), TFvelo is more flexible and can also model gene dynamics where expression initially decreases and later increases with the sine function.

$$y_g(t) = f\left(t; \Phi_g\right) = \alpha_g \sin\left(\omega_g t + \theta_g\right) + \beta_g, \tag{5}$$

where $\alpha_g, \beta_g, \theta_g$ are gene-specific parameters to be learned. $\omega_g$ is fixed as $2\pi$, so that the $y_g(t)$ is unimodal within $t \in [0,1)$, which follows the common assumption shared by most current approaches. By employing the sin function, TFvelo can model the dynamics of each gene

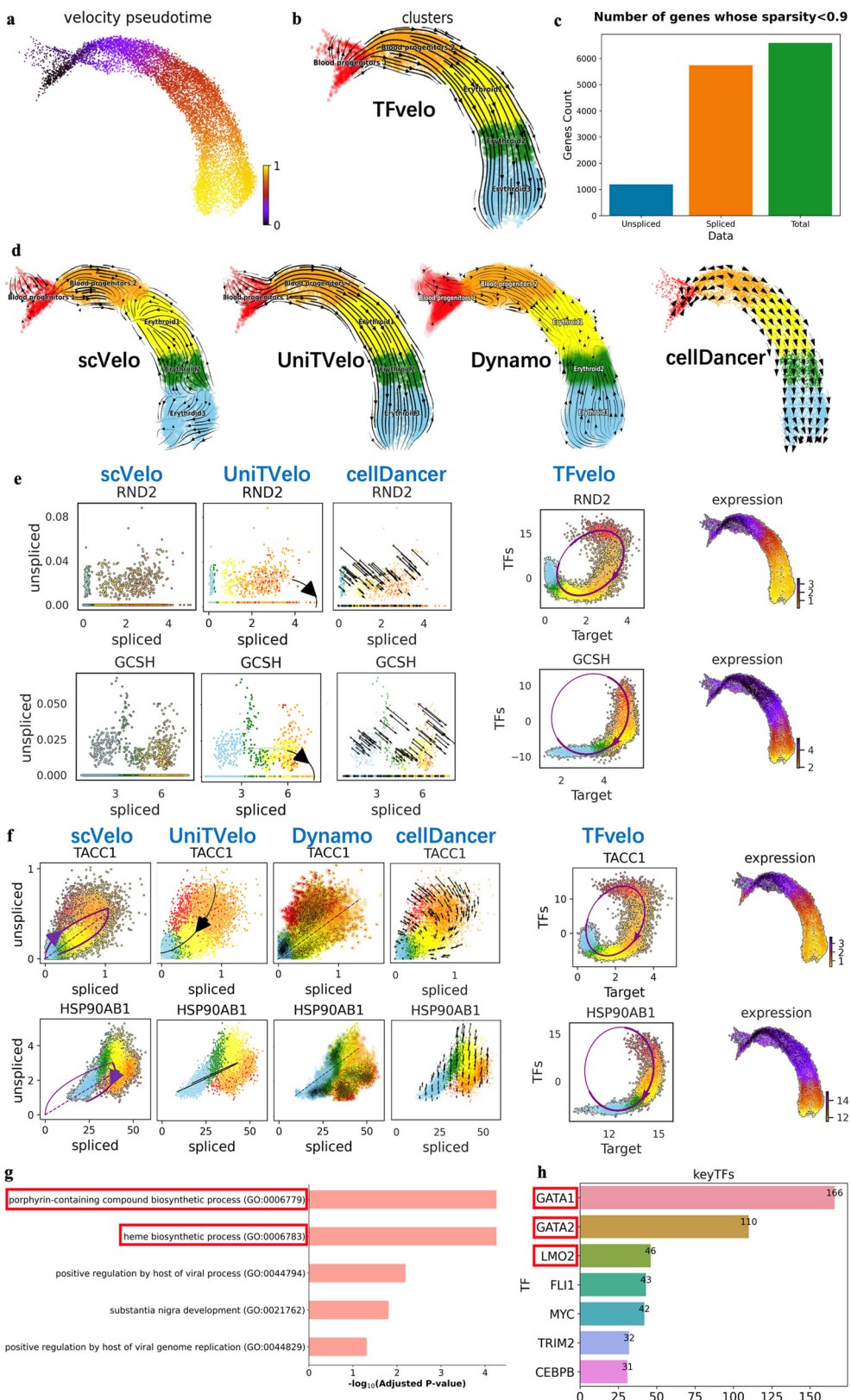

**Fig. 4 | Results of TFvelo on gastrulation erythroid dataset. a** Pseudo-time inferenced by TFvelo projected into a UMAP-based embedding. **b** The stream plot obtained by TFvelo. **c** The sparsity for unspliced, spliced and total mRNA counts. The sparsity of a gene is defined as:

$$\text{Sparsity} = \frac{\text{The number of cells that the count of this gene is zero}}{\text{The total number of cells}}.$$ **d** The stream plot obtained by baseline approaches. Each plot is drawn according to the method's tutorial. **e** The phase portrait fitting on genes with sparse unspliced counts. **f** The phase portrait fitting on TACC1 and HSP90AB1 with sparse unspliced counts. **g** The GO term biological process enrichment based on the best fitted genes. The adjusted *p*-value is computed using the Fisher exact test. **h** The number of target genes where the absolute value of the weight larger than 0.5 for each TF. Source data are provided as a Source Data file.

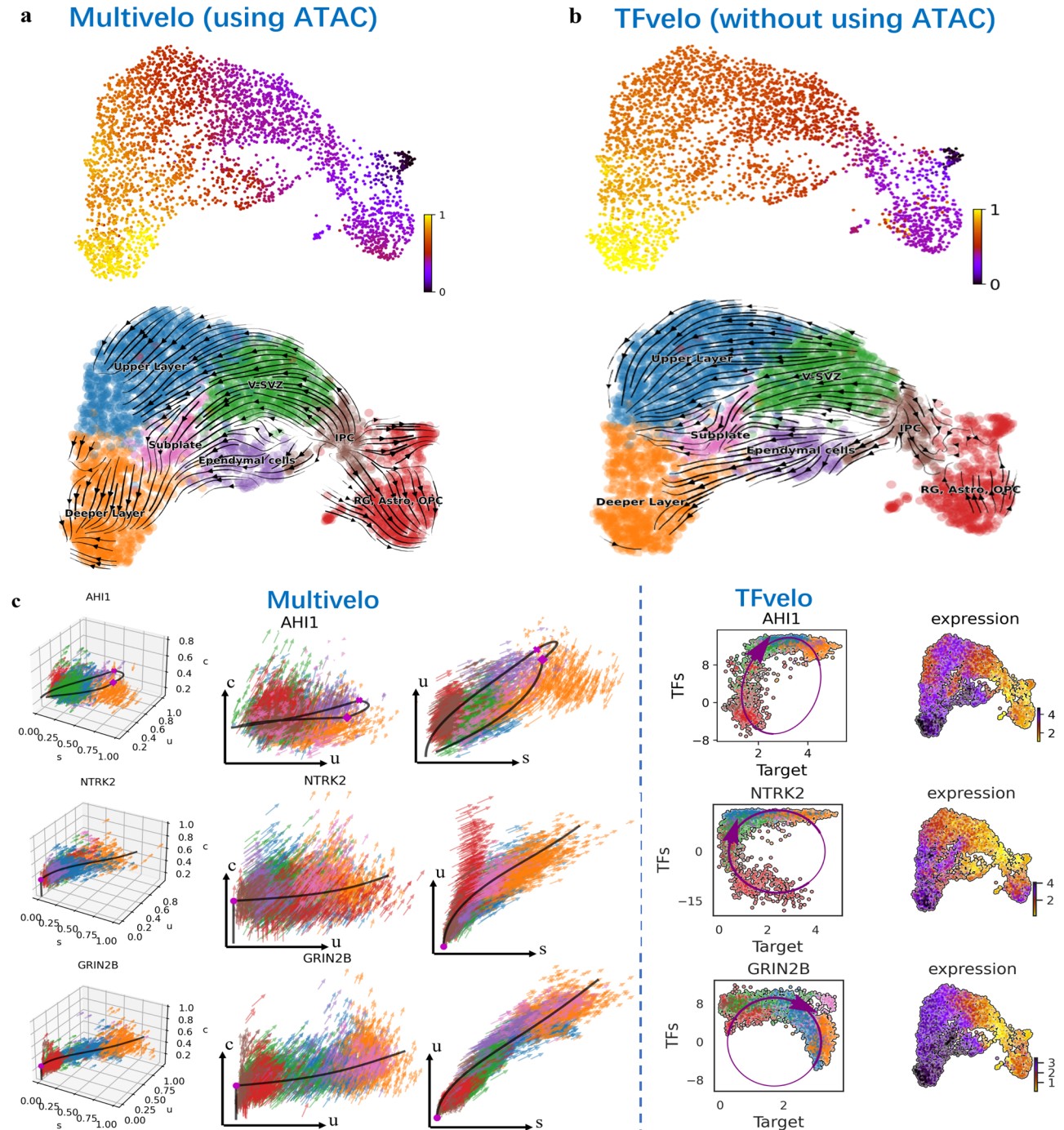

**Fig. 5 | Results of TFvelo on 10x multi-omics embryonic mouse brain dataset.**
**a** Pseudo-time and trajectory inferenced by Multivelo projected into a UMAP-based embedding. **b** Pseudo-time and trajectory inferenced by TFvelo projected into a UMAP-based embedding. **c** The comparison between Multivelo and TFvelo on three example genes, which are AHI1, NTRK2 and GRIN2B, respectively. For Multivelo plot, (**c**) means the chromatin accessibility in ATAC-seq.

without defining a switching time point between the of increasing and decreasing periods[9,10,14,18]. To optimize the parameters in TFvelo, a generalized EM[42,43] algorithm is adopted, as discussed in the following.

**Optimization of generalized EM algorithm**
To simplify the mathematical representation, here we denote the $\mathbf{W_g}, \mathbf{X_g}, y_g, \alpha_g, \beta_g, \theta_g, \gamma_g$ and $\omega_g$ as $\mathbf{W}, \mathbf{X}, y, \alpha, \beta, \theta, \gamma$ and $\omega$, respectively. As a result, the Eqs. (5, 4) can be written as

$$y(t) = \alpha \sin(2\pi t + \theta) + \beta, \qquad (6)$$

$$\frac{dy(t)}{dt} = \mathbf{W}\mathbf{X}(t) - \gamma y(t). \qquad (7)$$

From the dynamics model in Eqs. (6, 7), we can derive that,

$$\mathbf{W}\mathbf{X}(t) = \alpha\sqrt{4\pi^2 + \gamma^2}\sin(2\pi t + \theta + \phi) + \beta\gamma, \ \phi = \arctan\left(\frac{2\pi}{\gamma}\right), \qquad (8)$$

For modeling a gene, suppose observations $\mathbf{X_c^{obs}}$ and $y_c^{obs}$ be the normalized expression levels of TFs and the target gene within a cell $c$, $s_c^{obs}$ be the state space representation of the cell $c$, and $\hat{s}(t_c)$ be the

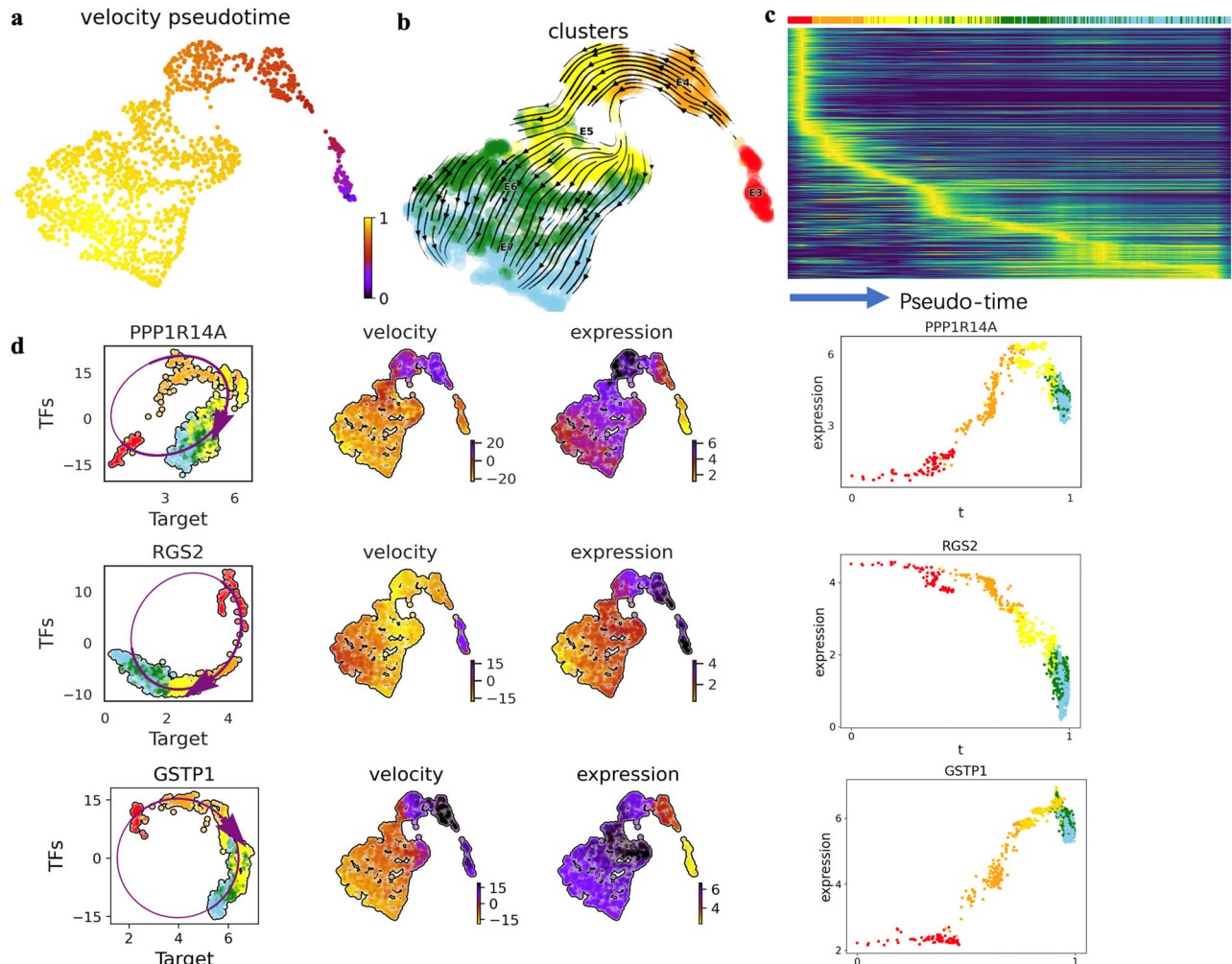

**Fig. 6 | Results of TFvelo on dataset without splicing information. a** Pseudo-time inferenced by TFvelo projected into a UMAP-based embedding. **b** The streamplot of UMAP. **c** The heatmap which shows the dynamics of gene expression resolved along pseudo-time in the top 300 likelihood-ranked genes. **d** The constructed dynamic models on three example genes with high likelihoods, which are PPP1R14A, RGS2 and GSTP1, respectively. From the left to the right within each row, the three subfigures show phase portrait fitting, distribution of velocity on UMAP, distribution of expression on UMAP, and the dynamics of expression resolved along with pseudo-time, respectively.

modeled state at time $t_c$, we have $s_c^{obs} = [\mathbf{W}\mathbf{X}_c^{obs}, y_c^{obs}]$ and $\hat{s}(t_c) = [\mathbf{W}\mathbf{X}(t_c), y(t_c)]$. Aiming at finding the phase trajectory $\hat{s}(t)$ that best fits the observations for all cells, we define the loss function as:

$$L = \sum_c \left\| s_c^{obs} - \hat{s}(t_c) \right\|^2 = \sum_c \left( \left(\mathbf{W}\mathbf{X}_c^{obs} - \mathbf{W}\mathbf{X}(t_c)\right)^2 + \left(y_c^{obs} - y(t_c)\right)^2 \right),$$

(9)

The residuals of the observations are defined as $e_c = sign(y_c^{obs} - y(t_c))||s_c^{obs} - \hat{s}(t_c)||^2$, which is assumed to follow a normal distribution $e_c \sim N(0,\sigma^2)$, where the gene-specific $\sigma$ is constant across cells, and the observations are independent and identically distributed. Then we can arrive at the likelihood function,

$$\mathcal{L}(\mathbf{W},\alpha,\beta,\theta,\gamma) = \frac{1}{\sqrt{2\pi}\sigma} * \exp\left(-\frac{1}{2\sigma^2}\sum_c ||s_c^{obs}(\mathbf{W}) - \hat{s}_{t_c}(\mathbf{W},\alpha,\beta,\theta,\gamma)||^2\right),$$

(10)

Where $\hat{s}_{t_c}(\mathbf{W},\alpha,\beta,\theta,\gamma)$ represents $\hat{s}(t_c;\mathbf{W},\alpha,\beta,\theta,\gamma)$. Finally, the optimization algorithm aims to minimize the negative log-likelihood

function:

$$l(\mathbf{W},\alpha,\beta,\theta,\gamma) = -\log\left(\frac{1}{\sqrt{2\pi}\sigma}\right) + \frac{1}{2\sigma^2}\sum_c ||s_c^{obs}(\mathbf{W}) - \hat{s}_{t_c}(\mathbf{W},\alpha,\beta,\theta,\gamma)||^2,$$

(11)

In standard EM algorithms, the E-step computes the expected value of likelihood function given the observed data and the current estimated parameters to update the latent variables, and the M-step consists of maximizing that expectation computed in E step by optimizing the parameters. While under some circumstances, there is an intractable problem in M step, which could be addressed by generalized EM algorithms. Expectation conditional maximization is one form of generalized EM algorithms, which makes multiple constraint optimizations within each M step[42]. In detail, the parameters can be separated into more subsets, and the M step consists of multiple steps as well, each of which optimizes one subset of parameters with the reminder fixed[43].

In TFvelo, all learnable parameters can be divided into three groups, which are cell-specific latent time $t \in R^{n_{cell}}$, weight of each TF $\mathbf{W} \in R^{n_{TF}}$ and scalars $[\alpha,\beta,\theta,\gamma]$ respectively, where $n_{cell}$ and $n_{TF}$

represent the number of cells and the number of involved TFs. Generalized EM algorithm is adopted so that the three group of parameters will be updated alternately to minimize the loss function in each iteration:

1. Assign $t$ within $[0, 1]$ for each cell by grid search. For each cell, on the phase plot, the closest point located on the curve described by the dynamic model is defined as the target point. The step (b) and (c) aim to minimize the average distance between cells and their corresponding target points.
2. Update $W$ by linear regression with bounds on these weights to minimize the loss function. Trust region reflective algorithm[59], which is an interior-point-like method is adopted to solve this linear least-squares problem with inequality constraint. In our implementation, the weights are bounded within $(-20, 20)$ by default.
3. Update $[\alpha, \beta, \theta, \gamma]$ with constraints of $\alpha \in (0, \infty)$, $\theta \in (-\pi, \pi)$ and $\gamma \in (0, \infty)$ to minimize the loss function.

By iteratively running the three steps, parameters of TFvelo for each target gene can be optimized. Furthermore, multiple downstream analysis could be conducted by combining the learned dynamics of all target genes.

## Initialization of the generalized EM algorithm

Since EM algorithms could be trapped in local minimum, parameters are initialized in the following steps. Firstly, those cells in which expression of target gene is 0 are filtered out. Then $\mathbf{X}, y$ are normalized by $\mathbf{X} = \frac{\mathbf{X}}{\text{var}(\mathbf{X})}$, $y = y / \text{var}(y)$, where $\text{var}(\cdot)$ represents the standard deviation function. The weights $\mathbf{W}$ is initialized with the value of spearman correlation of each TF-target pair. For parameters $\alpha, \beta$ and $\gamma$, according to the steady-state assumption that $\dot{y}|_{y = \max(y)} = 0$, $\gamma$ can be initialized by $\gamma = \frac{\mathbf{WX}}{y}|_{y \rightarrow y_{\max}}$. At the points of maximal and minimal $y$ value, we have $y_{max} = \alpha + \beta$ and $y_{min} = -\alpha + \beta$. As a result, $\alpha$ and $\beta$ can be initialized by $\alpha = \frac{y_{max} - y_{min}}{2}$ and $\beta = \frac{y_{max} + y_{min}}{2}$, respectively. These assumptions are only used for initialization and in the following optimization process, all parameters will be updated iteratively. The hyper-parameter $\omega$ is set as $2\pi$ and parameter $\theta$ is intialized with various values, so that the generalized EM algorithm will run at different start points. After that, the model with the lowest loss is finally selected.

## Inference of Pseudo-time

After constructing the dynamic model for each gene, TFvelo will infer a gene-agnostic pseudo-time for each cell based on the combination of RNA velocities of all genes. For this purpose, we first compute the cosine similarities between velocities and potential cell state transitions to get the transition matrix. Then the end points and root cells are obtained as stationary states of the velocity-inferred transition matrix and its transposed, respectively. After inferring the root cells' location on the embedding space, velocity pseudo-time, which measures the average number of steps it takes to reach a cell after start walking from the root cell, can be computed. The strategy of calculating pseudo-time is the similar with that in scVelo[10]. The different is that in scVelo, the transition matrix is obtained based on the abundance and velocity of spliced mRNA, while in TFvelo that is based on the abundance and velocity of total mRNA. Details are provided in the Supplementary Information section 6 titled "The root and end cells detection".

## Velocity streams on 2D embedding space

To illustrate cell transitions, we create stream plots in a 2D space using UMAP visualization as the default. To this end, we select genes whose loss is low and gene-specific time aligns with the pseudotime to construct velocity streams.

For the single cell datasets, we first select genes with modeling error in the bottom 50%. Since sine functions exhibit periodicity, we need to determine the order of cells instead of using latent time directly. We achieve this by analyzing the histogram of latent time. If there are several consecutive blank bins in the cell distribution of latent time, we set the minimum value of the next non-blank bin as the initial state (normalized latent time = 0). Conversely, the maximum value of the last non-blank bin will be set as the final state (normalized latent time = 1). The normalized latent time of all other cells will be mapped within the range of 0 to 1. Subsequently, we select genes where the normalized latent time aligns with the pseudotime based on the Spearman correlation coefficient between them.

## Metrics for evaluating in phase portrait

We propose three metrics for phase portrait fitting.

1. The intra-class distance on phase portrait, which measures the normalized distance between cells within the same type on the phase portrait. This reflects whether cells within the same type gather on the phase portrait, which is the lower the better. On unspliced/spliced based methods, the intra-class distance is defined as $Dist = \sum_k \sum_{j \in type_k} \left[ \left( \frac{u_j - u_c^k}{std(u)} \right)^2 + \left( \frac{s_j - s_c^k}{std(s)} \right)^2 \right]$, where $(u_c^k, s_c^k)$ means the center of cell type $k$, std(.) means the standard variance and $j$ means the index of a cell. For TFvelo, the distance can be defined by substituting $u$ with $WX$ and $s$ with $y$.

2. The inter-class distance on phase portrait, which measures the normalized distance between the distribution centers of different cell types on the phase portrait, which is defined as $Dist = \sum_{k1 \neq k2} \left[ \left( \frac{u_c^{k1} - u_c^{k2}}{std(u)} \right)^2 + \left( \frac{s_c^{k1} - s_c^{k2}}{std(s)} \right)^2 \right]$. This reflects whether cells from different types are separated well on the phase portrait, which is the higher the better.

3. The fitting error on phase portrait, which measures the normalized distance between each cell to the constructed model on the phase portrait. This reflects the fitting accuracy and calculated in the following way. On the phase portrait of each selected gene, calculate the normalized mean square error of the distance between each cell to the inferred model curve. For the TFvelo data, $Err = \sum_{j=1}^N \left[ \left( \frac{\mathbf{WX}_j - \mathbf{WX}(t_j)}{std(\mathbf{WX})} \right)^2 + \left( \frac{y_j - y(t_j)}{std(y)} \right)^2 \right]$, where $j$ means the index of a cell, $t_j$ means the latent time modeled for cell $j$ and $N$ is the total number of cells. $\mathbf{WX}_j$ and $y_j$ refer to the location of cell on the phase portrait. $\mathbf{WX}(t_j)$ and $y(t_j)$ refers to the model given by TFvelo. Similarly, the error for un/spliced data-based methods is defined as $Err = \sum_{j=1}^N \left[ \left( \frac{u_j - u(t_j)}{std(u)} \right)^2 + \left( \frac{s_j - s(t_j)}{std(s)} \right)^2 \right]$.

## Metrics for evaluating the velocity stream

1. Cross boundary direction correctness (CBDir) and within-cluster velocity coherence (ICVCoh). According to the definition given in VeloAE[11] and UniTVelo[13], CBDir evaluates the accuracy of transitions from a source cluster to a target cluster based on the information provided by boundary cells, which requires the ground truth annotation. ICVCoh is computed with a cosine similarity scoring function between cell velocities within the same cluster. We directly run the functions provided in UniTVelo[13] to obtain the of CBDir and ICVCoh.

2. Velocity consistency. We executed scVelo's velocity_confidence() function and interpreted the outcomes as velocity consistency. This is because that it indeed measures the consistency of velocities within neighboring cells, as opposed to the statistical definition of "confidence".

## GO term and KEGG pathway enrichment analysis

To perform the GO term and KEGG pathway enrichment analysis, we utilized the "gseapy.enrichr()" function in Python package "gseapy" with using Fisher's exact test by default. For each dataset, the background gene list consists of all genes within the original dataset before preprocessing.

## Weights normalization and TFs selection

Considering that the abundance of different TFs could vary a lot, the quantitative comparison between the weights of TFs is conducted after normalizing them. For each TF on modeling a target, we multiply the mean expression level by the learned weights to get the average influence of the TF to the target gene. For instance, the average influence of TF $i$ is $w_i * \bar{x}_i$. Then the normalized weights are defined as: $\widetilde{w}_i = \frac{w_i * \bar{x}_i}{\max_i(|w_i * \bar{x}_i|)}$. The TFs with high weights are select according to these normalized weights with default threshold of 0.5.

## Statistics & reproducibility

Four datasets are utilized in this manuscript, which is a proper number compared with relevant RNA velocity studies. No statistical method was used to predetermine sample size. No data were excluded from the analyses. The experiments were not randomized. The Investigators were not blinded to allocation during experiments and outcome assessment.

To reproduce the results shown in this paper, please use the demo code with default parameters at the GitHub repository, https://github.com/xiaoyeye/TFvelo.

## Reporting summary

Further information on research design is available in the Nature Portfolio Reporting Summary linked to this article.

## Data availability

All relevant data supporting the key findings of this study are available within the article and its Supplementary Information files. The pancreatic endocrinogenesis dataset[36] comprises the single-cell RNA-seq (10X) data of 27,998 genes of 3,696 pancreatic epithelial and Ngn3-Venus fusion cells sampled from mouse embryonic day 15.5. Data could be acquired from scvelo.datasets.pancreas(). The gastrulation erythroid dataset, which is selected from the transcriptional profiles of mouse embryos[48], provides expressions of 53,801 genes of 9,815 cells. This dataset is incorporated by scvelo.datasets.gastrulation_erythroid(). 10x embryonic mouse brain dataset can be accessed at the 10x website at https://www.10xgenomics.com/resources/datasets/fresh-embryonic-e-18-mouse-brain-5-k-1-standard-1-0-0. To ensure a fair comparison between TFvelo and Multivelo, TFvelo utilizes the same RNA data file as the one used in Multivelo (https://multivelo.readthedocs.io/en/latest/MultiVelo_Fig2.html), which consists of 3,365 cells and 936 genes. The preprocessed data used in this study is provided at https://github.com/xiaoyeye/TFvelo/blob/main/data/10x_mouse_brain/adata_rna.h5ad. Human preimplantation embryos dataset[55] is a single-cell RNA-seq dataset of 1,529 cells obtained from 88 human preimplantation embryos ranging from embryonic day 3 to 7. Data were downloaded with the accession number of E-MTAB-3929 from EMBL-EBI. In this dataset, only the RNA abundance is provided. As a result, those RNA velocity methods relying on the spliced/unspliced are not available. The ENCODE TF-target database[38] is available at: https://maayanlab.cloud/Harmonizome/dataset/ENCODE+Transcription+Factor+Targets. We also provide the preprocessed file that can be directly used in TFvelo at: https://github.com/xiaoyeye/TFvelo/blob/main/ENCODE.zip. The ChEA TF-target database[58] is available at: https://maayanlab.cloud/Harmonizome/dataset/CHEA+Transcription+Factor+Targets. We also provide the preprocessed file that can be directly used in TFvelo at: https://github.com/xiaoyeye/TFvelo/tree/main/ChEA. Source data are provided with this paper.

## Code availability

TFvelo is implemented in Python, based on the scVelo package[10]. The source code can be downloaded from the GitHub repository, https://github.com/xiaoyeye/TFvelo[60].

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

## Acknowledgements

This work was supported by the National Key R&D Program of China (2023YFF1204500 to Y.Y.), National Natural Science Foundation of China (No. 62073219 to H.S. and No. 62103262 to Y.Y.), and the Science and Technology Commission of Shanghai Municipality (No. 22511104100 to H.S.).

## Author contributions

Y.Y. and H.S. conceived and supervised the study. Y.Y. and J.L. proposed the computational model. J.L. developed the model and conducted data analysis. Y.Y., H.S. and X.P. provided advice on data analysis. J.L. drafted the manuscript. Y.Y. and H.S. revised the manuscript.

## Competing interests

The authors declare no competing interests.
