## [Peer Review File · Nature Communications]

TFvelo: gene regulation inspired RNA velocity estimationReviewer #1 (Remarks to the Author):

The authors claimed that "the RNA velocity can be represented as a linear combination of the expression levels of transcription factors (TFs)." RNA velocity is the information using both transcriptional and posttranscriptional regulation. The linear combination of TFs only can represent the transcriptional aspect, but not the posttranscriptional regulation. In this regard, Eq1 cannot hold. It is also not clear how the quantity of velocity is obtained by running scVelo.

Obviously, there is a phase delay between TF and the target genes. The information TFVelo uses is not RNA velocity but the dynamic relationships between TFs and their targets. It is the concept that TENET or SCRIBE uses to obtain causal relationships using transfer entropy. Therefore, it does make sense to evaluate its performance with TENET and SCRIBE rather than scVelo.

Reviewer #2 (Remarks to the Author):

Review of "TFVelo: gene regulation inspired RNA velocity estimation" (NCOMMS-23-16358)

Summary

The authors introduce TFVelo, an approach to estimate gene-wise velocities without using spliced/unspliced data. They argue that their approach includes the effects of gene regulation, is applicable to imaging data, where no spliced/unspliced count information is available, and accurately fits "clock-wise" phase portraits of TF-target gene relationships.

The key idea is to represent RNA velocity as a linear combination of TF expression levels. Rather than considering unspliced/spliced counts in phase space, TFVelo considers the expression level of a TF and its target(s). This is translated into one simple linear ODE per gene, that relates the velocity of that gene to the expression level of upstream TFs. The expression of the target gene itself is modeled as a sine function, and parameters are learned from data.

I have a number of concerns related to this work. First, the authors motivate their approach by emphasizing that they don't use unspliced/spliced count data. While I agree that this data source can be quite noisy in some situations, I don't think it is generally an improvement to have a method that does not make use of this additional information. Not using this information renders TFVelo a standard trajectory inference (TI) approach, applicable to the same data as all of the previous TI approaches. Accordingly, the authors should benchmark their method with approaches like PAGA, Monocle, Slingshot and Palantir, and demonstrate their advantages.

Second, the authors emphasize throughout the text that previous velocity approaches don't fit the data well. However, while many previous approaches are available, including scVelo, Velocity, LatentVelo, VeloVI, pyroVelocity, UniTVelo, etc., they only compare their approach with scVelo, and they don't actually show that scVelo does not fit the data well. If the authors include such a broad claim, they should demonstrate it explicitly, benchmark their approach with a number of popular velocity approaches, and explicitly show that these earlier approaches don't fit unspliced/spliced data well in phase space, in a quantitative manner.

Third, TFVelo estimates a velocity estimate and a latent time assignment per gene and cell, but the authors don't make use of this information directly in their data applications or benchmarks. Instead, they use scVelo to direct Markov-chain edges using their velocities and estimate scVelo's "velocity pseudotime" on this Markov chain. Next, they take derivatives of this pseudotime in UMAP space, apply heavy smoothing and post-processing, and visualize the resulting dynamics via streamlines in 2D UMAP. Given this heavy smoothing, their pseudotime and velocities are not comparable to scVelo's outputs, and I'm highly skeptical of the metrics they employ for comparisons, like the velocity confidence. Velocity confidence measures the extent to which neighboring velocities agree, given their extremely heavy smoothing, it comes as no surprise that they achieve high values in this metric - that is, however, in no way a validation of their approach.

Finally, in their implementation on GitHub, they mostly copy scVelo code, without appropriate, prominent acknowledgments of the source of this code. Even just from an implementation point of

view, this is bad practice: it creates a massive overload and makes it essentially impossible to maintain their code base. I would strongly advise against copying large parts of source code, it makes much more sense to add scVelo as a dependency and to import the required modules, e.g., for plotting, where needed. Also, for a publication in Nature Communications, I expect to see a minimal amount of documentation, testing, and tutorials, which is absent here.

Please see below for a detailed account of my comments.

Major comments

From a general point of view, TFVelo does not make use of the additional information provided by unspliced/spliced counts and instead fits TF-target gene relationships to uncover trajectories. As such, it's similar to other methods that infer trajectories based on scRNA-seq alone, such as Palantir, Monocle, PAGA, and Slingshot. It would be nice if the authors could outline the main advantages of TFVelo over these competing approaches, which work on the same input data. Of course, TFVelo can give additional information about gene regulation (even though I don't fully understand how this improves over model inputs), but the inference of cellular dynamics is still a claim the authors make, and this should be compared with competing approaches. Most pseudotime-based approaches yield information on gene dynamics. Additionally, in many of the systems the authors present here, including the pancreas or blood generation during mouse embryogenesis, the starting cell/state is known and can be provided to estimate pseudotime. Results, Line 113: "In addition, the TFs with non-zero positive weights are significantly enriched in TFs set functionally linked to the target gene". That appears to be a circular argument as the authors started out with a set of candidate TFs for each gene already. Is this a sanity check? The authors argue that models like scVelo and velocityto don't fit the observed co-expression of spliced/unspliced counts well, see e.g. in the intro. However, to motivate their model, they show that LASSO regression based on upstream TF expression (first section of the Results) can recover scVelo velocities - how much of a selling point is that, given that the authors argue against these very velocities?

Results, Line 127: "Fig. 1a provides an example on modeling the dynamics of gene LITAF, to explain why TFVelo outperforms existing methods (scVelo is used as the representative). " If the authors show a comparison with one method, scVelo, they cannot argue that their method outperforms all other existing methods. They don't show that here.

Results, Line 130: "(...) while transcriptional dynamics of RNA splicing may not provide sufficient signal". This is a really central claim of this paper that they use as motivation to develop their new approach. However, I'm currently missing any quantitative evaluation of this claim, they just cite (Bergen et al. 2021).

Results, Lines 195-200, the authors evaluate the performance of their method on data that has been simulated using a forward model that matches their assumptions. Unsurprisingly, they achieve low MSE loss. However, this is hard to judge, it would be helpful to include some baseline here so that the performance of this model can be compared to something.

Results, Line 205, "The pseudo-time predicted by TFVelo can identify the cell trajectory (Fig. 2a) well (...)". If I understood the suggested model correctly, then each target gene is fitted independently. Accordingly, I would expect to have one latent/pseudo-time per target gene. The authors describe in the Methods how the actual pseudotime is computed, it's not an aggregation over individual gene-specific latent times, but rather some DPT-like (Haghverdi et al. 2016) time inferred from the velocity-inferred transition matrix. As with the velocities, given that this method infers a time per gene, I would like to see an actual evaluation of these times (quantitatively). In scVelo, one latent time is estimated per gene, and these are pooled post-hoc to estimate a cell-specific latent time.

Results, Lines 206-207: " (...) and the arrows are drawn to reflect the direction of cells development (Fig. 2b, Methods) using the derivative of pseudo-time.". I would strongly discourage the use of 2D stream plots to compare the performance of velocity methods. It has been demonstrated in numerous previous publications (e.g. (Bergen et al. 2021; Lange et al. 2022; Marot-Lassauzaie et al. 2022)) that these low-dimensional representations are often misleading. Alternatively, the authors could use methods that directly compare high-dimensional velocity vector fields, e.g. CellRank (Lange et al. 2022) or Dynamo (Qiu et al. 2022). In addition, I don't quite understand why the authors consider the derivative of pseudotime here to visualize cell transitions, as their model directly learns a velocity per cell per gene. Why not make use of that

information directly? Given that this method estimates velocities, I would like to see an actual, quantitative evaluation of those velocities.

The problem of using low-dimensional velocity streams to judge the quality of their approach appears a few more times in the text, e.g. Results, Line 212: "In addition, compared to scVelo, TFVelo's result on Epsilon and Delta cells can better align with the cell differentiation process". In my opinion, 2D representations should not be used to judge the quality of their approach compared to previous approaches. Instead, in my opinion, the authors should use higher-dimensional, more robust means to compare differentiation trajectories throughout the Results section.

Results, Line 209, "TFVelo can even achieve a higher velocity confidence than scVelo without using splicing information (Fig. 2f), which indicates a higher consistency of RNA velocities across neighboring cells in the UMAP space". I do not think it is a good idea to use a 2D UMAP space for quantitative comparison of projected velocities. If the authors would like to use the velocity confidence metric to compare their approach to previous approaches, I would suggest considering neighborhood relationships in higher-dimensional spaces, and not in the 2D UMAP. Also, as I outline further down below, I find this comparison biased towards TFVelo, as the authors apply very heavy post-processing to their velocities, which is likely to result in misleading, overly-smooth 2D representations.

Results, Line 226, "By comparison, scVelo fails to detect any dynamics because of the insufficient information provided by unspliced mRNA". This could potentially be a strong selling point of TFVelo. Insufficient amounts of unspliced counts in important genes are an actual limitation of current velocity models that rely on sufficient levels of unspliced transcripts. Insufficient levels of unspliced counts have been explored a bit in the literature (Gorin et al. 2022; Sonesson et al. 2021), the authors should cite these works and explore this further. Being able to estimate robust velocities in the presence of very noisy unspliced count abundance would be a real selling point for TFVelo, I think.

Results, Line 233, "By comparison, scVelo cannot distinguish cells from multiple state on H19". By looking at the phase portrait, it seems like scVelo detected a down-regulation in this gene, which is in line with what TFVelo predicts. I don't understand what the authors refer to, this should be made explicit and quantified. Same for LITAF, seems like both scVelo and TFVelo fit a transient up-regulation. And also, for ECE1, I cannot follow the author's claims of better fits by their model. Based on my comments above, I disagree with the authors conclusion in Line 242: "These results suggest that TFVelo can provide more reasonable, robust and accurate dynamics for genes from single cell data.". I'm unsure what the authors mean by "reasonable"; "robustness" has not been quantified, as far as I'm aware, and "accurate" has also not been quantified, I think. None of the comparisons presented in Fig. 2 justify these statements, I think.

Results, Line 272, "The Gastrulation Erythroid dataset is the transcriptional profile for mouse embryo ". The data the authors use here represents a small subset of the original data presented in (Pijuan-Sala et al. 2019). The authors should mention this, and describe why they focus on this particular subset of the data. They should avoid the impression of cherry-picking that subset of the original data where their model yields nice results.

Results, Lines 369-379. In principle, it would be possible to distinguish between nuclear and cytoplasmic mRNA in MERFISH data and to use this information to fit a scVelo-type model, where nuclear and cytoplasmic counts take the role of unspliced and spliced transcripts, respectively. It would be nice if the authors could discuss this alternative approach, and outline potential drawbacks as well as advantages of their approach.

Discussion, Lines 423-426: "Compared with the previous RNA velocity model relying on splicing information, TFVelo can achieve better performance on those scRNA-seq datasets". I think this claim is too strong, given the analysis in the paper. Quantitative analysis was mostly done in terms of velocity confidence, which seems to rely on 2D UMAP information and overly heavy smoothing for TFVelo. Most of the other comparisons are qualitative, e.g. eye-balling where the streamlines point, and whether genes are visually up-or downregulated. To make such a broad claim in the discussion, the authors need to evaluate their model much more thoroughly against existing approaches, using robust metrics that operate in high-dimensional spaces. Also, given the wealth of velocity approaches that exist now (the authors list some of them in the intro), I wonder why they only compare with scVelo. There are known limitations of the scVelo velocity model (e.g. constant kinetic parameters, etc.), some of which have been partially resolved in follow-up models.

Supplementary Figure 2: "The stream derived from the pseudo-time is projected into the UMAP-

based embedding". The description of this appears in the Methods, Line 574-580. The authors don't work with the actual velocities their method estimates but rather use scVelo to estimate a "velocity pseudotime", which is a symmetrized version of DPT, and take derivatives of this pseudotime to draw stream-lines in a UMAP. It appears that they take derivatives of the pseudotime in the 2D UMAP space, which I would not advise doing, given the known pitfalls of 2D representations. Also, they apply further smoothing in the 2D UMAP by binning cells, and smoothing over neighboring bins. Given this extremely heavy post-processing and smoothing, the comparison with scVelo in terms of velocity consistency is biased towards TFVelo and no longer reliable. If the authors would like to use a metric like velocity confidence to compare with scVelo, they should compare quantities that are actually comparable, i.e., velocities estimated by either method (not pseudotime-derivatives), in the same space, with the same post-processing applied. The authors should comment on the scalability of their approach, compared to competing methods like scVelo.

Minor comments

Abstract: "most existing RNA velocity models can only be applied to datasets with unspliced/spliced or new/total RNA abundance information." Why would this be a limitation? Spliced/unspliced read abundance can be estimated from any standard scRNA-seq protocol, as shown already in (La Manno et al. 2018). It would be good to add a half-sentence, listing some examples where this information might not be available.

Intro: "However, it is cost and time-consuming to obtain these new omics data with additional labeling." I agree it would be costly to collect multi-modal single-cell data with additional metabolic labeling, but that's not the type of information required by the approaches outlined in the previous sentence: MultiVelo (Li et al. 2022) and Lior Pachter's (Gorin, Svensson, and Pachter 2020) Protein velocity models don't need labeling data.

Intro, Line 72: "Notably, the TFs with the positive weights are significantly enriched with TF set functionally linked to the target gene." What are these weights? At this point in the manuscript, the authors have not introduced any weights and this statement is confusing.

Results, Line 101. How are candidate TFs determined here? This is crucial and should be stated explicitly in the main.

Supplementary Figure 2: "The colorbar is feasible for all cell type annotations within this figure" There is no colorbar in this figure.

Results, Line 107 and 157. The definition of TF weights is not consistent, once it's w_g and once W_g . If the latter is meant to represent a matrix, that that's a contradiction with $W_g \in \mathbb{R}^{n_{TF}}$ (Line 157). I would urge the authors to use consistent definitions of all symbols throughout the main and methods, and to define these properly, including the correct shapes.

Also, I would be consistent in the use of bold-face for symbols. For example, w_g in Line 107 is bold-face, but t in Line 157 is not, even though it's shape $t \in \mathbb{R}^{n_{cell}}$ indicates that this is a vector. These inconsistencies make the mathematical modeling hard to follow.

Fig. 1, I cannot find panel b, I think it has been mislabelled as c. In panel c (there are two of them), the color bar should be labeled in the figure. Also, this figure contains text that is much too small, e.g. axis labels in c-e. In panel i, why would pseudo-time inference be a "downstream analysis"? I thought pseudotime was estimated jointly with the other model parameters? Panel j, why does this represent gene dynamics? Hard to see the link here. Panel k, I thought TF-target gene relationships were the input to the model, how can these be an output as well? This needs to be explained in more detail, what is the information the model gives here? Panel o, text overlaps with panel elements. Also, unclear whether the performance shown here just relates to the example shown in panel l, or to the entire simulated data, which should contain 100 synthetic gene dynamics (Results, Line 190).

Results, Line 203, "To evaluate TFVelo on scRNA-seq data, we first apply it to the dataset of E15.5 mouse pancreas". It would be nice to describe "the" pancreas dataset and to cite it here, where it is first considered.

Fig. 2d: This plot is missing a colorbar, x- and y-axis labels. Also, what's the relevance of revealing a gene expression cascade in this context? What biological process does it correspond to here? There are different endocrine populations formed in this process, this plot seems to smooth over all of them.

Results, Line 215, the fact that a subset of ductal cells is cycling has already been observed in the original publication (Bastidas-Ponce et al. 2019)

Fig. 2g-k, does TFVelo always fit the full circle, even though only part of these dynamics might actually be present in the data? See e.g. panel g.

Fig. 2c, I don't understand the "non-smoothness" the authors are referring to with red-circles for scVelo. From looking at the 2D stream plot, everything looks equally smooth, I think. Again, these statements have to be either removed or backed by quantitative analysis in a higher-dimensional space.

Results, Lines 259-268. It is unclear to me how TFVelo contributes to a better understanding of TF-target interactions. The authors should clarify how exactly they encode prior knowledge about putative regulators for each target gene, so that I'm able to judge to what extent the model has learned something new. Also, this claim should be quantified, beyond a single gene (JUND).

Results, Lines 297-308. To establish TFVelo as a framework to detect regulatory interactions would require the authors to present a systematic evaluation over a large set of previously reported gene interactions, and some metrics to score the quality of their predictions. Also, they should compare their tool with other GRN learning approaches.

The analysis done in figures 2-4 is essentially the same, maybe the authors could consider highlighting different parts of their method on different applications, where each analysis is most relevant, given the biology at hand?

Results, Lines 377-379. The authors filter out M-stage cells as they argue cycling dynamics are not supported by TFVelo. If I recall correctly, Bergen et al. showed in their original scVelo publication (Bergen et al. 2020) that they could resolve cycling ductal cells. Could the authors comment on why this is possible with scVelo, but not with TFVelo?

Discussion, Line 409: "However, the existing methods often fails to fit the cell dynamics well in the unspliced/spliced space, which is because of the weak signal and high noise at single cell resolution." The claim about "weak signal and high noise" is central to this paper and should be demonstrated and quantified much more.

Implementation

Most of the code is copied from scVelo's GitHub repository, without appropriate acknowledgments of the source. This is bad practice from two perspectives:

Original source code should be acknowledged properly, in a prominent place.

Copied code creates an unnecessary overhead, and increases the maintenance load. The authors should avoid copying any code, instead, add scVelo as a dependency, and import from scVelo where necessary.

Further, this package has neither proper documentation, testing, nor tutorials. I don't find this appropriate for publication in a high-impact journal like Nature Communications.

References

- Bastidas-Ponce, Aimée, Sophie Tritschler, Leander Dony, Katharina Scheibner, Marta Tarquis-Medina, Ciro Salinno, Silvia Schirge, et al. 2019. "Comprehensive Single Cell mRNA Profiling Reveals a Detailed Roadmap for Pancreatic Endocrinogenesis." *Development* 146 (12). <https://doi.org/10.1242/dev.173849>.
- Bergen, Volker, Marius Lange, Stefan Peidli, F. Alexander Wolf, and Fabian J. Theis. 2020. "Generalizing RNA Velocity to Transient Cell States through Dynamical Modeling." *Nature Biotechnology* 38 (12): 1408–14.
- Bergen, Volker, Ruslan A. Soldatov, Peter V. Kharchenko, and Fabian J. Theis. 2021. "RNA Velocity—current Challenges and Future Perspectives." *Molecular Systems Biology* 17 (8): e10282.
- Gorin, Gennady, Meichen Fang, Tara Chari, and Lior Pachter. 2022. "RNA Velocity Unraveled." *bioRxiv*. <https://doi.org/10.1101/2022.02.12.480214>.
- Gorin, Gennady, Valentine Svensson, and Lior Pachter. 2020. "Protein Velocity and Acceleration from Single-Cell Multiomics Experiments." *Genome Biology* 21 (1): 39.
- Haghverdi, Laleh, Maren Büttner, F. Alexander Wolf, Florian Buettner, and Fabian J. Theis. 2016. "Diffusion Pseudotime Robustly Reconstructs Lineage Branching." *Nature Methods* 13 (10): 845–48.
- La Manno, Gioele, Ruslan Soldatov, Amit Zeisel, Emelie Braun, Hannah Hochgerner, Viktor Petukhov, Katja Lidschreiber, et al. 2018. "RNA Velocity of Single Cells." *Nature* 560 (7719): 494–98.

Lange, Marius, Volker Bergen, Michal Klein, Manu Setty, Bernhard Reuter, Mostafa Bakhti, Heiko Lickert, et al. 2022. "CellRank for Directed Single-Cell Fate Mapping." *Nature Methods* 19 (2): 159–70.

Li, Chen, Maria C. Virgilio, Kathleen L. Collins, and Joshua D. Welch. 2022. "Multi-Omic Single-Cell Velocity Models Epigenome–transcriptome Interactions and Improves Cell Fate Prediction." *Nature Biotechnology*, October, 1–12.

Marot-Lassauzaie, Valérie, Brigitte Joanne Bouman, Fearghal Declan Donaghy, and Laleh Haghverdi. 2022. "Towards Reliable Quantification of Cell State Velocities." *bioRxiv*.
<https://doi.org/10.1101/2022.03.17.484754>.

Pijuan-Sala, Blanca, Jonathan A. Griffiths, Carolina Guibentif, Tom W. Hiscock, Wajid Jawaid, Fernando J. Calero-Nieto, Carla Mulas, et al. 2019. "A Single-Cell Molecular Map of Mouse Gastrulation and Early Organogenesis." *Nature* 566 (7745): 490–95.

Qiu, Xiaojie, Yan Zhang, Jorge D. Martin-Rufino, Chen Weng, Shayan Hosseinzadeh, Dian Yang, Angela N. Pogson, et al. 2022. "Mapping Transcriptomic Vector Fields of Single Cells." *Cell* 185 (4): 690–711.e45.

Soneson, Charlotte, Avi Srivastava, Rob Patro, and Michael B. Stadler. 2021. "Preprocessing Choices Affect RNA Velocity Results for Droplet scRNA-Seq Data." *PLoS Computational Biology* 17 (1): e1008585.

Reviewer #3 (Remarks to the Author):

The authors developed a new method TFvelo for calculating expression velocity of gene expression in single cell expression profiles using regression-based modeling of TF-target expression relationship. TFvelo looks useful to capture dynamics and key regulatory factors regulating dynamics. However, the current manuscript seems to lack significance as follows:

1. TFvelo depends on the regression of TF and target gene expression. The list of TFs must be an important factor for the TFvelo's performance. Please provide the database or criterion of TF list.

2. Figure 1o shows the performance of TFvelo in capturing dynamics of a synthetic data. Please provide the comparison with other GRN constructors such as SCENIC, SCODE, Scribe, GRNVBEM, CellOracle, LEAP, TENET, SINGE.

3. The comparison of TFvelo and scVelo in Figure 2 does not look like fair comparison. scVelo also provide critical gene set for the dynamics. The counter-examples also should be provided. Specifically, Please provide the phase portrait of the gene set provided by scVelo.

4. In overall, TFvelo seems a very useful bioinformatics tool. However, the manuscript does not provide new findings. Please provide what is new findings such as key regulator factors. And please validate the findings with at least one example.

Reviewer #4 (Remarks to the Author):

This manuscript by Li and colleagues describes an extension of RNA velocity, a method originally developed by La Manno et al. 2018 to estimate the evolution of gene expression over time from single-cell RNA-sequencing data. The proposed extension harnesses the relationship between expression of transcription factors and their target genes to predict target gene-specific RNA velocities, allowing velocity estimation in circumstances where unspliced (U) and spliced (S) content cannot be accurately obtained or measured.

Previous methods for estimating RNA velocity of single cells use U/S counts to fit gene-specific phase portraits and infer dynamic cell transitions based on this ratio. However, for some datasets, unspliced counts contain too high sparsity, are not made available when published (i.e. data privacy issues for raw human sequencing data), or cannot be easily measured with the data acquisition technique (i.e. spatial transcriptomics methods). The extension described in this

manuscript implements a linear model to fit gene expression for a weighted combination of transcription factors and a single target gene to a phase portrait, instead of using unspliced and spliced counts. The model is solved with the expectation-maximization (EM) algorithm to iteratively learn the transcription factor weights, the phase portrait shape parameters, and cellular latent time. TFvelo's implementation is built upon the previous scvelo framework from Bergen et al. 2020, which also learns a similar latent time and phase portrait parameters by EM.

The manuscript is organized as follows:

The authors begin by outlining the main claim of their work: given a target gene (Target), cell-specific velocity estimates inferred with scvelo (for that same target gene) can instead be inferred with a linear equation comprised of a weighted combination of the RNA expression levels of transcription factors (TF). These TFs are known to either positively or negatively regulate expression of such Target, according to database references, and can therefore have a positive or negative weight. First, the authors show that the weights for each transcription factor, when fit using LASSO regression, yield a velocity estimate for the Target gene that correlates well with scvelo estimates. Next, the authors use an EM algorithm to estimate velocities from paired TF-Target expression data and corresponding phase portraits (Figure 1). The model is evaluated on synthetic data and a small number of published datasets. The authors claim their method estimates velocities that more closely reflect known biology, such as during exit of pancreatic ductal cells from the cell cycle (Figure 2) and erythroid differentiation (Figure 3). The authors further apply TFvelo to a scRNA-seq dataset missing U/S counts (Figure 4) and to MERFISH spatial transcriptomics data, for which no U/S information is collected due to experimental limitations of probe design (Figure 5).

I enjoyed reading this work, and I think it contains interesting computational tricks of value worthy of publishing among the single cell and RNA velocity community. However, I have concerns for which I expect major revisions in order to accept it for publication in Nature Communications. (1) the method requires significantly more evidence to be considered a convincing and sufficiently characterized improvement over existing velocity methods; (2) while the conceptual underpinnings of TFvelo are clever, I am not fully convinced this manuscript introduced a substantial new advancement to the RNA velocity framework; (3) I have concerns about the ability to claim biologically-meaningful interpretations with a velocity estimation derived from transcription factors and their target genes, which behave according to more complex metabolic processes than those for traditional RNA velocity methods relying on linear progression from unspliced to spliced; (4) the code cannot be run and therefore assessing reproducibility of the method (an essential objective) is not possible at this time.

MAJOR COMMENTS

Point 1 - I am concerned about the underlying assumptions of RNA velocity that are transferred to this model but may not apply, or even be violated, when not using unspliced and spliced RNA information. Unspliced RNA molecules are directly converted into spliced RNA molecules, but transcription factors act in a more complex – and even indirect manner – on their target genes. Therefore, modeling these behaviors with a velocity is significantly more challenging and requires a more elaborate formulation than the one presented by the authors.

Point 2 - The methods section is too light on detail and does not fully explain the underlying assumptions made by the TFvelo model. The authors appear NOT to use a series of differential equations for inferring a time derivative of the velocity (as is the case for all prior velocity methods). Such an equation set will be challenging to define for complex TF regulation, but without them, the velocity vector becomes less physically interpretable, as transcription, splicing and degradation rate equivalents are not learned. The authors claim the weights of the TFs in TF-Target pairs are directly interpretable as the role of the TF in the velocity, which I agree with, but they provide insufficient examples. One example TF (JUND; Supplemental Figure 2) and distributions of the weights for all TF-Target pairs (Figures S2b, 3j) come across as cherry-picked. A more extensive analysis ought to be performed here: is there any enrichment for particular gene classes in the positive and negative weights? Do any TFs have different weights for different target genes? A single TF usually acts on multiple targets in the same pathway or trajectory, and I would

anticipate they have similar weights for each target gene in that path.

Point 3 - Likewise, I am doubtful whether the velocity estimate obtained using TF-Target combinations should even be identical to those obtained using unspliced-spliced information. The delay between a TF and its Target implies something about a more upstream rate of change than traditional RNA velocity. I am also not convinced it makes sense to average individual TF-Target velocities across all genes for a single cell, when TF-Target pairs influence different axes of variation that are simultaneously present in a sample. It is essential the authors more carefully reflect on these questions in the manuscript and more robustly evaluate them with TFvelo to ensure that the choices made by their model are physically-consistent with the underlying biological processes for which velocity is being estimated. For example, a recent method called MultiVelo (Li et al 2022) estimates a velocity from snATAC and snRNA-seq data. Comparing the velocity estimates obtained by MultiVelo to those from TFvelo, rather than only to the velocities obtained by scvelo, would help provide a significantly more suitable benchmark.

Point 4 - On a related note to Point 3 above, no comparisons were made by the authors to other modalities used to study gene regulatory behavior, such as single cell ATAC or CUT&Tag approaches. These methods would be a better proxy for inferring regulatory behavior between TFs and targets, and it is even possible to measure them jointly with scRNA-seq (permitting direct comparison to a traditional RNA velocity estimate).

Point 5 - It is unknown whether RNA expression levels of TFs correspond to protein levels. This is of course true for all genes measured by scRNA-seq, but it is especially critical when focusing on a model involving TFs for velocity specifically because the relationship between TFs and their targets is not unambiguous in the way it is between unspliced and spliced RNA. In the context of this manuscript, TFvelo essentially uses TF gene expression information as a proxy for TF protein binding to the DNA sequences and regulatory elements of target genes, all in order to predict a velocity that seems comparable to that estimated with U/S counts. TF activation/repression of target genes will depend on many cellular factors, including the rate of nuclear import of the TF proteins from the cytoplasm after translation, the general chromatin accessibility, and the influence of regulatory DNA elements. These challenges are one reason why measuring an accurate RNA velocity with single nuclei data is not straightforward: single nuclei data does not contain all spliced RNA content and therefore models RNA nuclear export (occurs on a too-fast time scale) instead of degradation (occurs on a slower timescale measurable in a static snapshot). Transcription factors may also act on different time scales not suitable to the time scale inferable from a static single cell snapshot. These caveats were not raised in the manuscript and without addressing Points 1-4 above, I am unsure whether we can fully trust the use of TF gene expression for velocity estimation of relevant target genes.

Point 6 - I applaud the authors for focusing in their manuscript on the fitting of phase portraits, as this is an essential component of traditional velocity estimation methods and has been regularly neglected by some recent velocity papers. However, it is not clear to me why it is necessary to fit phase portraits on transcription factors and their target genes using expectation-maximization (EM) in the first place, when the authors first show that their linear model can be sufficiently fit using a LASSO regression. What additional benefits are provided by using EM in this context? Why not just use a simple LASSO regression? It would be a good start to have more rigorous direct comparisons between TFvelo estimates obtained using the LASSO and EM.

Point 7 - The few case-by-case examples from the literature provided by the authors of TF-Target pairs with strong influences on the overall cellular velocity estimate do not fully convince me of the interpretability of the model. For example, I would expect to see much better characterized TF-target pairs used as validation that their model performs well, rather than obscure genes such as PPP1R14A, SERTAD1, RGS2, and GSTP1 (Figure 4). The authors weakly support their observations for such genes using literature of findings in zebrafish, yet to my understanding, the scRNA-seq data utilized is only from humans (also Figure 4).

Point 8 - Velocity confidence is used (Figures 2e, 3f) to show that TFvelo infers a "better" velocity estimate than scvelo. This metric was made available as part of the scvelo package, but it is not part of the original scvelo paper (Bergen et al 2020) and therefore not peer-reviewed. A better

benchmark to convince me of superior velocity estimates by TFvelo is to evaluate the quality of the phase portrait fits themselves. For the TFvelo phase portraits shown in the main/supplemental figures, side-by-side comparisons to the scvelo phase portraits are presented in a confusing manner (on far opposite sides of the panels) or are completely absent. Side-by-side comparisons between TFvelo and scvelo (and perhaps MultiVelo as suggested by Point 4) as well as a more robust metric for evaluating phase portrait fits, would better support the claim that TFvelo yields improved velocity estimates.

Point 9 - The EM implementation provided by the authors seems to not be significantly different from that previously published as scvelo by Bergen et al 2020. The model framework is, as far as I can see, a cookie-cutter copy of scvelo, with exception of a handful of lines implementing the author's linear model and adjusting the EM to enable the inference of three sets (instead of two sets) of parameters. Providing a more clear explanation of the method developed (see Point 2 and Point 6 above) would be appreciated to clarify (1) why the EM and phase portrait fits are necessary rather than a simple LASSO regression, and (2) how the formulation differs from scvelo.

Point 10 - After several troubleshooting attempts, I was unable to run the TFvelo code provided on the corresponding GitHub page, and the provided lab website page does not exist. The necessary database files containing information of TF-Target gene pairs are missing, and even when I find what I believe to be the correct files on the web, the code does not run (perhaps because the necessary database files I obtained from the web were not the same as those used by the authors). The ReadMe provided by the authors is also short and unclear. I think this is rather a problem with the usability of the package, rather than a fundamental bug in the code; with better documentation, it would likely be possible to run the model. However, I am currently unable to do so and cannot verify reproducibility of the results shown in the paper.

MINOR COMMENTS

Point 11 - I greatly appreciate the efforts by the authors to illustrate the performance of their model on simulated data, as I think there is an urgent need to evaluate RNA velocity models on simulated data where a suitable ground truth velocity is known. However, the performance evaluation of the simulated results is limited and relegated to the supplemental file. I think this is something that should be highlighted in the main text of the manuscript.

Point 12 - Throughout the manuscript, stream plots are used to show the velocity estimations on various datasets (including, but not limited to, Figures 2B, 2C, 3B, 3E, 4B, and 5B). Although this visualization method was introduced in scvelo, it can overly smoothen the velocity arrows and provide misleading interpretations. I suggest that the authors instead illustrate their velocity estimations using the traditional velocity quiver plots.

Point 13 - Velocity is typically present in a dataset as a delay in expression between two entities, in this case the TFs and the target. It would be useful to see visualization of the expression of both these entities (TF and target) along the learnt latent time, in order to assess whether that expected shift is indeed present after performing EM (and not just in the phase portraits themselves).

Point 14 - Although in some cases we might not be able to compute velocity information using unspliced/spliced, in two scRNA-seq datasets used by the authors (pancreas and erythroid gastrulation) we indeed have the possibility to estimate the velocity with scvelo. I had hoped for a direct comparison between the velocity vectors estimated using the two different models, not just the velocity confidence. How well do they correlate? Do they correlate differently for different cell types? This is related to Point 7 above.

Point 15 - The application of TFvelo to spatial transcriptomics data is especially exciting, as it is a modality in which velocity is difficult to infer since probes are rarely intron-specific. I also appreciate the authors correctly visualizing the velocity on a single cell UMAP space as opposed to arrows directly on the image, which has been done in several previous studies yet is completely incorrect nonsense (since cells don't physically move in space according to the velocities). However, I do not understand why the authors remove M phase cells before performing analysis.

They claim it is to avoid a circular trajectory, but even with the M phase cells, one could represent the velocity as a single linear trajectory along the cell cycle.

Point 16 - In Figure 2, the authors claim that upon velocity estimation with TFvelo, the progression of cells out of the cell cycle into ductal cells is "corrected" and better following a trajectory expected by the prior biological knowledge. However, it is concerning to me to observe the loss of a circular velocity along the cycling cells themselves, as cells are certainly progressing through the cell cycle phases prior to differentiation. I am unsure whether TF is indeed "correcting" an "incorrect" velocity in this context, or whether it simply fixes one velocity direction (from progenitors to ductal cells) while breaking another (progression in the cell cycle).

Point 17 - The majority of figures and figure legends in the manuscript are vague and difficult to follow. In fact, most supplemental figures do not have any figure captions. Figure legends are sometimes missing or illegible, making it very difficult to understand the color/label schemes of the corresponding panels.

We thank all reviewers for the constructive comments. We have addressed all their comments point by point, and improved our paper accordingly, including better explanation for our motivation, comparison with more existing methods with more quantitative evaluations on more datasets, and more comprehensive biological insight analysis for our results.

Reviewer #1:

1. The authors claimed that "the RNA velocity can be represented as a linear combination of the expression levels of transcription factors (TFs)"; RNA velocity is the information using both transcriptional and posttranscriptional regulation. (1) The linear combination of TFs only can represent the transcriptional aspect, but not the posttranscriptional regulation. (2) In this regard, Eq1 cannot hold. (3) It is also not clear how the quantity of velocity is obtained by running scVelo.

Reply: Thanks for your valuable comment.

(1) We acknowledge the difference between our velocity, inferred from TF in combination with the target, and the velocity determined from unspliced/spliced counts. Specifically, the latter is inferred by the relative unspliced/spliced abundance changing trend, and the former refers to the total mRNA abundance changing trend, of which the biological process has been described well by transcriptional regulation and RNA degradation in several peer reviewed papers [PMID: 21562557, 36762475, 35300460]. Since both velocity concepts focus on the changing trend of mRNA, the two could be highly correlated. As a result, the motivation for **Eq1** is that the value of RNA velocity modeled by scVelo could be approximated by a linear combination of abundance of TFs and the degradation rate of the target gene itself. As shown in revised **Fig. S1**, such hypothesis is well confirmed, indicating that TF information is sufficient for the inference of our new velocity. Different to the idea that conventional RNA velocity is based on the phase delay between unspliced and spliced RNA, TFvelo relies on the phase delay between TFs and target, which can provide comparable or even better temporal performance on predicting gene and single cell dynamics, as shown in our paper, further validating the velocity concept inferred by TF and target.

In addition, **Fig. R1** provides one Pancreas example that TFvelo can correctly find the dynamic of individual gene in the phase portrait, while conventional methods (scVelo, UniTVelo, Dynamo and cellDancer) cannot due to the severe noise and sparsity of unspliced/spliced abundance in single cell dataset. Furthermore, **Fig. R2** and revised **Fig. 4c&e** in main text illustrate that high sparsity in unspliced mRNA counts is a large challenge for phase portrait fitting for those un/spliced-based approaches. The results by UniTVelo and cellDancer cannot accurately capture the differentiation process, and ScVelo and Dynamo fail to generate dynamic models for these genes.

(2) We have updated our text for **Eq1** to prevent any misinterpretation. It now reads, "As a result, the RNA velocity of gene g can be estimated as equation 1", instead of "the RNA velocity can be written as equation 1".

(3) As for "how the quantity of velocity is obtained by running scVelo", we have run the scVelo pipeline with default settings to obtain the velocity and now have introduced this information in our supplementary information.

Figure R1. The comparison between different methods on modeling the dynamics of an example gene, LITAF. (a) The distribution of cell types on UMAP, in which the arrows represent the velocity by TFvelo. Panel (c) and (d) use the same colors in (a) to label these cell types. (b) The expression distribution of LITAF. (c) Phase portrait fitting of TFvelo. (d) Phase portrait fitting of methods based on splicing data, including scVelo, UniTVelo, Dynamo and cellDancer.

Figure R2. The challenges of sparsity for scRNA-seq data. (a) The number of genes whose sparsity is lower than 0.9, based on the unspliced, spliced and total mRNA abundance, respectively. The sparsity of a gene is defined as: $\text{Sparsity} = \frac{\text{The number of cells that the count of this gene is zero}}{\text{The total number of cells}}$. (b) Comparison on phase portrait fitting for SURF4 and CD24A on Pancreas dataset, where the unspliced count is very sparse. The last column is the distribution of gene expression on UMAP. The cells are colored in the same way as that in Fig. R1a. Results of scVelo and Dynamo are not shown because they can not construct dynamic models on these genes.

2. Obviously, there is a phase delay between TF and the target genes. The information TFvelo uses is not RNA velocity but the dynamic relationships between TFs and their targets. It is the concept that TENET or SCRIBE uses to obtain causal relationships using transfer entropy. Therefore, it does make sense to evaluate its performance with TENET and SCRIBE rather than scVelo.

Reply: We agree that reconstructing the gene dynamics and regulation causality is highly related. However, we now clarify that TFvelo has different goal and input/output to methods like TENET or SCRIBE. Using TF-target information, the goal of TFvelo is to infer RNA dynamics, pseudo-time, and cell trajectories, which is same to all RNA velocity models, rather than to infer casual gene relationships. And TFvelo has already selected TFs and their potential targets as input based on known TF databases. For better clarification, we now have introduced those gene regulation inference methods in **Introduction** and discussed the differences between TFvelo and them.

Although TFvelo is not a tool for TF-target prediction, it can provide biological insights from the perspective of gene regulation. Taking the results on pancreas dataset for example, we further do KEGG /GO term enrichment analysis for the best fitted genes, and then explore the important TFs involved in the biological process. Specifically, using the pancreas dataset as an example, we observed a strong enrichment of the best-fitted genes in KEGG pathways associated with insulin secretion and the glucagon signaling pathway (Fig. R3a and Fig. 3a in main text). Next, we conducted an analysis of the weights assigned to each TF, which suggests that REST and HMGN3 consistently exhibit high absolute weights (Fig. R3b and Fig. 3b in main text).

Figure R3. Analysis of results on Pancreas dataset. (a) The KEGG pathway enrichment from the genes that are best fitted by TFvelo on pancreas dataset. (b) The TFs that always have higher weights on multiple target genes. The counts represent the number of target genes for which the TF have an absolute weight larger than 0.5.

Reviewer #2:

Review of 'TFvelo: gene regulation inspired RNA velocity estimation' (NCOMMS-23-16358)

Summary

The authors introduce TFVelo, an approach to estimate gene-wise velocities without using spliced/unspliced data. They argue that their approach includes the effects of gene regulation, is applicable to imaging data, where no spliced/unspliced count information is available, and accurately fits 'clock-wise' phase portraits of TF-target gene relationships.

The key idea is to represent RNA velocity as a linear combination of TF expression levels. Rather than considering unspliced/spliced counts in phase space, TFVelo considers the expression level of a TF and its target(s). This is translated into one simple linear ODE per gene, that relates the velocity of that gene to the expression level of upstream TFs. The expression of the target gene itself is modeled as a sine function, and parameters are learned from data. I have a number of concerns related to this work.

First, the authors motivate their approach by emphasizing that they don't use unspliced/spliced count data. While I agree that this data source can be quite noisy in some situations, I don't think it is generally an improvement to have a method that does not make use of this additional information. Not using this information renders TFVelo a standard trajectory inference (TI) approach, applicable to the same data as all of the previous TI approaches. Accordingly, the authors should benchmark their method with approaches like PAGA, Monocle, Slingshot and Palantir, and demonstrate their advantages.

Reply:

1. We agree that the un/spliced counts are additional information, however, as shown in our paper, in many cases the sparsity and noise of un/splicing counts could result in failures on phase portrait fitting and dynamics inference. Now we have added quantitative analysis for the high noise and sparsity un/splices data in **Fig. 4** of main text, **Fig. S6** in Supplementary Information, and also our reply to your question 5. In contrast, by replacing the splicing information with the regulation information, TFvelo can model dynamics of genes that can not be processed by existing RNA velocity methods.

2. Following the reviewer's suggestion, now we have added the comparison results of pseudotime inference methods in **Fig. S5** and **Fig. S9** in Supplementary Information. As expected, the pseudotime results of these TI methods are highly consistent with that of TFvelo. Actually, TFvelo has several advantages compared with TI approaches. Firstly, TFvelo does not require the ground truth label of the initial cell state. Secondly, TFvelo can analyze the detailed dynamics of individual genes in phase portrait. Thirdly, TFvelo can also detect important genes and TFs. Considering that the core idea is to infer velocity-based pseudotime from the phase portrait fitting, we take more comprehensive comparison with those RNA velocity models in the main text.

Second, the authors emphasize throughout the text that previous velocity approaches don't fit the data well. However, while many previous approaches are available, including scVelo, Velocity, LatentVelo, VeloVI, pyroVelocity, UniTVelo, etc., they only compare their approach with scVelo, and they don't actually show that scVelo does not fit the data well. If the authors include such a broad claim, they should demonstrate it explicitly, benchmark their approach with a number of popular velocity approaches, and explicitly show that these earlier approaches don't fit unspliced/spliced data well in phase space, in a quantitative manner.

Reply: Thanks for your valuable comment.

Now we select scVelo, UniTVelo [PMID: 36329018], Dynamo [PMID: 22245546] and cellDancer [PMID: 37012448] as the baseline approaches, all of which are published in the high impact journals recently. scVelo and Dynamo have been widely used in single cell analysis. UniTVelo directly designs a profile function of time for modeling the gene expression level. cellDancer is a recent publication that allows a cell-specific transcription rate. In addition, we also compared TFvelo with Multivelo [PMID: 36229609] on one multi-omics dataset, which shows that TFvelo can get the better results with Multivelo, even without using the additional ATAC-seq data.

To qualitatively and quantitatively analyze the phase portrait fitting, we propose three metrics, which are the fitting error on phase portrait, the intra-class distance on phase portrait and the intra-class distance on phase portrait. With these quantitative analyses, TFvelo shows significantly improvement in the data fitting (**Fig. 2d-i**).

Third, TFvelo estimates a velocity estimate and a latent time assignment per gene and cell, but the authors don't make use of this information directly in their data applications or benchmarks. Instead, they use scVelo to direct Markov-chain edges using their velocities and estimate scVelo's velocity pseudotime; on this Markov chain. Next, they take derivatives of this pseudotime in UMAP space, apply heavy smoothing and post-processing, and visualize the resulting dynamics via streamlines in 2D UMAP. Given this heavy smoothing, their pseudotime and velocities are not comparable to scVelo's outputs, and I'm highly skeptical of the metrics they employ for comparisons, like the velocity confidence. Velocity confidence measures the extend to which neighboring velocities agree, given their extremely heavy smoothing, it comes as no surprise that they achieve high values in this metric - that is, however, in no way a validation of their approach.

Reply: In term of heavy smoothing, we apologize for the misleading. It is noticed that even in the old version, the smoothing strategy was applied only in the step of drawing stream plot. The velocities within neighbor cells used for measuring velocity confidence, or the pseudotime was actually not smoothed. As a result, the velocity confidence is still a fair way to compare the consistency of velocity.

Following your suggestions, we have updated the post-processing strategy and the **Methods section** in our paper. Specifically, we adopted the same strategy in previous methods like scVelo. With the velocity of each gene, we first calculate the velocity-inferred transition matrix, which is

defined as the cosine similarities between velocities and changing of expressions within neighboring cells. Then a distribution over root cells can be obtained from the transition matrix. Finally, we calculate the pseudotime, which measures the average number of steps it takes to arrive at a cell after start walking from one of the root cells.

After calculating the pseudotime, we select the genes with fitting time consistent with the cell pseudotime as the velocity genes. To visualize the result, we use the same strategy of obtaining stream plot in scvelo. Consequently, now there is no steps of taking derivative of pseudotime or smoothing the arrows.

Finally, in their implementation on GitHub, they mostly copy scVelo code, without appropriate, prominent acknowledgments of the source of this code. Even just from an implementation point of view, this is bad practice: it creates a massive overload and makes it essentially impossible to maintain their code base. I would strongly advise against copying large parts of source code, it makes much more sense to add scVelo as a dependency and to import the required modules, e.g., for plotting, where needed. Also, for a publication in Nature Communications, I expect to see a minimal amount of documentation, testing, and tutorials, which is absent here.

Reply: We apologize for the insufficient documentation on the GitHub page.

Due to the wide usage of scVelo and its clean, well-organized codes, we develop TFvelo based on scVelo. Following your suggestion, we have further developed the package of TFvelo, deleted those unnecessary parts copied from scvelo, and provided more comprehensive documentations and tutorials for TFvelo. We also added more appropriate, prominent acknowledgments for scvelo in our GitHub page.

In TFvelo, we have removed all functions which are exactly the same in scVelo. However, for some functions with identical names, we are not able to directly import them from scvelo for implementing TFvelo, and have to modify the codes due to the detailed differences between the functions in TFvelo and scvelo. For instance, in `scvelo.pp.moments()`, scVelo tries to obtain moments for "spliced" and "unspliced" expression, whereas in `TFvelo.pp.moments()`, we require moments for the "total" expression. And it is hard to pass a parameter into `scvelo.pp.moments()` to realize the function of `TFvelo.pp.moments()`.

Please see below for a detailed account of my comments.

Major comments

1. From a general point of view, TFVelo does not make use of the additional information provided by unspliced/spliced counts and instead fits TF-target gene relationships to uncover trajectories. As such, it's similar to other methods that infer trajectories based on scRNA-seq alone, such as Palantir, Monocle, PAGA, and Slingshot. It would be nice if the authors could outline the main advantages of TFVelo over these competing approaches, which work on the same input data. Of course, TFVelo can give additional information about gene regulation (even though I

don't fully understand how this improves over model inputs), but the inference of cellular dynamics is still a claim the authors make, and this should be compared with competing approaches. Most pseudotime-based approaches yield information on gene dynamics. Additionally, in many of the systems the authors present here, including the pancreas or blood generation during mouse embryogenesis, the starting cell/state is known and can be provided to estimate pseudotime.

Reply: Following your suggestions, now we have added the results of several representative pseudotime inference methods on pancreas and gastrulation erythroid (blood generation) datasets into the **Fig. S5** and **Fig. S9**. As expected, the pseudotime results of these TI methods are highly consistent with that of TFvelo. It is noticed that the TFvelo has several advantages compared with TI approaches. Firstly, TFvelo does not require the ground truth label of the initial cell state. Secondly, TFvelo offers an additional capability to analyze the detailed dynamics of individual genes in phase portrait and phase delay between TF and target genes (**Fig. 2c** and **Fig.3**). Thirdly, TFvelo can also detect important genes whose velocity that aligns with the inferred pseudotime, and TFs with the highest weights. As a result, the TFvelo results can be very biologically interpretable.

Considering that these tasks are the main tasks for existing RNA velocity methods, we take more comprehensive comparison with those RNA velocity models.

2. Results, Line 113: "In addition, the TFs with non-zero positive weights are significantly enriched in TFs set functionally linked to the target gene". That appears to be a circular argument as the authors started out with a set of candidate TFs for each gene already. Is this a sanity check?

Reply: We apologize for the confusion. In this scenario, we feed all TFs into LASSO model without any filtering or selection, and try to explore whether those TFs with non-zero weights tend to be enriched in the known TF set of the target gene, which is extracted from the ENCODE database. This positive enrichment analysis result indicates that the velocity of a target gene can be inferred based on the expression of its TFs.

3. The authors argue that models like scVelo and velocity don't fit the observed co-expression of spliced/unspliced counts well, see e.g. in the intro. However, to motivate their model, they show that LASSO regression based on upstream TF expression (first section of the Results) can recover scVelo velocities - how much of a selling point is that, given that the authors argue against these very velocities?

Reply: Previous velocity methods, such as scVelo, have demonstrated their capability on modeling gene dynamics, and further inferring the cell trajectory and pseudotime in several cases. The good performance of LASSO indicates that the dynamics derived from splicing information can also be extracted from transcriptional information (As discussed in the **Findings section**). However, due to sparsity and noise of single cell data, it is hard for existing RNA velocity approaches to extract and fit the gene dynamics well, especially in phase portrait. The quantitative analysis of the sparsity problem is discussed in details in our response to your question 5. In contrast, modeling velocity based on TF-target, TFvelo is a better way to address that issue and can achieve better performance on multiple single cell datasets.

4. Results, Line 127: “Fig. 1a provides an example on modeling the dynamics of gene LITAF, to explain why TFvelo outperforms existing methods (scVelo is used as the representative).” If the authors show a comparison with one method, scVelo, they cannot argue that their method outperforms all other existing methods. They don’t show that here.

Reply: We have now added more methods, including UniTVelo, Dynamo and cellDancer for comparison, all of which have been published recently and widely used. The same example gene, LITAF, from the pancreas dataset is selected to illustrate that these baseline methods can not fit the dynamic in phase portrait well, which motivates us to explore constructing gene dynamics based on regulation. In addition, we also compared TFvelo with Multivelo, which makes use of ATAC-seq and scRNA-seq. The result in Fig. 5 shows that even without the additional ATAC-seq, TFvelo is still better than Multivelo.

Specifically, as shown in Fig. R4, TFvelo correctly fits LITAF’s dynamics starting from ductal cells and ending with mature endocrine cells like Alpha or beta. By comparison, these baseline methods can not fit the process well. In detail, scVelo and Dynamo model the dynamics start from alpha cells (blue). UniTVelo models the dynamics start from some Ngn3 high EP cells (yellow), which is because the Ductal, Ngn3 low EP and Ngn3 high EP cells are mixed on the un/spliced phase portrait.

Figure R4. The comparison between different methods on modeling the dynamics of gene LITAF. (a) The distribution of cell types. The arrows reflect the velocity by TFvelo. (b) The expression distribution of LITAF on UMAP. (c) Phase portrait fitting of TFvelo. (d) Phase portrait fitting of methods based on splicing data, including scVelo, UniTVelo, Dynamo and cellDancer.

5. Results, Line 130: “(...) while transcriptional dynamics of RNA splicing may not provide sufficient signal”; This is a really central claim of this paper that they use as

motivation to develop their new approach. However, I'm currently missing any quantitative evaluation of this claim, they just cite (Bergen et al. 2021).

Following your suggestion, now we have quantitatively analyzed the sparsity of un/spliced data to better support our claims and motivation. Considering the high dropout rate and noise in scRNA-seq data, dividing total mRNA into the unspliced and spliced could result in much higher sparsity, as shown in **Fig. R5a**, consist with the findings shown in velocity [PMID: 30089906]. As a result, the methods based on splicing may fail to capture dynamics due to extreme sparsity. In contrast, TFvelo models the gene dynamics based on total expressions of each gene, thus can extract more information from multiple TFs and target.

Two sparse genes are shown in **Fig. R5b**, in which the sparsity of unspliced count for SURF4 is 99.7%, and that for CD24A is 99.8%. While both of them can pass the preprocessing of scVelo, scVelo cannot build dynamics for them, and UniTVelo or cellDancer cannot construct the correct dynamics, neither. Dynamo just filters them out before the step of dynamics construction.

Figure R5. The challenges of sparsity for scRNA-seq data. (a) The number of genes whose sparsity is lower than 0.9, based on the unspliced, spliced and total mRNA abundance, respectively. The sparsity of a gene is defined as: $\text{Sparsity} = \frac{\text{The number of cells that the count of this gene is zero}}{\text{The total number of cells}}$. (b) Comparison on phase portrait fitting for SURF4 and CD24A on Pancreas dataset, where the unspliced count is very sparse. The last column is the distribution of gene expression on UMAP. The cells are colored in the same way as that in **Fig. R4a**. Results of scVelo and Dynamo are not shown because they can not construct dynamic models on these genes.

We also introduce quantitative metrics to evaluate the phase portrait fitting to demonstrate the advantage of TFvelo, including:

- (1) The intra-class distance on phase portrait, which measures the normalized distance between cells within the same type on the phase portrait. This reflects whether cells within the same type gather on the phase portrait. The lower, the better.
- (2) The inter-class distance on phase portrait, which measures the normalized distance between the distribution centers of different cell types on the phase portrait. This reflects whether cells from different types are separated well on the phase portrait. The higher, the better.

(3) The fitting error on phase portrait, which measures the normalized distance between each cell to the constructed curve on the phase portrait. This reflects the fitting accuracy. The lower the better.

We then calculate these metrics based on the genes that can be commonly processed by all methods, and show the results on pancreas data in **Fig. R6** and revised **Fig.2**. For the “intra-class distance” and “inter class distance”, we take the Paired Samples T Test between the value obtained on the same gene by TFvelo’s phase portrait and the un/spliced phase portrait. Compared with those un/spliced based approaches, TFvelo can achieve lower intra-class distance, higher inter-class distance and lower fitting error.

In conclusion, transcriptional dynamics of RNA splicing may not provide sufficient signal, while TFvelo can address the problem for RNA velocity modeling.

Figure R6. The quantitative comparison on phase portrait between TFvelo and those RNA velocity methods based on un/spliced data. (a) The intra-class distance on phase portrait. (b) The inter-class distance on phase portrait. (c) The fitting error in phase portrait.

6. Results, Lines 195-200, the authors evaluate the performance of their method on data that has been simulated using a forward model that matches their assumptions. Unsurprisingly, they achieve low MSE loss. However, this is hard to judge, it would be helpful to include some baseline here so that the performance of this model can be compared to something.

Reply: We apologize for the insufficient explanation regarding the synthetic dataset. The purpose of the synthetic dataset in our study is to validate the feasibility of our computational method for modeling gene dynamics, which is a Generalized Expectation Maximization (GEM) algorithm. To the best of our knowledge, TFvelo is the first method to utilize GEM optimization in RNA velocity studies. In addition, the synthetic dataset was designed using the assumption that gene dynamics can be modeled by TF regulation, so the lack of un/splicing abundance makes this dataset not feasible to be processed by those un/spliced-based RNA velocity methods.

As for baselines, we now have added more comparison on multiple real world single cell datasets, with more metrics and more benchmark approaches.

We have improved our analysis for the synthetic dataset as well, by providing quantitative assessments on the inferred velocity and weights. As presented in Fig. R7 (Fig. 1n&o in main text), the Spearman correlation coefficient between the ground truth velocity/weights and inferred velocity/weights was quantified, showing a high performance.

Figure R7. Comparison between ground truth velocity/weights and inferred velocity/weights on synthetic dataset. The Spearman correlation coefficient between the ground truth and inferred value is shown. The red line refers to that the ground truth is equal to the inferred value.

7. Results, Line 205, ‘The pseudo-time predicted by TFvelo can identify the cell trajectory (Fig. 2a) well (...);’. If I understood the suggested model correctly, then each target gene is fitted independently. Accordingly, I would expect to have one latent/pseudo-time per target gene. The authors describe in the Methods how the actual pseudotime is computed, it’s not an aggregation over individual gene-specific latent times, but rather some DPT-like (Haghverdi et al. 2016) time inferred from the velocity-inferred transition matrix. As with the velocities, given that this method infers a time per gene, I would like to see an actual evaluation of these times (quantitatively). In scVelo, one latent time is estimated per gene, and these are pooled post-hoc to estimate a cell-specific latent time.

Reply:

(1) Visualization and quantitative analysis of gene-specific latent time.

In the baseline studies, only scVelo showed and analyzed the dynamics with gene-specific latent time in their paper. Following your suggestion, we show several of the genes for which both of TFvelo and scVelo can reconstruct dynamics (**Fig. R8**), and also provide the quantitative analysis (**Fig. R9**) about the gene-specific latent time.

To the best of our knowledge, we did not find any RNA velocity method that provides a quantitative evaluation of gene-specific time. To address the concern, for each method, we calculate the spearman correlation between gene-specific latent time and its cell-specific pseudotime on the genes that commonly modeled by both methods, since the cell-specific pseudotime inferred by both methods is always consistent with cell differentiation process according to the ground truth labels on this dataset (**Fig. 2a** in main text). As shown in **Fig. R9**, compared with scVelo, TFvelo achieves higher performance.

Figure R8. Comparison between scVelo and TFVelo on gene LITAF, H19 and MAML3. From left to right: Gene expression dynamics resolved along gene-specific latent time obtained by scVelo, the phase portrait fitting obtained by scVelo, Gene expression dynamics resolved along gene-specific latent time obtained by TFVelo, the phase portrait fitting obtained by TFVelo, and the expression level on UMAP.

Figure R9. The spearman correlation between gene-specific latent time and cell-specific pseudotime for all genes that are modeled by both scVelo and TFVelo.

(2) Why is the pseudotime computed in this way?

We followed the strategy adopted in previous methods including scVelo, which was implemented by `scvelo.tl.velocity_pseudotime()`. We calculate the pseudotime based on the directed velocity graph, which is defined as the cosine similarities between velocities and changing of expressions within neighboring cells, instead of the similarity-based diffusion kernel. For TFVelo, the

difference lies in that, the velocity and expression are defined with the total mRNA abundance, rather than spliced abundance.

8. Results, Lines 206-207: “(…) and the arrows are drawn to reflect the direction of cells development (Fig. 2b, Methods) using the derivative of pseudo-time.” I would strongly discourage the use of 2D stream plots to compare the performance of velocity methods. It has been demonstrated in numerous previous publications (e.g. (Bergen et al. 2021; Lange et al. 2022; Marot-Lassauzaie et al. 2022)) that these low-dimensional representations are often misleading. Alternatively, the authors could use methods that directly compare high-dimensional velocity vector fields, e.g. CellRank (Lange et al. 2022) or Dynamo (Qiu et al. 2022). In addition, I don’t quite understand why the authors consider the derivative of pseudotime here to visualize cell transitions, as their model directly learns a velocity per cell per gene. Why not make use of that information directly? Given that this method estimates velocities, I would like to see an actual, quantitative evaluation of those velocities.

Reply: Thank you for your suggestions.

We agree with your comments regarding the usage the derivative of pseudotime for post-processing, and have updated our post-processing strategy as well as the **Methods** section. The stream plot is now drawn in the way as those conventional approaches did, such as scVelo and CellRank (CellRank also calls functions from scVelo, e.g. `scv.pl.velocity_embedding_stream`), which is based on the transition probability matrix obtained from the high-dimensional velocity. In detail, we first calculate the velocity-inferred transition matrix, which is defined as the cosine similarities between velocities and changing of expressions within neighboring cells. Then a distribution over root cells can be obtained from the transition matrix. Finally, we calculate the pseudotime, which measures the average number of steps it takes to arrive at a cell after start walking from one of the root cells.

To address the concern regarding evaluation in high-dimensional space, on one hand, the high-dimensional velocity is assessed at a single-dimensional resolution, employing three quantitative metrics for phase portrait fitting of each individual gene. On the other hand, the pseudotime and velocity stream, derived from the high-dimensional velocity and abundance matrices, are subjected to quantitative evaluation using additional metrics. The details about these metrics are provided below. Consistent with other RNA velocity methods, TFvelo employs the 2D UMAP stream plot for visualizing the results.

As we addressed in response to your question 5, we have introduced three additional metrics for evaluating the fitting in the phase portrait. These metrics are “intra-class distance on phase portrait”, “inter-class distance on phase portrait” and “fitting error on phase portrait”. As shown in **Fig. R10 a-c** (also **Fig. 2d-f** in main text), in contrast to un/spliced-based approaches, the TFs-target space representation obtained through TFvelo can better distinguish cells from different types, and aggregate cells from the same type. Moreover, the TFvelo model significantly improves data fitting in the phase portrait.

Furthermore, we also employed two metrics adopted in VeloAE and UniTVelo, which are “Cross Boundary Direction Correctness (CBDir)” and “In Cluster Coherence (ICCoH)”. CBDir assesses

the correctness of transitions from one cell type to the next, utilizing boundary cells with ground truth annotations. ICCoh is computed using cosine similarity among cells within the same cluster, measuring the smoothness of velocity within the same clusters. As shown in **Fig. R10 d-e** (also **Fig. 2g-f** in main text), TFvelo achieves similar median value compared to UniTVelo, and significantly outperforms the other three methods.

Figure R10. Quantitative evaluation on pancreas dataset. (a) In cluster coherence measures the coherence of velocities within the same type of cell (the higher the better) (b) Cross boundary direction correctness measures the correctness of the inferred differentiation direction (the higher the better). (c) Fitting loss on phase portrait (the lower the better). (d) Cross Boundary Direction Correctness (the higher the better). (e) In Cluster Coherence (the higher the better). (f) Velocity confidence (the higher the better).

9. The problem of using low-dimensional velocity streams to judge the quality of their approach appears a few more times in the text, e.g. Results, Line 212: ‘In addition, compared to scVelo, TFvelo’s result on Epsilon and Delta cells can better align with the cell differentiation process’. In my opinion, 2D representations should not be used to judge the quality of their approach compared to previous approaches. Instead, in my opinion, the authors should use higher-dimensional, more robust means to compare differentiation trajectories throughout the Results section.

Reply: Thanks for emphasizing the limitation of evaluating the performance relying on 2D UMAP. We agree with your comments and have provided more quantitative evaluations, some of which have been employed in previous studies. Now, we conduct a more robust comparison by using six quantitative metrics in total. Please see our reply to your question 8 for details.

10. Results, Line 209, “TFvelo can even achieve a higher velocity confidence than scVelo without using splicing information (Fig. 2f), which indicates a higher consistency of RNA velocities across neighboring cells in the UMAP space”. I do not think it is a good idea to use a 2D UMAP space for quantitative comparison of projected velocities. If the authors would like to use the velocity confidence metric to compare their approach to previous approaches, I would suggest considering neighborhood relationships in higher-dimensional spaces, and not in the 2D UMAP. Also, as I outline further down below, I find this comparison biased towards TFVelo, as the authors apply very heavy post-processing to their velocities, which is likely to result in misleading, overly-smooth 2D representations.

Reply: Thanks for emphasizing the problem in 2D UMAP again. We agree with your comments and have provided more quantitative evaluations. Now we have updated our post-processing strategies. Please see our reply to your question 8.

As for the overly-smooth 2D representation, following your suggestion, we now have updated our post-processing strategies, please see our reply to your question 8 for details. And we apologize for the misleading. It is noticed that even in the old version, the smoothing strategy was applied only in the step of drawing stream plot. The velocities within neighbor cells used for measuring velocity confidence, or the pseudotime was actually not smoothed. As a result, the velocity confidence is still a fair way to compare the consistency of velocity. Please see the reply to your question 17, which is focusing on “overly-smooth”, for details.

11. Results, Line 226, “By comparison, scVelo fails to detect any dynamics because of the insufficient information provided by unspliced mRNA”. This could potentially be a strong selling point of TFVelo. Insufficient amounts of unspliced counts in important genes are an actual limitation of current velocity models that rely on sufficient levels of unspliced transcripts. Insufficient levels of unspliced counts have been explored a bit in the literature (Gorin et al. 2022; Sonesson et al. 2021), the authors should cite these works and explore this further. Being able to estimate robust velocities in the presence of very noisy unspliced count abundance would be a real selling point for TFVelo, I think.

Reply: Thank you for your suggestion and positive comments on this selling point. Now we have cited those papers you mentioned in the introduction, and further explored the situations with noisy and sparse unspliced counts. As shown in the response to your question 5, the high sparsity and noise of unspliced data is a large challenge for RNA velocity methods (**Fig. R5**). And we add the analysis as **Fig. 4** in main text and **Fig. S6** in SI.

In addition, we have also included in total six additional metrics and four existing conventional methods to compare the performance, as shown in **Fig. R10** and **Fig. 2** in main text. The comparison indicates that, TFvelo outperforms existing methods, in terms of phase portraits and velocity estimation for genes and cells.

12. Results, Line 233, “By comparison, scVelo cannot distinguish cells from multiple state on H19”. By looking at the phase portrait, it seems like scVelo detected a down-regulation in this gene, which is in line with what TFVelo predicts. I don’t understand what the authors refer to, this should be made explicit and quantified. Same for LITAF, seems like

both scVelo and TFVelo fit a transient up-regulation. And also, for ECE1, I cannot follow the author's claims of better fits by their model.

Reply: Apologize for the confusion. To explicitly and quantitatively compare the methods, now we have introduced three quantitative metrics to evaluate the model's fitting in the phase portrait, as shown in **Fig. R10** in response to question 8 and **Fig. 2** in main text, and rewritten the comparison parts.

In **Fig. R8 (Fig. S7 in SI)** and **Fig. R11 (Fig. 2c in main text)**, we provide examples of dynamic fitting for three genes. TFvelo is able to capture clear and accurate differentiation trajectories for all the example genes in the phase portrait, whereas all other methods cannot. As shown in **Fig. R8**, the gene-specific latent times obtained by TFvelo result in better data fitting compared to those obtained with scVelo.

Figure R11. Comparison of phase portrait fitting on LITAF, H19 and MAML3.

13. Based on my comments above, I disagree with the authors conclusion in Line 242: "These results suggest that TFvelo can provide more reasonable, robust and accurate dynamics for genes from single cell data." I'm unsure what the authors mean by "reasonable"; "robustness" has not been quantified, as far as I'm aware, and "accurate" has also not been quantified, I think. None of the comparisons presented in Fig. 2 justify these statements, I think.

Reply: we now have rewritten the conclusion in our paper, and added more comparison to demonstrate that TFvelo can achieve high accuracy, consistence and robustness, and the results by TFvelo can provide important biological insights.

(1) Accuracy: On the one hand, we evaluate the **fitting error in phase portrait**. Because only scVelo and UniTVelo show the constructed curve on phase portrait, we compare TFvelo with them using this metric, please see the detailed results in **Fig. R6c** in the response to your

question 5 (**Fig. 2f** in main text). On the other hand, we also evaluate the accuracy of cell trajectory by employing the metric “**Cross Boundary Direction Correctness**”, which is adopted in VeloAE and UniTVelo (**Fig. R10d** in our response to your question 8 and **Fig. 2g** in main text).

- (2) Consistence: In addition to the **velocity confidence** (**Fig. R10f** in the response to your question 8 and **Fig. 2i** in main text), we also assess the velocity consistence within each cell type, using the metric “**In Cluster Coherence**” (**Fig. R10e** in our response to your question 8 and **Fig. 2h** in main text), which is adopted in VeloAE and UniTVelo.
- (3) Robustness: Following your suggestion, we have provided more analysis on the cases where unspliced counts are sparse or noisy to better show the motivation and advantage of TFvelo. Results are show in **Fig. R5** in our reply to your question 5 (also **Fig. 4 c,e** in main text).
- (4) Biological insights: we further do KEGG /GO term enrichment analysis for the best fitted genes, and then explore the important TFs involved in the biological process. Specifically, using the pancreas dataset as an example, we observed a strong enrichment of the best-fitted genes in KEGG pathways associated with insulin secretion and the glucagon signaling pathway (**Fig. R12a** and **Fig. 3a** in main text). Next, we conducted an analysis of the weights assigned to each TF, which suggests that REST and HMGN3 consistently exhibit high absolute weights (**Fig. R12b** and **Fig. 3b** in main text).

When examining the UMAP distribution, it is clear that REST expression decreases during differentiation, while HMGN3 expression increases (**Fig. R12c** and **Fig. 3c** in main text). Previous research has identified REST as a key negative regulator of endocrine differentiation during pancreas organogenesis [PMID: 34385258, 32375045]. Earlier study has also report HMGN3 to be a regulator for insulin secretion in pancreatic cells [PMID: 19651901]. We further analyze the weights of these two TFs on modeling genes within the insulin secretion pathway. REST always has a negative weight, while HMGN3 always has a positive weight (**Fig. R12d** and **Fig. 3d** in main text), consistent with the previous findings.

Figure R12. Analysis of results on Pancreas dataset. (a) The KEGG pathway enrichment from the genes that are best fitted by TFvelo on pancreas dataset. (b) The TFs that always have higher weights on multiple target genes. The counts represent the number of target genes for which the

TF have an absolute weight larger than 0.5. (c) The distribution of REST and HMGN3 on UMAP. (d) The weights distribution of TFs REST and HMGN3 on modeling in the genes that in the insulin secretion pathway.

14. Results, Line 272, “The Gastrulation Erythroid dataset is the transcriptional profile for mouse embryo”;. The data the authors use here represents a small subset of the original data presented in (Pijuan-Sala et al. 2019). The authors should mention this, and describe why they focus on this particular subset of the data. They should avoid the impression of cherry-picking that subset of the original data where their model yields nice results.

Reply: We are sorry for the lack of introduction for this the dataset. Previous RNA velocity methods always select this subset to test their model. It is widely used in the these studies and can be directly loaded by `scvelo.dataset.gastrulation_erythroid()`. We use the same setting for a fair comparison with pervious methods. Now we clarify it in **Data Availability Statement**.

15. Results, Lines 369-379. In principle, it would be possible to distinguish between nuclear and cytoplasmic mRNA in MERFISH data and to use this information to fit a scVelo-type model, where nuclear and cytoplasmic counts take the role of unspliced and spliced transcripts, respectively. It would be nice if the authors could discuss this alternative approach, and outline potential drawbacks as well as advantages of their approach.

Reply: Thanks for your suggestion. We now have discussed this alternative approach in the section about this MERFISH dataset. The advantage for TFvelo is that TFvelo does not need the nuclear and cytoplasmic counts, which is rarely provided in single-cell datasets.

16. Discussion, Lines 423-426: “Compared with the previous RNA velocity model relying on splicing information, TFvelo can achieve better performance on those scRNA-seq datasets”;. I think this claim is too strong, given the analysis in the paper. Quantitative analysis was mostly done in terms of velocity confidence, which seems to rely on 2D UMAP information and overly heavy smoothing for TFVelo. Most of the other comparisons are qualitative, e.g. eye-balling where the streamlines point, and whether genes are visually up-or downregulated. To make such a broad claim in the discussion, the authors need to evaluate their model much more thoroughly against existing approaches, using robust metrics that operate in high-dimensional spaces. Also, given the wealth of velocity approaches that exist now (the authors list some of them in the intro), I wonder why they only compare with scVelo. There are known limitations of the scVelo velocity model (e.g. constant kinetic parameters, etc.), some of which have been partially resolved in follow-up models.

Reply: Thanks for the suggestion. Now we have rewritten our claims and added more comparison with more existing methods and more quantitative metrics to support the claim. In the discussion, we conclude that “Experiments on a synthetic dataset validate that TFvelo could detect the underlying dynamics from the data. Results on various scRNA-seq datasets, a MERFISH dataset and a 10x multi-omics dataset further demonstrate that TFvelo”. Please see the details in the discussion section.

Specifically, we include scVelo, UniTVelo, Dynamo and cellDancer as baselines for the velocity/pseudotime comparison. In addition, as we replied to your question 5 (**Fig. R6** and **Fig.**

2d-f in main text), we introduce three metrics to evaluate the fitting in phase portrait, which are “fitting error on phase portrait”, “intra-class distance on phase portrait” and “inter-class distance on phase portrait”. Furthermore, as we replied to your questions 8 (**Fig. R10** and **Fig. 2g-i** in main text), we also employ three quantitative metrics for velocity comparison, including the “In Cluster Coherence (ICCoh)” and the “Cross Boundary Direction Correctness (CBDir)” adopted in VeloAE and UniTVelo, and the velocity confidence.

As for the “heavy smoothing”, we apologize for the misleading. Even in the original manuscript, the smoothing was only applied to the arrow plot, which would not influence the velocity confidence. And following the reviewer’s suggestion, now we update our post-processing procedure to adopt the strategy that employed in scVelo to draw the arrow plot, without derivative of pseudotime or heavy smoothing. Details are provided in our reply to your question 17 and also the **Methods** section in main text.

17. Supplementary Figure 2: The stream derived from the pseudo-time is projected into the UMAP-based embedding. The description of this appears in the Methods, Line 574-580. The authors don’t work with the actual velocities their method estimates but rather use scVelo to estimate a velocity pseudotime, which is a symmetrized version of DPT, and take derivatives of this pseudotime to draw stream-lines in a UMAP. It appears that they take derivatives of the pseudotime in the 2D UMAP space, which I would not advise doing, given the known pitfalls of 2D representations. Also, they apply further smoothing in the 2D UMAP by binning cells, and smoothing over neighboring bins. Given this extremely heavy post-processing and smoothing, the comparison with scVelo in terms of velocity consistency is biased towards TFVelo and no longer reliable. If the authors would like to use a metric like velocity confidence to compare with scVelo, they should compare quantities that are actually comparable, i.e., velocities estimated by either method (not pseudotime-derivatives), in the same space, with the same post-processing applied.

Reply: We apologize for the misleading here.

(1) We clarify that after obtaining the velocity of each gene with TFVelo, we calculate the transition probability matrix based on the genes velocity and expression difference within neighbor cells. Then pseudo time can be inferred based on the transition probability matrix, using the similar strategy that applied in scVelo. The difference is that in scVelo, the transition probability matrix is obtained with splicing velocity.

(2) Following your suggestions, we have updated our post-processing methods. After obtaining the model on each gene, we firstly filter the genes according to the loss in phase portrait. Then calculate the pseudotime in the same way as that in scVelo based on filtered genes. After that, we select the genes with fitting time consistent with the pseudotime for further cell velocity inference. Finally, we can obtain the stream plot from these selected genes, where the strategy for plotting stream arrows is the same with that in scVelo.

(3) As we replied to your questions 8 (**Fig. R10** and also **Fig. 2d-i** in main text), we have added more quantitative metrics for evaluation.

(4) In addition, we clarify that even in our old version, the smoothing strategy was only applied in the step of drawing stream plot. The velocity or the pseudotime is not smoothed. As a result, the velocity confidence, which is based on the velocities with neighboring cells should still be a fair comparison in term of velocity consistency.

18. The authors should comment on the scalability of their approach, compared to competing methods like scVelo.

Reply: The time complexity with respect to the number of samples (cells) for TFvelo is the same as scVelo. However, TFvelo requires more running time than scVelo due to an additional step involving the optimization of weights $W \in R^{n_{TF}}$ through linear regression in each iteration, where $n_{TF} \sim 10^1$ represents the number of TFs involved in modeling this gene. This additional step is necessary to incorporate regulatory information, ensuring a better fit for gene dynamics in the phase portrait and improving robustness for handling high sparsity data.

Minor comments

19. Abstract: “most existing RNA velocity models can only be applied to datasets with unspliced/spliced or new/total RNA abundance information”; Why would this be a limitation? Spliced/unspliced read abundance can be estimated from any standard scRNA-seq protocol, as shown already in (La Manno et al. 2018). It would be good to add a half-sentence, listing some examples where this information might not be available.

Reply: Thanks for suggestion. Now we put emphasis on discussing the high noise and sparse in un/spliced data, which is quantitatively analyzed (please see our response to your question 5 for details). We also give some examples where un/spliced information might not be provided, including those datasets from imaging-based technologies, like MERFISH (Fig. 6e,f).

20. Intro: “However, it is cost and time-consuming to obtain these new omics data with additional labeling”; I agree it would be costly to collect multi-modal single-cell data with additional metabolic labeling, but that’s not the type of information required by the approaches outlined in the previous sentence: MultiVelo (Li et al. 2022) and Lior Pachter’s (Gorin, Svensson, and Pachter 2020) Protein velocity models don’t need labeling data.

Reply: Sorry for the unclear statements. Here we refer to that it is cost and time-consuming for methods with additional information, including both labeling data and other omics data. Now we have rewritten it more carefully.

21. Intro, Line 72: “Notably, the TFs with the positive weights are significantly enriched with TF set functionally linked to the target gene”; What are these weights? At this point in the manuscript, the authors have not introduced any weights and this statement is confusing.

Reply: We apologize for the absence of this part. The weights analysis is provided in **Table S1** in Supplementary Information. We have added the statement into the **Findings** section that “as shown in **Table S1**, the TFs with non-zero positive weights are significantly enriched in TFs set functionally linked to the target gene (according to ENCODE TF-target datasets) with p value of $6.66e-06$ (one-sided t-test)”.

22. Results, Line 101. How are candidate TFs determined here? This is crucial and should be stated explicitly in the main.

Reply: Sorry for the miss part. Actually, we did not apply any selection to TFs. All TFs passing the preprocess were fed into the LASSO regression. We have clarified in the Findings section that “we input all TFs [PMID: 29425488] into LASSO, and get a sparse weights-vector where only some of the TFs have non-zero weights”.

23. Supplementary Figure 2: The colorbar is feasible for all cell type annotations within this figure; There is no colorbar in this figure.

Reply: We have updated our figures and legends accordingly.

24. Results, Line 107 and 157. The definition of TF weights is not consistent, once it's w_g and once W_g . If the latter is meant to represent a matrix, that's a contradiction with $W_g \in \mathbb{R}^{n_{TF}}$ (Line 157). I would urge the authors to use consistent definitions of all symbols throughout the main and methods, and to define these properly, including the correct shapes. Also, I would be consistent in the use of bold-face for symbols. For example, w_g in Line 107 is bold-face, but t in Line 157 is not, even though it's shape $t \in \mathbb{R}^{n_{cell}}$ indicates that this is a vector. These inconsistencies make the mathematical modeling hard to follow.

Reply: Thanks for the helpful suggestion. We have corrected it.

25. Fig. 1, I cannot find panel b, I think it has been mislabelled as c. In panel c (there are two of them), the color bar should be labeled in the figure. Also, this figure contains text that is much too small, e.g. axis labels in c-e. In panel i, why would pseudo-time inference be a downstream analysis? I thought pseudotime was estimated jointly with the other model parameters? Panel j, why does this represent gene dynamics? Hard to see the link here. Panel k, I thought TF-target gene relationships were the input to the model, how can these be an output as well? This needs to be explained in more detail, what is the information the model gives here? Panel o, text overlaps with panel elements. Also, unclear whether the performance shown here just relates to the example shown in panel l, or to the entire simulated data, which should contain 100 synthetic gene dynamics (Results, Line 190).

We apologize for these mistakes. We have updated our figures and correct them according to your helpful suggestions.

(1) Why would pseudo-time inference be a downstream analysis?

The models are firstly constructed for each gene individually, and the pseudotime inference is a following step based on all learnt models for all genes.

(2) Why does this represent gene dynamics?

Phase portrait is a widely used way to show the spliced and unspliced RNA dynamics in RNA velocity methods. Here through the phase portrait we can simultaneously describe the change of both the target and TF along with inferred latent time.

(3) Considering that TF-target gene relationships were the input to the model, how can these be an output as well?

Sorry for the confusion and we agree with your comments. Since TFvelo has already used as input the TF-target pair information from known databases, we now update our statements and introduce TFvelo as an RNA velocity method that can provide regulation analysis in the pseudotime trajectory, rather than a GRN inference method. The regulation analysis is conducted in terms of

KEGG/GO term enrichment and key TFs analysis. Please find the details in our reply to your question 13 (**Fig. R12**).

(4) Unclear whether the performance shown here just relates to the example shown in panel l, or to the entire simulated data, which should contain 100 synthetic gene dynamics

We have updated these panels to show the relationship between ground truth weights/ velocity and inferred weights/ velocity. We introduce more details about it in the section “**TFvelo can reconstruct the dynamic model and detect the regulation relationship on synthetic datasets**” in main text. In the updated **Fig. 1 n,o**, the plots and statistical results are obtained on the entire simulated data, which contains 200 synthetic gene dynamics now.

26. Results, Line 203, “To evaluate TFvelo on scRNA-seq data, we first apply it to the dataset of E15.5 mouse pancreas”; It would be nice to describe the pancreas dataset and to cite it here, where it is first considered.

Reply: Thanks for your careful reviewing. Now we cite it as you suggested.

27. Fig. 2d: This plot is missing a colorbar, x- and y-axis labels. Also, what’s the relevance of revealing a gene expression cascade in this context? What biological process does it correspond to here? There are different endocrine populations formed in this process, this plot seems to smooth over all of them.

Reply: Now we have updated the figures. This heatmap (**Fig. R13**) can show the changing of transcriptomics along the pseudotime, and is widely adopted in previous velocity studies. In addition, these endocrine populations, including alpha, beta and gamma cells, are differentiated from pre-endocrine cells independently, which arrive at different cell fates and share the latter pseudotime (the right down black box in the figure). As a result, they can not be well separated in the axis of pseudotime on this heatmap. This is also a common phenomenon in results of other approaches.

Figure R13. Gene expression dynamics resolved along pseudo time.

28. Results, Line 215, the fact that a subset of ductal cells is cycling has already been observed in the original publication (Bastidas-Ponce et al. 2019)

Reply: Thanks for pointing it out. Following your suggestions, now we have updated the results and on pancreas dataset (Fig. 2 and Fig. 3 in main text), which could provide the more comprehensive analysis.

29. Fig. 2g-k, does TFVelo always fit the full circle, even though only part of these dynamics might actually be present in the data? See e.g. panel g.

Reply: TFVelo does not always fit a full circle. It can fit the dynamic of a gene with part of an elliptic curve on the phase portrait (some examples are shown in Fig. R14).

Figure R14. Phase portrait fitting for more gene examples.

30. Fig. 2c, I don't understand the non-smoothness; the authors are referring to with red-circles for scVelo. From looking at the 2D stream plot, everything looks equally smooth, I think. Again, these statements have to be either removed or backed by quantitative analysis in a higher-dimensional space.

Reply: Following your suggestions, now we have removed the red-circles and provided more quantitative comparisons with more baseline approaches to better support our statements, instead of relying on the 2D steam plot. Details are provided in our response to your question 8.

31. Results, Lines 259-268. It is unclear to me how TFVelo contributes to a better understanding of TF-target interactions. The authors should clarify how exactly they encode prior knowledge about putative regulators for each target gene, so that I'm able to judge to what extent the model has learned something new. Also, this claim should be quantified, beyond a single gene (JUND).

Reply: Sorry for the confusion and we agree with your comments. Since TFVelo has already used as input the TF-target pair information from known databases, we now update our statements and introduce TFVelo as an RNA velocity method that can provide regulation analysis in the pseudotime trajectory, rather than a GRN method that can infer de novo TF-target regulation.

Biological interpretability is now conducted with GO term and KEGG pathways enrichment analysis, instead of relying on individual TF-target pair. On the pancreas dataset, we provide a more detailed analysis on the two TFs that may play the most significant roles according to the learned weights. The results are shown in **Fig. R12** to your question 13 (**Fig. 3** in main text).

In addition, **Fig. R15** (**Fig. 3e** in main text) provide the visualization of the abundance of TF and target along the latent time. There is a clear phase delay between learned representation of TFs and target gene. We further analyze the phase delay between target gene and individual TF with high absolute weight. Within these two examples, HMGN3 has a positive weight on modeling SURF, where a gap between the phases of them can be clearly observed. On the contrary, REST has a negative weight on modeling ECE1, where their changing shows obvious negative correlation. These biological-inspired analysis can help to better understanding of TF-target interactions.

Figure R15: Phase visualization of gene SURF4 and ECE1 on Pancreas dataset. On each row, from the left to the right: the dynamics on phase portrait, the expression on UMAP, the phase delay between the learned regulation representation (WX) and target gene, and the phase delay between individual TF with the high weight and target gene.

32. Results, Lines 297-308. To establish TFVelo as a framework to detect regulatory interactions would require the authors to present a systematic evaluation over a large set of previously reported gene interactions, and some metrics to score the quality of their predictions. Also, they should compare their tool with other GRN learning approaches.

Reply: Sorry for the confusion again. As explained by the response to question 31, we update our statements and introduce TFVelo as an RNA velocity method that can analyze the underlying regulation which supports the gene dynamics, rather than a GRN method to infer casual gene relationships. And TFVelo has already selected TFs and their potential targets as input based on known TF databases. For clearer clarification, we now have introduced those gene regulation inference methods in the **Introduction** section and discussed the difference between them and TFVelo.

33. The analysis done in figures 2-4 is essentially the same, maybe the authors could consider highlighting different parts of their method on different applications, where each analysis is most relevant, given the biology at hand?

Reply: Thanks for the suggestion. We have added more comprehensive analysis based the results on each dataset. On Pancreas dataset, we provide biological analysis based on KEGG pathway enrichment and the weights of TFs (**Fig. 3** in main text). On gastrulation erythroid dataset, we provide a sparsity analysis in terms of unspliced, spliced, and total mRNA abundance to illustrate the advantage of TFVelo (**Fig. 4** in main text). On 10x mouse brain multi-omics dataset, we show 3D phase portrait by adding the spliced and unspliced information back (**Fig. 5** in main text). On human ESC and MERFISH dataset, we point out the advantage of TFVelo that it can work without splicing information (**Fig. 6** in main text).

34. Results, Lines 377-379. The authors filter out M-stage cells as they argue cycling dynamics are not supported by TFVelo. If I recall correctly, Bergen et al. showed in their original scVelo publication (Bergen et al. 2020) that they could resolve cycling ductal cells. Could the authors comment on why this is possible with scVelo, but not with TFVelo?

Reply: Now we have updated TFVelo procedures as the reviewer suggested. As a result, the results on MERFISH data are also updated consequentially, and now the cycling dynamics can be captured by TFVelo. Generally speaking, the previous post-processing steps aimed at selecting the genes that are best fit and then constructed velocity arrows based on them. Because the pseudotime of terminal points were assigned as 0 or 1, respectively, it is not feasible to show cell cycle process with pseudotime inference approaches. To adding the M state, instead of relying on pseudotime, we use the information on phase portrait to select genes. Please see the details in our **Methods** section.

35. Discussion, Line 409: ‘‘However, the existing methods often fails to fit the cell dynamics well in the unspliced/spliced space, which is because of the weak signal and high noise at single cell resolution.’’ The claim about ‘‘weak signal and high noise’’ is central to this paper and should be demonstrated and quantified much more.

Reply: Thanks again for this comment. Now we have added a series of quantitative analysis on the noisy and sparsity of un/spliced data, to better illustrate our motivation and the advantage of

TFvelo. Please find the details in our answers to your question 5 (**Fig. R5**), the updated **Results** section in main text (**Fig. 2d-i**), and the Supplementary Information (**Fig. S6**).

Implementation

36. Most of the code is copied from scVelo's GitHub repository, without appropriate acknowledgments of the source. This is bad practice from two perspectives:

Original source code should be acknowledged properly, in a prominent place.

Copied code creates an unnecessary overhead, and increases the maintenance load. The authors should avoid copying any code, instead, add scVelo as a dependency, and import from scVelo where necessary.

Further, this package has neither proper documentation, testing, nor tutorials. I don't find this appropriate for publication in a high-impact journal like Nature Communications.

Reply: Thanks for the valuable suggestion for implementation.

(1) Because of the wide usage of scvelo and its clean, well-organized and efficient codes, we now develop python package "TFvelo" based on the structure of scvelo. We have added more appropriate, prominent acknowledgments for scvelo in our GitHub page, and also included that in our license.

(2) Following your suggestion, we have deleted the functions that can be directly import from scVelo. However, the imported functions from scvelo are not always feasible for implementing TFvelo. Because there are always some differences between the functions in TFvelo and scvelo, even they have the similar function. For instance, in scvelo.pp.moments(), they obtain moments for "spliced" and "unspliced" expression, whereas in TFvelo.pp.moments(), we require moments for the "total" expression. So, we have to modify these functions even they have the same name.

(3) We apologize for the current insufficient documentation on the GitHub page. Now we have further developed the package of TFvelo, and provided more comprehensive documentations and tutorials for TFvelo.

References

Bastidas-Ponce, Aimée, Sophie Tritschler, Leander Dony, Katharina Scheibner, Marta Tarquis-Medina, Ciro Salinno, Silvia Schirge, et al. 2019. "Comprehensive Single Cell mRNA Profiling Reveals a Detailed Roadmap for Pancreatic Endocrinogenesis." *Development* 146 (12). <https://doi.org/10.1242/dev.173849>.

Bergen, Volker, Marius Lange, Stefan Peidli, F. Alexander Wolf, and Fabian J. Theis. 2020. "Generalizing RNA Velocity to Transient Cell States through Dynamical Modeling." *Nature Biotechnology* 38 (12): 1408-1414.

Bergen, Volker, Ruslan A. Soldatov, Peter V. Kharchenko, and Fabian J. Theis. 2021. "RNA Velocity: current Challenges and Future Perspectives." *Molecular Systems Biology* 17 (8): e10282.

Gorin, Gennady, Meichen Fang, Tara Chari, and Lior Pachter. 2022. "RNA Velocity Unraveled." *bioRxiv*. <https://doi.org/10.1101/2022.02.12.480214>.

Gorin, Gennady, Valentine Svensson, and Lior Pachter. 2020. "Protein Velocity and Acceleration from Single-Cell Multiomics Experiments." *Genome Biology* 21 (1): 39.

Haghverdi, Laleh, Maren B. Hemberg, F. Alexander Wolf, Florian Buettner, and Fabian J. Theis. 2016. "Diffusion Pseudotime Robustly Reconstructs Lineage Branching." *Nature Methods* 13 (10): 845–848.

La Manno, Gioele, Ruslan Soldatov, Amit Zeisel, Emelie Braun, Hannah Hochgerner, Viktor Petukhov, Katja Lidschreiber, et al. 2018. "RNA Velocity of Single Cells." *Nature* 560 (7719): 494–498.

Lange, Marius, Volker Bergen, Michal Klein, Manu Setty, Bernhard Reuter, Mostafa Bakhti, Heiko Lickert, et al. 2022. "CellRank for Directed Single-Cell Fate Mapping." *Nature Methods* 19 (2): 159–170.

Li, Chen, Maria C. Virgilio, Kathleen L. Collins, and Joshua D. Welch. 2022. "Multi-Omic Single-Cell Velocity Models Epigenome–transcriptome Interactions and Improves Cell Fate Prediction." *Nature Biotechnology*, October, 1–12.

Marot-Lassauzaie, Valérie, Brigitte Joanne Bouman, Fearghal Declan Donaghy, and Laleh Haghverdi. 2022. "Towards Reliable Quantification of Cell State Velocities." *bioRxiv*. <https://doi.org/10.1101/2022.03.17.484754>.

Pijuan-Sala, Blanca, Jonathan A. Griffiths, Carolina Guibentif, Tom W. Hiscock, Wajid Jawaid, Fernando J. Calero-Nieto, Carla Mulas, et al. 2019. "A Single-Cell Molecular Map of Mouse Gastrulation and Early Organogenesis." *Nature* 566 (7745): 490–495.

Qiu, Xiaojie, Yan Zhang, Jorge D. Martin-Rufino, Chen Weng, Shayan Hosseinzadeh, Dian Yang, Angela N. Pogson, et al. 2022. "Mapping Transcriptomic Vector Fields of Single Cells." *Cell* 185 (4): 690–711.e45.

Soneson, Charlotte, Avi Srivastava, Rob Patro, and Michael B. Stadler. 2021. "Preprocessing Choices Affect RNA Velocity Results for Droplet scRNA-Seq Data." *PLoS Computational Biology* 17 (1): e1008585.

Reviewer #3:

The authors developed a new method TFvelo for calculating expression velocity of gene expression in single cell expression profiles using regression-based modeling of TF-target expression relationship. TFvelo looks useful to capture dynamics and key regulatory factors regulating dynamics. However, the current manuscript seems to lack significance as follows:

Reply: Thanks for your encouragement and constructive comments. We now have improved our paper according to your suggestions.

1. TFvelo depends on the regression of TF and target gene expression. The list of TFs must be an important factor for the TFvelo's performance. Please provide the database or criterion of TF list.

Reply: Sorry for missing the list. Now we provide the list of TFs in Section 7 of SI. The TFs list are provided by the *Cell* paper: The Human Transcription Factors with PMID of 29425488.

2. Figure 1o shows the performance of TFvelo in capturing dynamics of a synthetic data. Please provide the comparison with other GRN constructors such as SCENIC, SCODE, Scribe, GRNVBEM, CellOracle, LEAP, TENET, SINGE.

Reply: Sorry for the misleading. We agree that reconstructing the gene dynamics and regulation causality is highly related. However, TFvelo has different goal and input/output to methods like TENET or SCENIC. Same to all RNA velocity models, the goal of TFvelo is to infer RNA dynamics, pseudo-time, and cell trajectories using TF-target information, rather than to infer casual gene relationships. And TFvelo has already used TFs and their annotated target pairs as input based on known TF databases. As a result, it is unfeasible to compare TFvelo with other GRN methods due to different input information and goal. For clearer clarification, we now have introduced those gene regulation inference methods and discussed the difference between them and TFvelo in the **Introduction** section in main text.

In addition, the main purpose of conducting the synthetic dataset analysis is to verify the performance of TFvelo for modeling each gene, which is a generalized Expectation Maximization (GEM) algorithm. To the best of our knowledge, TFvelo is the first GEM method in this topic. To better illustrate the performance, as shown in **Fig. R16**, the updated **Fig. 1 l-o** and **Section 3** in **SI**, we compare the ground truth weights/ velocities and the inferred weights/ velocities, with high spearman correlation coefficient shown in the panels respectively, indicating that TFvelo can reconstruct the gene dynamics well from the synthetic data.

Figure R16. Comparison between ground truth velocity/weights and inferred velocity/weights on all synthetic dataset. The Spearman correlation coefficient is between the ground truth and inferred value.

3. The comparison of TFvelo and scVelo in Figure 2 does not look like fair comparison. scVelo also provide critical gene set for the dynamics. The counter-examples also should be provided. Specifically, Please provide the phase portrait of the gene set provided by scVelo.

Reply: Following your suggestions, now we show phase portrait for counter-examples in **Fig. R17** for a fair comparison, in most of which TFvelo also performs well. To be noticed, the cells in purple, pink, blue and lighter blue mean different endocrine populations, which are differentiated from pre-endocrine (in green) cells independently. So, they could share the latent time and mixed on the plot in **Fig. R17**. By comparison, the cells in red (Ductal), yellow (Ngn3 high EP) and green (pre-endocrine) should be ordered clearly. Actually, scVelo selected four genes for illustrating the phase portrait fitting, which are NNAT, CPE, PPP3CA and ACTN4, on the pancreas dataset. On NNAT, Scvelo can only model the small part of up-regulation process in beta cells (labeled in light blue), while TFvelo can detect much more dynamics on other cells types as well, especially the early differentiation process from Ductal (red) to pre-endocrine (green). On CPE, although both methods can the detect up-regulation dynamic, scVelo failed to model the red part, which means the ductal cells. On PPP3CA, both methods can not correctly capture dynamics of the whole differentiation process. The ACTN4 gene is filtered out during preprocessing in TFvelo.

Due to the high sparsity and noise in unspliced/spliced expression, it is hard to find a gene whose phase portrait is well fitted by conventional RNA velocity approach. That is the motivation for TFvelo, in which a representation of TF regulation is learned to generate a better gene dynamics and phase portrait.

Figure R17: Comparison between scVelo and TFVelo on NNAT, CPE and PPP3CA, which are the genes demonstrated in scVelo paper. In each row, the first two columns from the left show the expression along with the gene-specific latent time, and spliced/unspliced level in phase portrait by scVelo. The next two columns show the results by TFVelo. The last column is the distribution of gene expression on UMAP space. According to the ground truth label, during the differentiation process, the cell order should follow the sequence of red (Ductal cells), yellow (Ngn3 high EP), green (pre-endocrine), and then other colors (endocrine, including alpha, beta, gamma and delta).

In addition, we provide comprehensive analysis, especially for the sparse and noisy unspliced/spliced data to better illustrate the superior of TFVelo. Considering the high dropout rate and noise in scRNA-seq data, dividing total mRNA into the unspliced and spliced could result in much higher sparsity (**Fig. R18a** and **Fig. S6** in SI), consist with the findings shown in velocyto [PMID: 30089906]. As a result, the methods based on splicing may fail to capture dynamics due to extreme sparsity. In contrast, TFVelo models the gene dynamics based on total expressions of each gene, thus can extract more information from multiple TFs and target. Two sparse genes are shown in **Fig. R18b** (**Fig. 4e** in main text and **Fig. S6** in SI), in which the sparsity of unspliced count for SURF4 is 99.7%, and that for CD24A is 99.8%. While both of them can pass the preprocessing of scVelo, scVelo cannot build dynamics for them, and UniTVelo or cellDancer cannot construct the correct dynamics, neither. Dynamo just filters them out before the step of dynamics construction. **Fig. R19** (**Fig. 1a** and **Fig. 2c** in main text) shows three genes for example to illustrate the influence of noisy in splicing data. Based on the UMAP distribution, the expression levels of LITAF, H19 and MAML3 initially increase and later decrease. TFVelo can clearly capture this process, while scVelo cannot.

Figure R18. The challenges of sparsity for scRNA-seq data. (a) The number of genes whose sparsity is lower than 0.9, based on the unspliced, spliced and total mRNA abundance, respectively. The sparsity of a gene is defined as: $\text{Sparsity} = \frac{\text{The number of cells that the count of this gene is zero}}{\text{The total number of cells}}$. (b) Comparison on phase portrait fitting for SURF4 and CD24A on Pancreas dataset, where the unspliced count is very sparse. The last column is the distribution of gene expression on UMAP. Results of scVelo and Dynamo are not shown because they can not construct dynamic models on these genes.

Figure R19. Comparison between scVelo and TFvelo on LITAF, H19 and MAML3. From left to right on each row: Gene expression dynamics resolved along gene-specific latent time obtained by scVelo, the phase portrait fitting obtained by scVelo, gene expression dynamics resolved along gene-specific latent time obtained by TFvelo, the phase portrait fitting obtained by TFvelo, and the expression level on UMAP.

4. In overall, TFvelo seems a very useful bioinformatics tool. However, the manuscript does not provide new findings. Please provide what is new findings such as key regulator factors. And please validate the findings with at least one example.

Reply: Thanks for your encouragement and suggestion.

As for new findings with TFvelo, we now focus on TFs which always have large weight values for multiple genes, and explore the functions of them. It is found the most of these TFs have been verified by recent studies that they play important roles for regulating the corresponding process.

Specifically, using the pancreas dataset as an example, we observed a strong enrichment of the best-fitted genes in KEGG pathways associated with insulin secretion and the glucagon signaling pathway (Fig. R20a). Additionally, we conducted an analysis of the weights assigned to each TF (Transcription Factor). Our findings show that REST and HMG3 consistently exhibit high absolute weights for different target genes (Fig. R20b).

When examining the UMAP distribution, it is clear that REST expression decreases during differentiation, while HMG3 expression increases (Fig. R20c). Previous research has established REST as a key negative regulator of endocrine differentiation during pancreas organogenesis [PMID: 34385258, 32375045]. Earlier study has also report HMG3 to be a regulator for insulin secretion in pancreatic cells [PMID: 19651901]. We further analyze the weights of these two TFs on modeling genes within the insulin secretion pathway. REST always has a negative weight, while HMG3 always has a positive weight (Fig. R20d), which is again consistent with the previous findings.

Figure R20. Analysis of results on Pancreas dataset. (a) The KEGG pathway enrichment from the genes best fitted by TFvelo on pancreas dataset. (b) The TFs that always have high weights on multiple target genes. The count represents the number of target genes for which the TF has an absolute weight value larger than 0.5. (c) The distribution of REST and HMG3 on UMAP. (d) The weights distribution of TFs REST and HMG3 on the genes in the insulin secretion pathway.

Reviewer #4:

This manuscript by Li and colleagues describes an extension of RNA velocity, a method originally developed by La Manno et al. 2018 to estimate the evolution of gene expression over time from single-cell RNA-sequencing data. The proposed extension harnesses the relationship between expression of transcription factors and their target genes to predict target gene-specific RNA velocities, allowing velocity estimation in circumstances where unspliced (U) and spliced (S) content cannot be accurately obtained or measured.

Previous methods for estimating RNA velocity of single cells use U/S counts to fit gene-specific phase portraits and infer dynamic cell transitions based on this ratio. However, for some datasets, unspliced counts contain too high sparsity, are not made available when published (i.e. data privacy issues for raw human sequencing data), or cannot be easily measured with the data acquisition technique (i.e. spatial transcriptomics methods). The extension described in this manuscript implements a linear model to fit gene expression for a weighted combination of transcription factors and a single target gene to a phase portrait, instead of using unspliced and spliced counts. The model is solved with the expectation-maximization (EM) algorithm to iteratively learn the transcription factor weights, the phase portrait shape parameters, and cellular latent time. TFvelo's implementation is built upon the previous scvelo framework from Bergen et al. 2020, which also learns a similar latent time and phase portrait parameters by EM.

The manuscript is organized as follows:

The authors begin by outlining the main claim of their work: given a target gene (Target), cell-specific velocity estimates inferred with scvelo (for that same target gene) can instead be inferred with a linear equation comprised of a weighted combination of the RNA expression levels of transcription factors (TF). These TFs are known to either positively or negatively regulate expression of such Target, according to database references, and can therefore have a positive or negative weight. First, the authors show that the weights for each transcription factor, when fit using LASSO regression, yield a velocity estimate for the Target gene that correlates well with scvelo estimates. Next, the authors use an EM algorithm to estimate velocities from paired TF-Target expression data and corresponding phase portraits (Figure 1). The model is evaluated on synthetic data and a small number of published datasets. The authors claim their method estimates velocities that more closely reflect known biology, such as during exit of pancreatic ductal cells from the cell cycle (Figure 2) and erythroid differentiation (Figure 3). The authors further apply TFvelo to a scRNA-seq dataset missing U/S counts (Figure 4) and to MERFISH spatial transcriptomics data, for which no U/S information is collected due to experimental limitations of probe design (Figure 5).

I enjoyed reading this work, and I think it contains interesting computational tricks of value worthy of publishing among the single cell and RNA velocity community. However, I have concerns for which I expect major revisions in order to accept it for publication in Nature Communications. (1) the method requires significantly more evidence to be considered a convincing and sufficiently characterized improvement over existing velocity methods; (2) while the conceptual underpinnings of TFvelo are clever, I am not fully convinced this manuscript introduced a substantial new advancement to the RNA velocity framework; (3) I have concerns about the ability

to claim biologically-meaningful interpretations with a velocity estimation derived from transcription factors and their target genes, which behave according to more complex metabolic processes than those for traditional RNA velocity methods relying on linear progression from unspliced to spliced; (4) the code cannot be run and therefore assessing reproducibility of the method (an essential objective) is not possible at this time.

Reply: Thanks so much for your encouragement and sharing those biological insights, which not only helps us in the revision, but also inspire us for the future exploration.

(1) We now have added more baseline approaches, including UniTVelo (NC 2022), Dynamo (Cell 2022), VeloDancer (NBT 2023), into comparison. In addition, we employed more metrics for evaluating the methods from different aspects, including the “In Cluster Coherence (ICCoH)” and the “Cross Boundary Direction Correctness (CBDir)” proposed by UniTVelo. We have also introduced three additional metrics for evaluating the fitting in phase portrait, which are “fitting error on phase portrait”, “intra-class distance on phase portrait” and “inter-class distance on phase portrait”. These metrics can better show the advantage of TFvelo quantitatively. Please see our reply to your question 8 for details (**Fig. 2d-i** in main text).

(2) Inspired by your comments, we have re-organized our introduction and provide more quantitative analysis to better explain the advancement of TFvelo. Firstly, TFvelo is a new concept of velocity. We agree that our velocity inferred from changing trend of TF with target and the velocity by unspliced/spliced counts are different. However, the underlying idea is similar that the velocity can be inferred by the abundance of two biological entities with phase delay, such as un/spliced RNA count and TF/target expression level. Secondly, as shown in our revised paper, TFvelo can provide comparable or even better temporal performance on predicting gene and single cell dynamics (**Fig. 2** and **Fig. 3**), further validating the velocity concept inferred by TF and target. Furthermore, due to sparsity and noise of single cell data, it is hard for existing RNA velocity approaches to extract and fit the gene dynamics well, especially in phase portrait. In contrast, by modeling velocity based on transcriptional regulation, TFvelo can address this issue and achieve better performance on different single cell datasets. In conclusion, given the new quantitative analysis, comparison, explanation and discussion following the reviewer suggestions, we believe that TFvelo introduced a substantial new advancement to the RNA velocity framework.

(3) While we agree that the transcription regulation mechanisms is very complicated, recent studies have validated the feasibility of linear models for modeling the transcriptional regulatory within genes on single-cell RNA data [PMID: 21562557 (*Nature*, 2011), 36762475, 35300460], indicating that linear model is a simple but efficient tool for transcriptional regulation modelling. Despite of the difference between TFvelo and spliced-based velocity method, TFvelo can always provide comparable or even better temporal performance on predicting gene and single cell dynamics, as shown in our paper. In addition, TFvelo can correctly find the dynamic of individual gene in the phase portrait in the Pancreas dataset (**Fig. 2c** and **Fig. S7**), while spliced-based velocity methods (scVelo, UniTVelo, Dynamo and cellDancer) cannot due to the severe noise and sparsity of unspliced/spliced abundance, further validating the velocity concept inferred by TF and target.

(4) We apologize for the insufficient documentation on the GitHub page. We have uploaded the TF-Target list from ENCODE and ChEA, refactorized our code, and created better documents and tutorials to guarantee the usage and reproducibility.

MAJOR COMMENTS

Point 1 - I am concerned about the underlying assumptions of RNA velocity that are transferred to this model but may not apply, or even be violated, when not using unspliced and spliced RNA information. Unspliced RNA molecules are directly converted into spliced RNA molecules, but transcription factors act in a more complex – and even indirect manner – on their target genes. Therefore, modeling these behaviors with a velocity is significantly more challenging and requires a more elaborate formulation than the one presented by the authors.

Reply: Firstly, while we agree that the complexity of transcription mechanisms is beyond a linear model, recent studies have validated the feasibility of linear models for inferring transcriptional regulatory networks on single-cell RNA data [PMID: 21562557 (*Nature*, 2011), 36762475, 35300460], indicating that linear model is a simple but efficient tool for transcriptional regulation modelling. Secondly, from the aspect of computation, a linear model is easy to optimize, especially when it is nested in a generalized EM approach. And we agree that the model for transcriptional regulation is optional and more complicated ones is worthy trying in future. Thirdly, despite of the difference between TFvelo and spliced-based velocity method, TFvelo can always provide comparable or even better temporal performance on predicting gene and single cell dynamics, as shown in **Figs. 2-6** of our paper, indicating that TFvelo can effectively capture the developmental dynamics in various tissues. Fourthly, TFvelo can correctly find the dynamic of individual gene in the phase portrait in single cell dataset with severe noise and sparsity of unspliced/spliced abundance (**Fig. 4d** in main text and **Fig. S6** in SI), while spliced-based velocity methods (scVelo, UniTVelo, Dynamo and cellDancer) cannot, further validating the velocity concept inferred by TF and target.

As a result, we propose TFvelo as a novel gene dynamic modeling method to for RNA velocity single cell trajectory, pseudotime analysis, etc.

Point 2 - The methods section is too light on detail and does not fully explain the underlying assumptions made by the TFvelo model. The authors appear NOT to use a series of differential equations for inferring a time derivative of the velocity (as is the case for all prior velocity methods). Such an equation set will be challenging to define for complex TF regulation, but without them, the velocity vector becomes less physically interpretable, as transcription, splicing and degradation rate equivalents are not learned. The authors claim the weights of the TFs in TF-Target pairs are directly interpretable as the role of the TF in the velocity, which I agree with, but they provide insufficient examples. One example TF (JUND; Supplemental Figure 2) and distributions of the weights for all TF-Target pairs (Figures S2b, 3j) come across as cherry-picked. A more extensive analysis ought to be performed here: is there any enrichment for particular gene classes in the positive and negative weights? Do any TFs have different weights for different target genes? A single TF usually acts on multiple targets in the same pathway or trajectory, and I would anticipate they have similar weights for each target gene in that path.

Reply: Thanks for these very valuable suggestions.

(1) For the methods section, we now have updated it carefully and provided more details.

(2) As for the differential equations, inspired by UniTVelo, we directly design the expression level of a gene as a function of time, without using a series of ordinary differential equations (ODEs).

Since the solution to previous ODE-based methods is always an analytical function of time, $s(t)$, which increases first and decreases later, it is reasonable to directly design $s(t)$ as a Gaussian function (adopted in UniTVelo) or sine function, still with physical interpretation. Beyond Gaussian, the sine function can further model dynamics starting with down-regulation and turning to up-regulation later (Fig. 4f in main text).

(3) As for the weights analysis, following the reviewer's suggestion, we now have conducted additional GO term and KEGG pathways enrichment analysis for both target genes and TFs with high weights in each dataset, rather than analysis on individual TF-target pair. Especially on the pancreas dataset (Fig. 3 in main text), we provide the following analysis:

a. GO term/ KEGG pathway enrichment analysis for the best fitted genes.

b. To select the TFs that always have high weight on multiple target genes and further analyze the function of these TFs.

c. As suggested, to analyze the TFs' weights on modeling those gene in the same pathway.

Please see the following for details.

Using the pancreas dataset as an example, for the best-fitted genes, we observed a strong KEGG pathways enrichment associated with insulin secretion and the glucagon signaling pathway (Fig. R21a). Additionally, we conducted an analysis of the weights assigned to each TF. It is found that REST and HMGN3 consistently have high absolute weights for different target genes (Fig. R21b). When examining the UMAP distribution, it is clear that REST expression decreases during differentiation, while HMGN3 expression increases (Fig. R21c). Previous study has established REST as a key negative regulator of endocrine differentiation during pancreas organogenesis [PMID: 34385258, 32375045]. Earlier study has also report HMGN3 to be a regulator for insulin secretion in pancreatic cells [PMID: 19651901]. As suggested, we further analyze the weights of these two TFs on modeling genes within the insulin secretion pathway. Although the weights on different target genes are not exact the same, but it is clear that REST consistently has negative weights, while HMGN3 consistently has positive weights (Fig. R21d), which is consistent with the previous findings.

Figure R21. Analysis of results on Pancreas dataset. (a) The KEGG pathway enrichment for the genes best fitted by TFvelo on pancreas dataset. (b) The TFs that always have high weight

values on multiple target genes. The counts mean the number of target genes where the TF have an absolute weight larger than 0.5.(c) The distribution of REST and HMGN3 on UMAP. (d) The weights distribution of TFs REST and HMGN3 on the genes in the insulin secretion pathway.

Point 3 - Likewise, I am doubtful whether the velocity estimate obtained using TF-Target combinations should even be identical to those obtained using unspliced-spliced information. The delay between a TF and its Target implies something about a more upstream rate of change than traditional RNA velocity. I am also not convinced it makes sense to average individual TF-Target velocities across all genes for a single cell, when TF-Target pairs influence different axes of variation that are simultaneously present in a sample. It is essential the authors more carefully reflect on these questions in the manuscript and more robustly evaluate them with TFvelo to ensure that the choices made by their model are physically-consistent with the underlying biological processes for which velocity is being estimated. For example, a recent method called MultiVelo (Li et al 2022) estimates a velocity from snATAC and snRNA-seq data. Comparing the velocity estimates obtained by MultiVelo to those from TFvelo, rather than only to the velocities obtained by scvelo, would help provide a significantly more suitable benchmark.

Reply: (1) We agree that the velocity concept in TFvelo, which refers to the changing trend of total RNA expression level, is different to the velocity by un/spliced based methods, which refers to the changing trend of spliced RNA expression level. However, the two could be highly correlated since TF regulation is an up-stream process of the splicing, with a signal cascade from TF to unspliced RNA (transcription) to spliced RNA (post-transcription). In addition, such correlation has been validated by two aspects. Firstly, by **Eq1** and revised **Fig. S1**, it is proved that the value of RNA velocity modeled by scVelo could be approximated by a linear combination of abundance of TFs, indicating that TF information is sufficient to infer our new velocity. Secondly, TFvelo can always provide comparable or even better temporal performance on predicting gene and single cell dynamics (**Fig. 2** and **Fig. 4** in main text).

Furthermore, TFvelo also benefits a lot from the new velocity concept. Due to sparsity and noise of single cell data, it is hard for existing RNA velocity approaches to extract and model the gene dynamics well, especially in phase portrait. In contrast, TFvelo can address this issue and achieve better performance on different single cell datasets.

(2) We clarify that the velocity of each gene is not simply averaged. The velocity of a cell is a vector in R^{n_gene} , where n_gene is the number of genes. After obtaining the velocity of each gene, we can further get the transition matrix based on the velocity and the expression state of each cell. Then TFvelo calculates the pseudotime based on the directed velocity graph. This is the strategy that has been validated and widely used in previous methods like scVelo.

(3) As for the comparison with Multivelo, many thanks the reviewer for such constructive suggestion. Multivelo uses ATAC-seq to provide transcriptional regulation information, while TFvelo uses TFs, thus validating the feasibility of TFvelo theoretically. From the practice aspect, we show that TFvelo can achieve similar results as Multivelo, even without the ATAC-seq data, further validating the superior of TFvelo. Please see our response to your point 4 for the details (**Fig. 5** in main text).

Point 4 - On a related note to Point 3 above, no comparisons were made by the authors to other modalities used to study gene regulatory behavior, such as single cell ATAC or CUT&Tag

approaches. These methods would be a better proxy for inferring regulatory behavior between TFs and targets, and it is even possible to measure them jointly with scRNA-seq (permitting direct comparison to a traditional RNA velocity estimate).

Reply: We agree that inferring RNA velocity based on multi-omics information, such as Multivelo, is an intriguing approach. Following the reviewer's suggestion, in addition to scVelo, UniTVelo, Dynamo, and cellDancer as benchmark methods on single cell RNA-seq data, we also compare Multivelo on the joint dataset it used, which is 10x multi-omics embryonic mouse brain dataset where both scRNA-seq and ATAC-seq are available. The comparison result is shown in **Fig. 5** and **Fig. R22**.

Specifically, Multivelo is an RNA velocity model that combines both single cell RNA-seq and ATAC-seq to capture RNA dynamics. For TFvelo, we use only the scRNA-seq data, and for Multivelo as described in the paper, both RNA-seq and ATAC-seq are used. In general, the pseudotime inferred by TFvelo is close to that of Multivelo results (**Fig. R22a,b**). Both methods can correctly identify the differentiation direction in most cells. Notably, TFvelo achieves this performance without ATAC-seq data. **Fig. R22c** shows the phase portrait fitting by both methods, in which TFvelo provide 3D phase portrait plotting, using the spliced, unspliced, and learned representation of TFs, while Multivelo shows an additional c-u phase portrait, which is the 2D projection into the space of ATAC-seq and unspliced mRNA.

The correct dynamics can be clearly shown in the TFvelo's 3D phase portrait. As for AHI1 gene, Multivelo mistakenly constructs a cyclic dynamic, initially going up and then going down. In contrast, TFvelo correctly captures the dynamic, with expression increasing consistently from V-SVZ cells (in green). For NTRK2 gene, based on the ground truth expression distribution on UMAP, expression decreases initially, reaching its lowest level in IPC cells (in brown). TFvelo captures such dynamic correctly, while Multivelo misses the decreasing process at the beginning. For GRIN2B gene, TFvelo and Multivelo detect the same dynamics, with expression increasing from the beginning to the terminal stage. These results indicates that without additional ATAC-seq data TFvelo can generate better dynamics than Multivelo. This might be because that the high sparsity and binary nature of ATAC-seq data [PMID: 32620137], which could lead to the mixing of cells from different types, is a challenge for model fitting.

We have not seen any study exploring RNA velocity with CUT&Tag data yet. We really appreciate this suggestion and plan to investigate how to model transcriptional regulation-based velocity using CUT&Tag in the future.

Figure R22. Results of TFvelo on 10x multi-omics embryonic mouse brain dataset. (a) Pseudo-time and trajectory inferred by Multivelo projected into a UMAP-based embedding. (b) Pseudo-time and trajectory inferred by TFvelo projected into a UMAP-based embedding. (c) The comparison between Multivelo and TFvelo on three example genes, AHI1, NTRK2 and GRIN2B, respectively. For Multivelo plot, *c* means the chromatin accessibility measured by ATAC-seq. For TFvelo the 3D phase portrait is obtained by combining the learned representation of TFs (*WX*), spliced and unspliced RNA level.

Point 5 - It is unknown whether RNA expression levels of TFs correspond to protein levels. This is of course true for all genes measured by scRNA-seq, but it is especially critical when focusing on a model involving TFs for velocity specifically because the relationship between TFs and their targets is not unambiguous in the way it is between unspliced and spliced RNA. In the context of this manuscript, TFvelo essentially uses TF gene expression information as a proxy for TF protein binding to the DNA sequences and regulatory elements of target genes, all in order to predict a velocity that seems comparable to that estimated with U/S counts. TF activation/repression of target genes will depend on many cellular factors, including the rate of nuclear import of the TF proteins from the cytoplasm after translation, the general chromatin accessibility, and the influence of regulatory DNA elements. These challenges are one reason why measuring an accurate RNA velocity with single nuclei data is not straightforward: single nuclei data does not contain all spliced RNA content and therefore models RNA nuclear export (occurs on a too-fast time scale) instead of degradation (occurs on a slower timescale measurable in a static snapshot).

Transcription factors may also act on different time scales not suitable to the time scale inferable from a static single cell snapshot. These caveats were not raised in the manuscript and without addressing Points 1-4 above, I am unsure whether we can fully trust the use of TF gene expression for velocity estimation of relevant target genes.

Reply: We agree that the regulation between a TF and a target contains quite complex mechanisms, including the nuclear import of the TF proteins, the general chromatin accessibility, and the influence of regulatory DNA elements. However, as we explained in the response to your point 3, theoretically, given that a series of GRN methods have been proposed to detect gene regulation relationships successfully based on RNA-seq, it is reasonable and widely validated to regard TF RNA expression level as a proxy for TF protein binding to the DNA sequences and regulatory elements of target genes [PMID: 21562557 (Nature, 2011), 36762475, 35300460], thus the transcript expression could be used to infer the TF regulation details in turn. In addition, since TF regulation is very upstream of the signal cascade, the velocity by TF could be highly correlated with that by spliced RNA, which is also validated by our results (e.g., gene H19 shown in **Fig. S7**). And from the practice aspect, TFvelo can further address issues of noise and sparsity and achieve comparable or better performance on different single cell datasets (**Fig 3e,f** and **Fig. S6**). What is more, the recently published method, Multivelo combines the ATAC-seq and un/spliced RNA for dynamic modeling, further validating the feasibility of velocity inference based on transcriptional regulation.

As for the time scales, it has been reported that splicing can be accomplished within 30 seconds [PMID: 24637398], which is much shorter than the time scale of the whole differentiation process described in scRNA-seq datasets. As a result, the phase delay between the unspliced and spliced mRNA could be too short to be captured from the phase portrait. And it might also be one reason of why the theoretical curve can not be observed on the unspliced-spliced phase portrait (**Fig. R23** and **Fig. S12** in SI). **Fig. R23 a,b** provide a simulation to illustrate that even with the same level of noise, a shorter phase delay between the variables could make the phase portrait fitting more challenging. **Fig. R23c** shows that the spliced and unspliced change almost synchronously in the scRNA-seq dataset, while the TFvelo could detect a clearer phase delay for dynamic modeling.

We agree that TF may act on different time scales, with a larger range from minutes to hours [PMID: 27546191]. On the one hand, regulation of certain time scale only can describe biological process of its corresponding time scale. On the other hand, rather than merely one snapshot, single cell RNA-seq data actually measures a group of snapshots across the biological progress people are interested in. As a result, TFvelo is expected to model various biology processes well, which is also validated by the results across our paper.

Figure R23. The comparison of the phase delay in splicing and regulation. (a) In simulation, the phase delay between unspliced and spliced mRNA abundance, which could be very short compared to the range of Pseudotime. The colorbar reflects Pseudotime. (b) The phase delay between TFs and target, which is much longer than that in splicing. The colorbar reflects Pseudotime. (c) Comparison of the phase portrait fitting and expression level along pseudotime obtained by scVelo and TFVelo on two example genes.

Point 6 - I applaud the authors for focusing in their manuscript on the fitting of phase portraits, as this is an essential component of traditional velocity estimation methods and has been regularly neglected by some recent velocity papers. However, it is not clear to me why it is necessary to fit phase portraits on transcription factors and their target genes using expectation-maximization (EM)

in the first place, when the authors first show that their linear model can be sufficiently fit using a LASSO regression. What additional benefits are provided by using EM in this context? Why not just use a simple LASSO regression? It would be a good start to have more rigorous direct comparisons between TFvelo estimates obtained using the LASSO and EM.

Reply: We would like to clarify it. LASSO is one kind of linear regression methods for supervised learning, which can only be applied when the ground truth labels are known. EM is an unsupervised learning framework to optimize multiple groups of parameters iteratively.

In the Findings section, we want to show that the RNA velocity modeled by previous approaches (like scVelo) can also be approximately estimated based on the abundance of TFs with a linear model. We apply LASSO to this supervised learning task, where the velocities given by scVelo are regarded as ground truth labels. As a result, we draw the following conclusions from the **Findings** section: The dynamic constructed from splicing could also potentially be learned from the abundance of TFs.

In the implementation of TFvelo, given that no ground truth label is provided, we need to learn the velocity in an unsupervised framework. We utilized a Generalized Expectation-Maximization (GEM) algorithm to optimize the three groups of parameters iteratively in the following way:

- (1) Updates the weights of the TFs with a linear regression model with hard constraint.
- (2) Update the latent time of cells with grid search.
- (3) Update the parameters in the dynamic equation.

In conclusion, LASSO is employed in the **Findings** section to show that the output of scVelo can also be fitted as a linear combination of TFs' expression level in a supervised manner. While the EM framework is employed in our TFvelo model to infer velocity in an unsupervised manner.

Point 7 - The few case-by-case examples from the literature provided by the authors of TF-Target pairs with strong influences on the overall cellular velocity estimate do not fully convince me of the interpretability of the model. For example, I would expect to see much better characterized TF-target pairs used as validation that their model performs well, rather than obscure genes such as PPP1R14A, SERTAD1, RGS2, and GSTP1 (Figure 4). The authors weakly support their observations for such genes using literature of findings in zebrafish, yet to my understanding, the scRNA-seq data utilized is only from humans (also Figure 4).

Reply: We agree that it is not a solid explanation with only these selected genes, especially the example gene based on the evidence from a zebrafish study. Inspired by your suggestion, we have performed more characterized TF-target pairs used as validation for our model, including (a) Enrichment analysis of KEGG pathways / GO term analysis for the best fitted genes. (b) The function analysis of those TFs which always have high weights on each dataset.

The analysis on pancreas dataset is shown in our reply to your point 2 (**Fig. R21**), point 13 (**Fig. R25**) and also in **Fig. 3** of main manuscript. On the human embryonic stem cell (ESC) dataset, we find the top 3 TFs always having high weights are MYC, GATA2 and KLF4 (**Fig. R24**). Among them, MYC and KLF4 are on the KEGG pathway “signaling pathways regulating pluripotency of stem cells”. Early study has provided evidence that complex regulation of GATA2 is important for the development, expansion, and maintenance of human ESC-derived cells [PMID: 31178416].

Figure R24. The TFs that always have high weights on multiple target genes. The counts mean the number of target genes for which the TF have an absolute weight larger than 0.5.

Point 8 - Velocity confidence is used (Figures 2e, 3f) to show that TFvelo infers a better velocity estimate than scvelo. This metric was made available as part of the scvelo package, but it is not part of the original scvelo paper (Bergen et al 2020) and therefore not peer-reviewed. A better benchmark to convince me of superior velocity estimates by TFvelo is to evaluate the quality of the phase portrait fits themselves. For the TFvelo phase portraits shown in the main/supplemental figures, side-by-side comparisons to the scvelo phase portraits are presented in a confusing manner (on far opposite sides of the panels) or are completely absent. Side-by-side comparisons between TFvelo and scvelo (and perhaps MultiVelo as suggested by Point 4) as well as a more robust metric for evaluating phase portrait fits, would better support the claim that TFvelo yields improved velocity estimates.

Reply: Following the reviewer’s suggestion, we propose three additional metrics to evaluate the quality of the fitting in phase portrait, which are “fitting error on phase portrait”, “intra-class distance on phase portrait” and “inter-class distance on phase portrait”. The metrics demonstrate that, in contrast to un/spliced-based approaches, the phase portrait by TFvelo can better distinguish cells from different types, and aggregate cells from the same type (**Fig. 2 d-f** and **Fig. S10 a-c**).

In addition, we introduce more metrics to assess the inferred trajectory, including the “Cross Boundary Direction Correctness (CBDir)” and the “In Cluster Coherence (ICCo)” (**Fig. 2 g,h** and **Fig. S10 g,h**), which are adopted in VeloAE and UniTVelo. CBDir measures the correctness of transitions from one cell type to the next using boundary cells given ground truth. ICCo is calculated using cosine similarity among cells within the same cluster, which measures the smoothness of velocity in high dimensional space within cluster.

We also revised the figures about side-by-side comparisons with more methods in **Fig. 2c** and **Fig. 4e,f** as the reviewer suggested.

Point 9 - The EM implementation provided by the authors seems to not be significantly different from that previously published as scvelo by Bergen et al 2020. The model framework is, as far as I can see, a cookie-cutter copy of scvelo, with exception of a handful of lines implementing the author's linear model and adjusting the EM to enable the inference of three sets (instead of two sets) of parameters. Providing a more clear explanation of the method developed (see Point 2 and Point 6 above) would be appreciated to clarify (1) why the EM and phase portrait fits are necessary rather than a simple LASSO regression, and (2) how the formulation differs from scvelo.

Reply: We apologize for the confusion. To better explain the difference between TFvelo and scVelo in terms of implementation and modelling, we reorder the question list.

(1) How does the formulation of TFvelo differ from scvelo?

- a. TFvelo models the phase delay in transcriptional regulation, while previous methods like scvelo models the phase delay in splicing;
- b. In TFvelo, a learnable representation of TFs is introduced to model the regulation, which allows a better gene-dynamics-fitting in phase portrait.

(2) Implementation: why develop TFvelo based on scVelo?

Because of the wide usage of scvelo and its clean, well-organized and efficient codes, we develop TFvelo based on the structure of scvelo. We have added more appropriate, prominent acknowledgments for scvelo in our GitHub page, and also included that in our license. In the implementation, the differences include:

- a. Preprocessing: TFvelo needs to obtain the TFs of each target gene, which is not required by scVelo.
- b. Initialization: TFvelo provide specific initialization strategy for multiple parameters. Please find the details in our Method section.
- c. Optimization: to model the transcript regulation dynamics, scvelo uses an EM approach, where E step assigns latent time, and M step optimizes the curve in phase portrait. While TFvelo introduces an additional group of parameters representing the weights of potential TFs and employs a generalized EM approach to allow an additional M step in each iteration.

(3) Why the EM and phase portrait fits are necessary rather than a simple LASSO?

As we clarified in pint 6, LASSO is employed in the *Findings* section to show that the output of scVelo can also be fitted as a linear combination of TFs' expression level in a supervised manner. While the EM framework is employed in our TFvelo model to infer velocity in an unsupervised manner.

Phase portrait fitting for each individual gene is the basis for velocity-based Pseudotime inference, which combines the inferred velocity on all genes with a transition probability matrix. So to accurately fit the gene dynamics in phase portrait is necessary.

Point 10 - After several troubleshooting attempts, I was unable to run the TFvelo code provided on the corresponding GitHub page, and the provided lab website page does not exist. The necessary database files containing information of TF-Target gene pairs are missing, and even when I find what I believe to be the correct files on the web, the code does not run (perhaps because the necessary database files I obtained from the web were not the same as those used by the authors). The ReadMe provided by the authors is also short and unclear. I think this is rather a problem with the usability of the package, rather than a fundamental bug in the code; with better documentation,

it would likely be possible to run the model. However, I am currently unable to do so and cannot verify reproducibility of the results shown in the paper.

Reply: We apologize for the insufficient documentation on the GitHub page. We have uploaded the TF-Target list from ENCODE and ChEA, refactorized our code, and created better documents and tutorials for usage and reproducibility.

MINOR COMMENTS

Point 11 - I greatly appreciate the efforts by the authors to illustrate the performance of their model on simulated data, as I think there is an urgent need to evaluate RNA velocity models on simulated data where a suitable ground truth velocity is known. However, the performance evaluation of the simulated results is limited and relegated to the supplemental file. I think this is something that should be highlighted in the main text of the manuscript.

Reply: Following the reviewer's suggestion, in addition to phase portrait visualization and AUROC, we now have provided more quantitative evaluation on simulated dataset, including the spearman correlation coefficient (SPCC) between the ground truth velocity and the inferred velocity for target gene, and the SPCC between the ground truth weights and inferred weights for TF. We have also provided more analysis about the simulation in both the main text (**Fig. 1 l-o**) and SI (**Fig. S3, Fig. S4**).

Point 12 - Throughout the manuscript, stream plots are used to show the velocity estimations on various datasets (including, but not limited to, Figures 2B, 2C, 3B, 3E, 4B, and 5B). Although this visualization method was introduced in scvelo, it can overly smoothen the velocity arrows and provide misleading interpretations. I suggest that the authors instead illustrate their velocity estimations using the traditional velocity quiver plots.

Reply: Following your suggestions, we also provide the quiver plot in **Fig. S5** and **Fig. S9** of SI. However, considering that the stream-plot is widely adopted in most of the recently published RNA velocity papers, including scVelo, UniTVelo, Dynamo, veloVI and Multivelo, we put the steam plots in main manuscript for consistence.

Point 13 - Velocity is typically present in a dataset as a delay in expression between two entities, in this case the TFs and the target. It would be useful to see visualization of the expression of both these entities (TF and target) along the learnt latent time, in order to assess whether that expected shift is indeed present after performing EM (and not just in the phase portraits themselves).

Reply: Following the reviewer's suggestion, the visualization along the latent time for the learnt representation of TFs and target are shown in the **Fig. R25**. We can always find a clear phase delay between TFs and target gene. We further analyze the phase delay between target gene and individual TF with high absolute weight. Within these two examples, HMGN3 has a positive weight on modeling SURF, where a gap between the phases of them can be clearly observed. On the contrary, REST has a negative weight on modeling ECE1, where they show obvious negative correlation. In conclusion, the expected shift is indeed present after performing EM.

Figure R25: Phase visualization of gene SURF4 and ECE1 on Pancreas dataset. On each row, from the left to the right: the dynamics on phase portrait, the expression on UMAP, the phase delay between the learned regulation representation (WX) and target gene, and the phase delay between individual TF with the high weight and target gene.

Point 14 - Although in some cases we might not be able to compute velocity information using unspliced/spliced, in two scRNA-seq datasets used by the authors (pancreas and erythroid gastrulation) we indeed have the possibility to estimate the velocity with sevelo. I had hoped for a direct comparison between the velocity vectors estimated using the two different models, not just the velocity confidence. How well do they correlate? Do they correlate differently for different cell types? This is related to Point 7 above.

Reply: Following your suggestions, we analyzed the cosine similarity between the velocity vectors obtained by scVelo and TFvelo, based on the genes modeled by both methods. Results (**Fig. R26**) suggest that the velocities always positively correlate across different cell types, and they indeed correlate differently for different cell types.

Figure R26. The cosine similarity between the velocity vectors obtained by scVelo and TFVelo.

Point 15 - The application of TFVelo to spatial transcriptomics data is especially exciting, as it is a modality in which velocity is difficult to infer since probes are rarely intron-specific. I also appreciate the authors correctly visualizing the velocity on a single cell UMAP space as opposed to arrows directly on the image, which has been done in several previous studies yet is completely incorrect nonsense (since cells don't physically move in space according to the velocities). However, I do not understand why the authors remove M phase cells before performing analysis. They claim it is to avoid a circular trajectory, but even with the M phase cells, one could represent the velocity as a single linear trajectory along the cell cycle.

Reply: Now we use the MERFISH dataset without filtering out cells in M phase. Following your suggestion, we update the post-processing strategy on MERFISH dataset. Results show that these genes can be well fitted and the whole arrow plot also indicates the cell cycle dynamics (**Fig. R27**).

Figure R27: Results on MERFISH dataset. (a) The stream plot on UMAP. (b) The constructed dynamic models on three example genes with high likelihoods, which are NAV2, TLL1 and KLF7, respectively. From the left to the right within each row, the three subfigures show phase portrait fitting, distribution of velocity on UMAP, and distribution of expression on UMAP, respectively.

Point 16 - In Figure 2, the authors claim that upon velocity estimation with TFvelo, the progression of cells out of the cell cycle into ductal cells is corrected; and better following a trajectory expected by the prior biological knowledge. However, it is concerning to me to observe the loss of a circular velocity along the cycling cells themselves, as cells are certainly progressing through the cell cycle phases prior to differentiation. I am unsure whether TF is indeed correcting; an incorrect; velocity in this context, or whether it simply fixes one velocity direction (from progenitors to ductal cells) while breaking another (progression in the cell cycle).

Reply: The motivation behind TFvelo is to achieve better dynamic fitting in the phase portrait for following velocity-based pseudotime inference. Consequently, after modeling all genes, we would select the best-fitted genes based on both the fitting error and the consistency between the gene-specific latent time and the inferred cell-specific pseudotime. In the pancreas dataset, given that differentiation is the main trend described in the data, with the procedure, the pseudotime and stream plot primarily reflect the dynamics of differentiation.

Point 17 - The majority of figures and figure legends in the manuscript are vague and difficult to follow. In fact, most supplemental figures do not have any figure captions. Figure legends are

sometimes missing or illegible, making it very difficult to understand the color/label schemes of the corresponding panels.

Reply: Sorry for these mistakes. We now have corrected these figures.

Reviewer #1 (Remarks to the Author):

The idea of using TFs for velocity is well reasoned and validated. The comparison with other approaches well presented. I agree with publication to Nature Communication.

Reviewer #2 (Remarks to the Author):

Review of "TFvelo: gene regulation inspired RNA velocity estimation" (NCOMMS-23-16358A-Z)

Summary

The authors have improved their manuscript by adding comparisons with more recent velocity models, including metrics to rely less on 2D UMAP interpretation, and clarifying that their method is not a GRN inference framework. They also added some comparisons with Palantir and PAGA, as baseline pseudotime approaches. However, I disagree with their claimed advantages over pseudotime methods:

They don't need access to the initial cells: in all of the examples that they show, the initial cells are clear from prior knowledge and can be provided to a pseudotime method.

TFvelo can analyze the detailed dynamics of genes in phase space: that is true, but it remains unclear to me what exactly can be learned from that.

TFvelo can detect important genes and TFs: Palantir (Setty et al. 2019), and other approaches can also do that. Palantir looks at genes that correlate with diffusion potential, Monocle (Cao et al. 2019) just looks at genes that correlate with pseudotime, and TradeSeq (Van den Berge et al. 2020) can even test for detailed dynamics (up-and-down-regulation, periodic activation, etc.). CellRank 2 (Weiler et al. 2023), based on the PseudotimeKernel (without RNA velocity), can even look at genes that are important for specific trajectories with learned fate probabilities.

Another problem with this manuscript is that it claims to be a "velocity model," which, in my understanding, it is not, as it does not model the time delay between any two molecular modalities, such as spliced and unspliced counts. It just considers mRNA counts of TFs and putative-regulated genes. I find statements like the following confusing:

(L259) "To draw a conclusion, TFvelo can address the current issue caused by noisy un/spliced data for RNA velocity modeling by using the learned feature with transcription regulation."

In my understanding, TFvelo does not solve a problem in RNA velocity modeling of spliced/unspliced counts, it rather follows a completely different route, which is more related to earlier pseudotime approaches, based on just (total) mRNA counts. Solving the problem of modeling noisy counts, in my opinion, would require a new statistical approach, which is better adapted to the noise distribution and potential biases in spliced/unspliced counts and their detection.

I comment on individual points in the following.

Re point 2: I appreciate the author's efforts to include more recent velocity approaches and to compare them more quantitatively. However, for the genes shown in Fig. 2c, scVelo seems to work just as well: H19 is correctly predicted to be downregulated, and MAML3 is correctly predicted to be upregulated. It would be nice to evaluate this more systematically by using a list of genes with known important functions in this biological process and checking how many of them are not fitted correctly with competing approaches.

I'm not sure the comparison in panel f is valid, as these models are very different, and directly comparing them might grant unfair advantages to their method.

Re point 3: TFvelo learns a cell-specific latent time when it fits the EM model. This latent time is much less smoothed than the velocity-pseudotime the authors compute later on and should be visualized and evaluated quantitatively to assess model performance. For example, the authors could check how well their actual latent time (not the velocity pseudotime) correlates with a simple

Palantir pseudotime.

Identifying initial cells based on the stationary distribution of the transpose transition matrix is not straightforward and should be illustrated better. How automatic is this really? How much do the authors need to manually tune this procedure to identify meaningful initial cells? As this is an important claim of the paper (‘velocity-pseudotime’ without specifying initial cells), I expect to see more analysis on the initial cells they predict automatically. Also, the initial distribution is a distribution, how do the authors threshold it to find the set of initial cells?

The authors construct a transition matrix to infer their velocity pseudotime. Why can they not use the same transition matrix for embedding projection? Why do they employ additional gene filtering to learn a different transition matrix? A “velocity graph” is also just a transition matrix, maybe normalized differently.

Re my software point: The authors now acknowledge scVelo in their GitHub README, but they include neither a proper citation to the paper nor a link to the package there.

As to the example of the moments function, the functionality TFVelo requires seems very simple: while scVelo applies k-NN imputation separately to unspliced and spliced counts, TFVelo applies this to total counts stored in AnnData `.X`. Rather than copying the code and modifying it locally, the best practice would be to add that functionality via a PR to scVelo and import the modified function. This has at least two advantages:

The entire community benefits from the added functionality in scVelo.

TFVelo benefits from future improvements in scVelo. In this case, potential bug fixes and improvements to k-NN imputation.

In my opinion, copying code to add very minor functionality, which could be useful outside of TFVelo, is not the right way to advance software for the community.

Re my original major comments

Re point 5: I appreciate the author's efforts to quantitatively evaluate sparsity levels in spliced/unspliced counts. In SFig. 6, the authors show that, as expected, the sparsity level among unspliced counts is higher compared to spliced counts. Yet, this is a dataset where velocity approaches, including scVelo, have been shown several times to work very well. For example, scVelo captures the cell cycle among ductal cells, which is missed by TFVelo, judging from the projection in Fig. 2b. On the other hand, scVelo does not perform well on the gastrulation erythroid dataset of Fig. 4. Thus, I think there are more nuanced reasons why velocity approaches work well on some datasets, and not on others, that do not just depend on overall sparsity, but probably on the capture rate of some important genes for a given biological process, time scales, etc.

Re point 6: The authors still do not include any baselines in their simulated data. I understand that they don't simulate spliced/unspliced counts, so they cannot apply a model like scVelo. Yet, they could include another simple (baseline) model to put their own performance into context.

Re point 7.1: Based on their histogram in Fig. R9, many individual genes will still have latent time assignments, which are inconsistent with the overall cell ordering. Can the authors comment on this point and illustrate what happens to these genes? Do they assume an entirely reversed order in latent time?

Re point 7.2: That is not the only pseudotime scVelo provides. scVelo also outputs the cell-specific latent time, which is obtained by post-hoc pooling of gene-specific latent times. If the authors follow the scVelo implementation of velocity-pseudotime, i.e., DPT on the velocity-derived transition matrix, then they must also include a symmetrization step, as DPT cannot deal with complex eigenvalues that arise in the decomposition of non-symmetric-transition-matrices. In that case, their pseudotime would be based on the “un-directed” velocity graph rather than the “directed” velocity graph.

Re point 10: I could not find a direct answer to my question - the kNN-graph used to compute

velocity confidence should not be based on a 2D UMAP but on a higher-dimensional space (e.g., a 30-dim PCA or scVI latent space).

Re point 15: that is a practical issue, related mostly to cell/nuclei segmentation in FISH-based spatial transcriptomics data, this could be explained better.

Re point 18: as a result, would TFvelo be applicable to the large-scale datasets we are generating nowadays? If not, that would be something to mention as a limitation in the discussion.

Re my original minor comments

Re point 27: I disagree, approaches like CellRank learn trajectory-specific weights per cell and include these in a GAM-fit to show trajectory-specific gene expression trends in heatmaps.

References

Cao, Junyue, Malte Spielmann, Xiaojie Qiu, Xingfan Huang, Daniel M. Ibrahim, Andrew J. Hill, Fan Zhang, et al. 2019. "The Single-Cell Transcriptional Landscape of Mammalian Organogenesis." *Nature* 566 (7745): 496–502.

Setty, Manu, Vaidotas Kisieliovas, Jacob Levine, Adam Gayoso, Linas Mazutis, and Dana Pe'er. 2019. "Characterization of Cell Fate Probabilities in Single-Cell Data with Palantir." *Nature Biotechnology* 37 (4): 451–60.

Van den Berge, Koen, Hector Roux de Bézieux, Kelly Street, Wouter Saelens, Robrecht Cannoodt, Yvan Saeys, Sandrine Dudoit, and Lieven Clement. 2020. "Trajectory-Based Differential Expression Analysis for Single-Cell Sequencing Data." *Nature Communications* 11 (1): 1201.

Reviewer #3 (Remarks to the Author):

The authors properly answered all the points I raised.

Reviewer #4 (Remarks to the Author):

This revised manuscript by Li and colleagues describes TFvelo, a method that harnesses the relationship between expression of transcription factors (TF) and their target genes (Target) to predict gene-specific temporal changes using single cell RNA sequencing (scRNA-seq) data. TFvelo is inspired by traditional RNA velocity, which uses the delay between unspliced (U) and spliced (S) entities to infer the rate of change in gene expression. Here, the authors use pre-defined relationships between TFs and their Target genes to model gene regulatory changes in a similar manner to the scvelo implementation of RNA velocity. The model is solved iteratively with expectation-maximization (EM) by learning the transcription factor weights, the phase portrait shape parameters, and cellular latent time.

With this revised manuscript, the authors have made several noteworthy improvements to both the manuscript and the code base that enhance the overall quality and reproducibility of the results. There are still some concerns that I have, outlined below, but if these are addressed, then I would be supportive of this study's publication in *Nature Communications*.

Some of the most important changes made to the manuscript based on the major comments from my previous review, with any remaining concerns, are stated below.

1. Direct performance comparisons to another RNA velocity-inspired method inferring velocities from gene regulatory information, MultiVelo. I appreciate the authors making a direct comparison between TFvelo and MultiVelo, which estimates gene-wise velocity using jointly-profiled chromatin accessibility (snATAC) and gene expression (snRNA). Since chromatin accessibility is influenced by the activity of transcription factor binding to DNA, there are some important parallels between MultiVelo and TFvelo that are not shared with other velocity approaches. I appreciate the author's effort to make this comparison, and I think it greatly strengthens the manuscript.

The authors claim velocities inferred with TFvelo are more accurate than those from MultiVelo due to the sparsity of ATAC measurements. Although I think it is reasonable to suggest that TFvelo might, in some circumstances, recover more biologically-correct velocity vector fields, this could also be due to the more direct link between a set of TFs and their target, whereas gene activity or peaks obtained using snATAC are more challenging to directly link to a specific gene (i.e., an enhancer and other regulatory region may lie far from the gene it affects). I would qualify the statements in lines 392-396 with a bit more caution. Also, the reference to Figure 5c in lines 396-400 seem a bit of a contradiction to me: if the unspliced reads do not contain enough information in these data, why do they confirm the results obtained with TFvelo? It does not make much sense to be visualizing all three components (spliced, unspliced, and TF expression) on the same "phase portrait" plot.

Furthermore, I would appreciate if the authors could make the code and analysis notebook used to generate the results of this comparison available on their GitHub. As far as I can see, only the analysis using their tool on the relevant dataset is available, and not the results when comparing directly to MultiVelo.

2. Incorporation of new metrics to evaluate performance across different velocity models. Several new metrics are applied to compare TFvelo to other velocity approaches. However, the presentation of these metrics is a bit confusing, as so many are introduced without much explanation as to why so many different metrics are necessary. Furthermore, the sections of the manuscript describing these metrics reads a bit more like a figure legend as opposed to a results section. I suggest the authors choose a few metrics to focus on in the main text, while moving the remaining ones to the new supplementary note. Personally, I found the intra/inter class distance on the phase portrait, the in-cluster coherence, and the cosine similarity (such as in Figure R26) metrics to be the most informative.

I am also a bit concerned there is a bias towards better performance for velocity methods that are specifically relying on a similarly formulated temporal model to TFvelo and UniTVelo. For instance, it is a bit unclear to me exactly why TFvelo and UniTVelo specifically perform much better on the in-cluster coherence and velocity confidence metrics (Figure 2h and 2i). The results of the metrics need to be explained in the text, not just stated.

Moreover, I am concerned about the use of the velocity confidence metric. None of the methods benchmarked against are probabilistic, and therefore confidence cannot be directly evaluated. Probabilistic models for RNA velocity do exist, but are not explored (VeloVAE, VeloVI, and Pyro-Velocity, among others). Given the use of multiple other metrics, I suggest the authors remove the velocity confidence metric and any related claims from their manuscript. To examine confidence of a velocity estimate, a probabilistic model in which velocity is implemented using variational inference (VI) should be employed instead.

3. Application of TFvelo to a spatial transcriptomics dataset. I appreciate the authors re-running their analysis including the cells in M phase. However, I am still not fully convinced by the results on the MERFISH data, as I think they are difficult to interpret from the figures provided by the authors. From the stream plots alone (Figure 6e-f and Figure R27), it is unclear whether there is a strong enough velocity signal being detected to obtain a biologically-meaningful direction. In no figure panel is it clearly annotated which phases the various cells belong to, and therefore the counterclockwise vector field broadly indicated by the authors (Figure 6e) is difficult to biologically assess (do cells flow in the correct cell cycle direction). Likewise, NAV2, TLL1, and KLF7 are indeed more expressed in different areas of the UMAP space, but it is unclear which cell cycle phases they are meant to represent. Although there is value in showing TFvelo's application to datasets lacking unspliced information, in which traditional RNA velocity cannot be performed, I suggest some reworking of these analyses to increase interpretability, if they are to be included in the study.

4. The code is now better documented and can be run. I would, however, request the specific code or notebooks utilized to generate the analyses and visualizations showed in the figures of the manuscript. At the moment, some, but not all, of these analyses are provided. In particular, the MERFISH analyses (Figure 6) is absent.

Minor Comments

- I generally suggest reordering the bar plots in Figure 2 to sort the methods from best to worst performing (instead of keeping the methods in the same order in each panel), while leaving the colors the same across panels (as is currently the case).
- In Figures 2d and 2e, the x-axis labels should be "TFvelo" and "scvelo", rather than "TFvelo" and "Un/spliced", in order for better clarity on which RNA velocity implementation TFvelo is being benchmarked against.
- In Figure 2g, I do not see any significant differences in cross boundary correctness, so I am not sure what I am supposed to take away from this evaluation.
- In Figure 4d, the velocity is visualized differently for cellDancer (with discrete vector fields) compared to the other methods illustrated (with stream plots). Is there a particular reason for this? If not, I suggest using the same plotting style for all methods, to facilitate a fairer comparison of the velocity vector fields
- The size of the plots for MultiVelo and TFvelo in Figure 5c are different from each other. Please make them more uniform in size. Additionally, the axes labels in the 3D plots (5c) and the cluster labels (5a) are difficult to read.
- How does TFvelo define a particular TF weight as "significant" versus "not significant"?
- I think the illustrated delay between TF and Target shown in Figure R23 a and b is pretty impressive, and should be included in the manuscript for the purpose of illustrating the additional signal present when aggregating TF-Targets together compared to using unspliced and spliced alone. Likewise, I appreciated the cosine similarity analysis in Figure R26 and think it is more valuable to be included in the main text rather than some of the other benchmarking metrics.

We thank all reviewers again for their constructive comments, which have helped us strengthen this work a lot. We appreciate reviewers 1 and 3 for their support on publication, and the patience and valuable questions by reviewers 2 and 4. Now, we have addressed their remaining concerns point by point.

Reviewer #1 (Remarks to the Author):

The idea of using TFs for velocity is well reasoned and validated. The comparison with other approaches well presented. I agree with publication to Nature Communication.

Reviewer #2 (Remarks to the Author):

Review of TFvelo: gene regulation inspired RNA velocity estimation; (NCOMMS-23-16358A-Z)

Summary

The authors have improved their manuscript by adding comparisons with more recent velocity models, including metrics to rely less on 2D UMAP interpretation, and clarifying that their method is not a GRN inference framework. They also added some comparisons with Palantir and PAGA, as baseline pseudotime approaches. However, I disagree with their claimed advantages over pseudotime methods:

They don't need access to the initial cells: in all of the examples that they show, the initial cells are clear from prior knowledge and can be provided to a pseudotime method.

TFVelo can analyze the detailed dynamics of genes in phase space: that is true, but it remains unclear to me what exactly can be learned from that.

TFVelo can detect important genes and TFs: Palantir (Setty et al. 2019), and other approaches can also do that. Palantir looks at genes that correlate with diffusion potential, Monocle (Cao et al. 2019) just looks at genes that correlate with pseudotime, and TradeSeq (Van den Berge et al. 2020) can even test for detailed dynamics (up-and-down-regulation, periodic activation, etc.). CellRank 2 (Weiler et al. 2023), based on the PseudotimeKernel (without RNA velocity), can even look at genes that are important for specific trajectories with learned fate probabilities.

Reply:

We now have updated our statements in ***Introduction***. Now, we place emphasis on introducing TFvelo as a novel and alternative approach to perform analyses conducted in previous RNA velocity studies, rather than claiming these advantages over TI methods. We state that while TI methods enable pseudotime analysis at both the cell and gene levels, they typically require the annotation of initial cells [DOI:10.1038/s41592-023-01994-w]. As for your questions:

(1) *“In all of the examples, the initial cells are clear from prior knowledge and can be provided to a pseudotime method.”*

R: We acknowledge that the initial cells are known in these datasets, which are used to perform pseudotime methods, and to validate RNA velocity approaches. However, scRNA-seq datasets do not inherently come with cell type labeling. Annotating cell types in single-cell data requires a lot of efforts with unavoidable errors and biases. With the explosive growth of single-cell sequencing data and the limited number of experts available, a substantial amount of future data will not be appropriately labeled. Therefore, it would be an advantage for TFvelo and all other RNA velocity methods that they can infer cell fate without relying on such annotations. The recently published veloVI also highlights the limitation regarding the requirement of the initial state annotation in many trajectory inference approaches [DOI:10.1038/s41592-023-01994-w].

(2) *“It remains unclear what exactly can be learned from the phase space fitting.”*

R: (a) Compared with these pseudotime inference methods, TFvelo and other RNA velocity methods are mechanistic approaches [DOI:10.1038/s41592-023-01994-w] where phase portrait fitting plays the core role in modeling the dynamics of each individual gene. That is to say, phase portrait fitting is the basis for cell fate prediction in RNA velocity methods, allowing alternative mechanistic approaches for temporal order analysis based on individual gene-specific modeling.

(b) In contrast to other RNA velocity methods, TFvelo provides a novel approach to modeling each gene from the regulatory perspective. For example, the phase portraits in TFvelo allow for the direct, individual analysis of phase transition between the target gene and its TFs.

(c) Furthermore, the fitting of phase portraits can be used to evaluate gene modeling, thereby enhancing the robustness and interpretability of downstream cell fate predictions. A recent *Nature Reviews Genetics* paper [DOI: 10.1038/s41576-023-00586-w] also suggests that it is important to analyze the robustness of RNA velocity inference by verifying the phase portraits matching the expected shape to ensure that the underlying model assumptions could hold in the scRNA-seq data.

(3) *“Other pseudotime inference approaches can also detect important genes.”*

We apologize for the inaccurate statement, and agree that some pseudotime inference approaches can also detect important genes. We have updated the claim in the **Introduction** and mentioned that these methods can also provide gene-level analysis. TFvelo provide a novel and alternative way of analyzing the dynamics between each target gene and its TFs.

Another problem with this manuscript is that it claims to be a “velocity model,” which, in my understanding, it is not, as it does not model the time delay between any two molecular modalities, such as spliced and unspliced counts. It

just considers mRNA counts of TFs and putative-regulated genes. I find statements like the following confusing:

(L259) ‘To draw a conclusion, TFvelo can address the current issue caused by noisy un/spliced data for RNA velocity modeling by using the learned feature with transcription regulation.’

In my understanding, TFVelo does not solve a problem in RNA velocity modeling of spliced/unspliced counts, it rather follows a completely different route, which is more related to earlier pseudotime approaches, based on just (total) mRNA counts. Solving the problem of modeling noisy counts, in my opinion, would require a new statistical approach, which is better adapted to the noise distribution and potential biases in spliced/unspliced counts and their detection.

Reply: We agree that TFvelo is different with conventional RNA velocity approaches. Inspired by your question, we have summarized the connection between TFvelo and these conventional RNA velocity approaches, and the reason why TFvelo could be regarded as an RNA velocity approach for the following reasons.

(1) Based on its physical meaning, the “RNA velocity” is supposed to reflect the changing rate of RNA abundance. In conventional approaches, RNA velocity is actually defined as the changing rate of spliced RNA abundance. Since the phase delay between TFs and target could also provide temporary information, we developed TFvelo, where the “velocity” is the changing rate of total RNA abundance. Theoretically, both can quantify the RNA changing rate, and be further used to infer the cell fate.

(2) In practice, TFvelo can accomplish analysis performed by previous RNA velocity studies, such as gene-specific phase portrait fitting, velocity modeling, inferring pseudotime without annotation of initial states, and visualizing cell fate with the velocity-based transition stream plot.

(3) Furthermore, TFvelo provide a novel way to address several fundamental challenges in RNA velocity methods, including the sparsity and noisy in single-gene unspliced/spliced data. As a result, TFvelo always performs better with gene dynamics fitting with less loss (**Figs. R1, R2** in our response to your next question), higher coherence of velocity (**Fig. 2** in main text) and the more accurate inference on velocity stream (**Fig. 4** in main text).

We appreciate the suggestion of development of new statistical models to address noisy data. As explained above, TFvelo offers an alternative approach to address the challenges of high noise and sparsity in un/spliced data. We also agree that directly modeling the noise could be beneficial, and are open to exploring this direction in future.

We have also updated our *Introduction* to clearly explain the connection between TFvelo and conventional RNA velocity methods.

I comment on individual points in the following.

Re point 2: I appreciate the author's efforts to include more recent velocity approaches and to compare them more quantitatively. However, for the genes shown in Fig. 2c, scVelo seems to work just as well: H19 is correctly predicted to be downregulated, and MAML3 is correctly predicted to be upregulated. It would be nice to evaluate this more systematically by using a list of genes with known important functions in this biological process and checking how many of them are not fitted correctly with competing approaches.

Reply: Thank you for your suggestion. We selected genes according to the paper that proposed the pancreas dataset [DOI: 10.1242/dev.173849]. From the section of “single cell RNA-seq of the embryonic EP-enriched pancreatic epithelial cells” in that paper, 33 genes were introduced as marker genes in the pancreas differentiation process, which are DLK1, CPA1, MYC, NOTCH2, PTF1A, CEL, RBPJL, SOX9, ANXA2, BICC1, NGN3, HES6, FEV, CCK, NEUROD1, RBP4, PYY, CHGB, MDK, BTF3, VTN, JAM3, CBX3, HMGN1, YBX1, REEP5, SPP1, BTBD17, GADD45A, VWA5B2, TOX3, TMEM97, FAM183B, CBFA2T3, RCOR2, SMARCD2, INSM1, CBFA2T2, SPP1, TMSB10, MDK, MARCKSL1, CDK4, and SOX4. Among them, 22 genes are selected by the pre-processed step of both TFvelo and scVelo, which are FAM183B, RBP4, SPP1, SOX9, GADD45A, MDK, TMEM97, DLK1, VWA5B2, ANXA2, BICC1, PYY, FEV, CCK, CHGB, CPA, BTBD17, JAM3, NEUROD1, NOTCH2, RBPJL and TOX3.

We next compare the fitting by TFvelo and scVelo on these 22 genes in **Figs. R1**, which is quantitatively evaluated by the Spearman correlation between these gene-specific latent time and Palantir pseudotime (**Fig. R2**). As shown in the phase portrait fitting of all the 22 genes in **Fig. R1**, scVelo fails to fit 10 genes whose Spearman correlation is set as 0. In contrast, TFvelo gets high spearman correlation (larger than 0.6) for 11 genes, and a much higher median value (0.58) compared to scVelo (0.00), indicating that TFvelo can provide a much more accurate fitting for these important genes.

Figure R1.1. The comparison between TFvelo and scvelo on the genes selected from the pancreas study, including the dynamics fitting in phase portrait (1st and 3rd columns)

for TFvelo and scvelo respectively), and the expression along gene-specific latent time (2nd and 4th columns for TFvelo and scvelo respectively). The spearman correlation between each gene-specific time with Palantir pseudotime is shown in the plot.

Figure R1.2. The comparison between TFvelo and scvelo on the genes selected from the pancreas study, including the dynamics fitting in phase portrait (1st and 3rd columns for TFvelo and scvelo respectively), and the expression along gene-specific latent time (2nd and 4th columns for TFvelo and scvelo respectively). The spearman correlation between each gene-specific time with Palantir pseudotime is shown in the plot.

Figure R1.3. The comparison between TFvelo and scvelo on the genes selected from the pancreas study, including the dynamics fitting in phase portrait (1st and 3rd columns for TFvelo and scvelo respectively), and the expression along gene-specific latent time (2nd and 4th columns for TFvelo and scvelo respectively). The spearman correlation between each gene-specific time with Palantir pseudotime is shown in the plot.

Figure R2. The comparison between TFvelo and scvelo, in terms of the distribution of Spearman correlation between the gene-specific latent time and the Palantir pseudotime, on functional genes selected from the pancreas study. The median value of Spearman correlations obtained by each method is shown in the figure.

I'm not sure the comparison in panel f is valid, as these models are very different, and directly comparing them might grant unfair advantages to their method.

Reply: The phase portrait fitting is the basic step for RNA velocity, and the fitting error could be a good metric to evaluate the modeling performance of each individual gene. To address the reviewer's concern, we now have adopted two strategies to make the comparison fair. (a) The fitting error is calculated based on those genes which are modeled by both TFvelo and scVelo. (b) Normalization is applied to each dimension (spliced and unspliced for conventional approaches, target and TFs for TFvelo). In addition, we now move this panel into the **Supplementary Information**.

Re point 3: TFVelo learns a cell-specific latent time when it fits the EM model. This latent time is much less smoothed than the velocity-pseudotime the authors compute later on and should be visualized and evaluated quantitatively to assess model performance. For example, the authors could check how well their actual latent time (not the velocity pseudotime) correlates with a simple Palantir pseudotime.

Identifying initial cells based on the stationary distribution of the transpose transition matrix is not straightforward and should be illustrated better. How automatic is this really? How much do the authors need to manually tune this procedure to identify meaningful initial cells? As this is an important claim of the paper ('velocity-pseudotime' without specifying initial cells), I expect to see more analysis on the initial cells they predict automatically. Also, the initial distribution is a distribution, how do the authors threshold it to find the set of initial cells?

The authors construct a transition matrix to infer their velocity pseudotime. Why can they not use the same transition matrix for embedding projection? Why do they employ additional gene filtering to learn a different transition matrix? A “velocity graph” is also just a transition matrix, maybe normalized differently.

Reply:

(1) “TFVelo learns a cell-specific latent time when it fits the EM model. This latent time is much less smoothed than the velocity-pseudotime and should be visualized and evaluated quantitatively to assess model performance.”

R: In the generalized EM framework, the gene-specific latent time is assigned to cells when modeling the dynamic of each individual gene. By integrating the learned dynamics across all genes, TFVelo infers the cell-specific pseudotime based on the directed transition matrix, following the strategy scVelo used. Given your comments in *Re point 7.2* that “post-hoc pooling of gene-specific latent times”, we speculate that the latent time in “TFVelo learns a cell-specific latent time when it fits the EM model”, could be a pooled time over all genes for cell c :

$$t_c^{latent} = \frac{1}{n_{gene}} \sum_{g=1}^{n_{gene}} t_{g,c}$$

where $t_{g,c}$ is the gene-specific time for cell c on gene g , and the number of genes (n_{gene}) is 2000 on this pre-processed pancreas dataset. The analysis for this pooled latent time t^{latent} after being scaled to the range of 0 to 1 is shown in **Fig. R3**. The spearman correlation between it and the Palantir pseudotime is 0.788.

Figure R3. The visualization of the Palantir pseudotime and the pooled latent time, which is obtained by averaging the gene-specific latent time modeled on all genes.

(2) “Identifying initial cells based on the stationary distribution of the transpose transition matrix is not straightforward and should be illustrated better.”

The strategy is the same with that in scVelo, the root and end cells are automatically computed and all hyperparameters are identical across all scRNA-seq datasets without being tuned individually. The root and end cells are derived from the stationary states of the Markov chains, represented by the velocity-inferred directed transition matrix

and its transpose, respectively. This process involves identifying the 10 eigenvectors with the largest eigenvalues using the `scipy.sparse.linalg.eigs()` function. After that, among the 10 selected eigenvectors, those corresponding to eigenvalues lower than 0.999 are further filtered out. Subsequently, using the left eigenvectors and the connectivity graph between cells, which is constructed during preprocessing, we can get the distribution of root/end cells.

Using the same strategy with `scvelo`, after smoothing the probability of root/end cell with the connectivity graph, the one cell with the highest probability will be utilized as the root/end cell in pseudotime inference (**Fig. R4**). Pseudotime for each cell is computed as the mean value of: (a) The number of steps it takes to reach the cell after starting to walk from the root cell, normalized to a range between zero and one, and (b) One minus the normalized number of steps it takes to reach the end point after starting to walk from the cell.

Figure R4. The root/end cell used for pseudotime inference on different datasets.

(3) “*Why can they not use the same transition matrix for embedding projection? Why do they employ additional gene filtering to learn a different transition matrix?*”

After obtaining the pseudotime, the gene filtering step is utilized to select well-fitted genes whose dynamics align with the inferred pseudotime. This is because TFvelo aims to improve cell fate prediction through a more accurate dynamic modeling of each gene. To achieve this goal, using TFs-target information and learning a representation of TFs could theoretically result in better phase portrait fitting. In practice, the additional filtering step can further remove the noise in the dynamics of these ultimately selected genes.

To illustrate the robustness of TFvelo with this strategy, we show the performance with different number of selected genes. Using the top-10, top-30, top-50, and top-100 genes where the gene-specific latent time best aligns with the pseudotime, we can always get similar stream plots to that of **Fig. R5**.

Figure R5. The stream-plot obtained from different number of velocity genes.

Re my software point: The authors now acknowledge scVelo in their GitHub README, but they include neither a proper citation to the paper nor a link to the package there.

As to the example of the moments function, the functionality TFVelo requires seems very simple: while scVelo applies k-NN imputation separately to unspliced and spliced counts, TFVelo applies this to total counts stored in AnnData `X`. Rather than copying the code and modifying it locally, the best practice would be to add that functionality via a PR to scVelo and import the modified function. This has at least two advantages: The entire community benefits from the added functionality in scVelo. TFVelo benefits from future improvements in scVelo. In this case, potential bug fixes and improvements to k-NN imputation.

In my opinion, copying code to add very minor functionality, which could be useful outside of TFVelo, is not the right way to advance software for the community.

Reply: Thanks for your suggestion, now we have included the citation of their paper and a link to their package. We acknowledge the value in integrating certain functions into the current version of scVelo, as suggested by the reviewer. Following your suggestions, we have contacted the developer of scVelo to explore the possibility of extending `scvelo.pp.moments()` to incorporate k-NN smoothing for the total counts. This function would be optional and applicable when `AnnData.layers['total']` exists. (Currently, the information about total counts is neither presented nor utilized in the

scvelo pipeline, where `AnnData.X` is exactly identical to `AnnData.layers['spliced']`.)

Beyond scVelo framework, to implement TFVelo we have added and modified a series of functions, which include the TF selection in preprocessing, the learning of TF weight, the usage of the generalized EM framework, and the strategy for optimization involving multiple initializations to effectively address the expanded optimization space, etc. And other functions are also modified to make them harmonize with each other. Given these substantial differences, it will take some time to incorporate most of these functions by submitting pull requests to the scVelo repository.

In addition, we would like to clarify that, in this study, our contribution is to introduce the new method TFVelo to model the dynamics of each RNA from a novel aspect of regulation, which can be used to further infer the cell fate. And our current released code could (a) satisfy the reproducing requirement, (b) be used for anyone who want to run TFVelo on their data, (c) potentially be further developed by the anyone who needs some functions in TFVelo.

Therefore, currently we maintain TFVelo as an independent tool, which allows us to directly offer the implementation and reproducibility to the community at this stage. Once the scvelo team agrees to incorporate features into their functions, we can submit pull requests (PRs) and import them from scvelo in the future.

Re my original major comments

Re point 5: I appreciate the author's efforts to quantitatively evaluate sparsity levels in spliced/unspliced counts. In SFig. 6, the authors show that, as expected, the sparsity level among unspliced counts is higher compared to spliced counts. Yet, this is a dataset where velocity approaches, including scVelo, have been shown several times to work very well. For example, scVelo captures the cell cycle among ductal cells, which is missed by TFVelo, judging from the projection in Fig. 2b. On the other hand, scVelo does not perform well on the gastrulation erythroid dataset of Fig. 4. Thus, I think there are more nuanced reasons why velocity approaches work well on some datasets, and not on others, that do not just depend on overall sparsity, but probably on the capture rate of some important genes for a given biological process, time scales, etc.

Reply: We agree with your analysis and appreciate for your idea about the capture rate and time scales. This study is motivated by the observation that in practice, only a very small number of genes appear to obey the interpretable kinetics adopted in spliced RNA velocity models, which is a major challenge in interpreting RNA velocity results. The failure could be much more complicated than our previous analysis described.

In addition to the sparsity and noise in unspliced data, we now have investigated that the short time scale of splicing process could also make it hard to extract dynamics from the delay between unspliced and spliced RNA. Splicing can be accomplished within 30 seconds [DOI: 10.4161/nucl.28056], a duration much shorter than the timescale of the

biological process described in most scRNA-seq datasets. Consequently, the phase delay between the unspliced and spliced mRNA could be too brief to be captured by the phase portrait analysis (**Fig. 3f,g** and **Fig. R6**). Simulations in **Fig. R6a, b** indicate that, with the same level of noise, a shorter phase delay between entities (**Fig. R6a**) makes the phase portrait fitting more challenging. **Fig. R6c** shows that in real scRNA-seq data, the spliced and unspliced changes almost synchronously, in contrast, a much clearer phase delay can be detected by TFvelo for the same gene.

We now have updated our *Introduction* and *Results* to include these further analyses.

Figure R6. The comparison of the phase delay in splicing and regulation. (a) In simulation, the phase delay between two variables with short delay, i.e., the unspliced and spliced mRNA abundance. The colorbar reflects pseudotime. (b) In simulation, the phase delay between two variables with long delay, i.e., the TF and target RNA abundance. The colorbar reflects pseudotime. (c) Comparison of the phase portrait fitting and expression level along pseudotime obtained by scVelo and TFvelo on two

example genes.

Re point 6: The authors still do not include any baselines in their simulated data. I understand that they don't simulate spliced/unspliced counts, so they cannot apply a model like scVelo. Yet, they could include another simple (baseline) model to put their own performance into context.

Reply: Following your suggestion, we employ a vanilla EM approach as baseline here, by removing the optimization step on TF weight, as well as the strategy of multiple points initialization. As shown in **Fig. R7**, the performance of the baseline is significantly worse than TFvelo. The velocity inferred by vanilla EM exhibits weak correlation with the ground truth velocity, whereas TFvelo's predictions have higher correlation (**Fig. R7a,b**). In **Fig. R7c**, three examples are presented where the EM baseline fails to extract dynamics, in contrast, TFvelo accurately extracts dynamics from the simulation data.

Figure R7. The performance of the simple EM baseline on synthetic dataset. (a) The joint distribution between the ground truth velocity and inferred velocity by EM baseline, with spearman correlation in the bottom right. (b) The joint distribution between the ground truth velocity and inferred velocity by TFvelo approach. (c) Three examples for the dynamics fittings comparison between the baseline and TFvelo.

Re point 7.1: Based on their histogram in Fig. R9, many individual genes will still have latent time assignments, which are inconsistent with the overall cell ordering. Can the authors comment on this point and illustrate what happens to these genes? Do they assume an entirely reversed order in latent time?

Reply: To address your concerns, we show the typical failure cases in **Fig. R8** and **Supplementary Fig. S12**. Firstly, for genes with low expression levels, cells tend to be positioned in the TFs-axis in phase portraits, making their fitting challenging (**Fig. R8a**). Secondly, some genes exhibit a reversed order in latent time modeling (**Fig. R8b**).

Thirdly, TFvelo may encounter difficulties in effectively modeling some genes due to high noise, resulting in poor fits (**Fig. R8c**). We have also mentioned it in the *Discussion* section. However, as demonstrated, for example, in our response to your Re point 2 (**Figs. R1, R2**), TFvelo could still capture the dynamics accurately for more genes than the baseline approach.

Figure R8. The failure case of TFvelo on modeling individual genes. (a) TFvelo cannot fit the data when the gene is expressed in only one cell type. (b) TFvelo infers inverse dynamics. (c) TFvelo is affected by high noise.

Re point 7.2: That is not the only pseudotime scVelo provides. scVelo also outputs the cell-specific latent time, which is obtained by post-hoc pooling of gene-specific latent times. If the authors follow the scVelo implementation of velocity-pseudotime, i.e., DPT on the velocity-derived transition matrix, then they must also include a symmetrization step, as DPT cannot deal with complex eigenvalues that arise in the decomposition of non-symmetric-transition-matrices. In that case, their pseudotime would be based on the *un-directed* velocity graph rather than the *directed* velocity graph.

Reply: We apologize that we did not provide enough details about the velocity pseudotime, which might result in your misunderstanding. In fact, the scVelo implementation of velocity-pseudotime is based on a directed transition matrix without any symmetrization steps, and TFvelo follows the same approach. In both scVelo and TFvelo, velocity pseudotime is a random-walk based distance measure on the directed velocity graph. Contrarily to diffusion Pseudotime (DPT), it implicitly infers the root and end cells instead of using the similarity-based diffusion kernel. The detailed steps are provided:

- (1) Construct velocity graph: Computes velocity graph based on the cosine similarities between velocities and potential cell state transitions. This is different with those DPT approaches, which use the similarity-based diffusion kernel.
- (2) Find the root and terminal states. The endpoints and root cells are obtained as stationary states of the velocity-inferred transition matrix and its transpose. The

scipy.sparse.linalg.eigs(T) function is employed in the implementation, where T is the transition matrix, and it is not necessary for it to be a symmetrized matrix. This is because the scipy.sparse.linalg.eigs() function can handle non-symmetric matrices.

(3) Calculate pseudotime. After identifying root and terminal cells from the velocity-inferred transition matrix, pseudotime for each cell is computed as the mean value of: (a) The number of steps it takes to reach the cell after starting to walk from the root cell, normalized to a range between zero and one, and (b) One minus the normalized number of steps it takes to reach the end point after starting to walk from the cell.

Re point 10: I could not find a direct answer to my question -the kNN-graph used to compute velocity confidence should not be based on a 2D UMAP but on a higher-dimensional space (e.g., a 30-dim PCA or scVI latent space).

Reply: To better address your concern, we now provide a more detailed explanation. In fact, the velocity confidence is computed based on higher-dimensional space in the following steps.

- (1) Get the velocity vector for each cell i , $v_i \in R^{n_genes}$, where n_genes means the number of velocity genes.
- (2) Norm the velocity vector of each cell using its L2 norm, $\bar{v}_i = v_i/||v_i||$
- (3) Get the neighbors of each cell, where the neighborhood graph is constructed based on all genes in the pre-processing steps. As a result, the KNN-graph used to compute velocity confidence is the same with the KNN-graph used to perform imputation in pre-processing.
- (4) Calculate the coherence of velocity within neighboring cells, which is measured by the dot product: *velocity confidence at cell i* = $\frac{1}{|s(i)|} \sum_{j \in s(i)} \bar{v}_i \cdot \bar{v}_j$, where $s(i)$ means the set of neighbors for cell i .

As a result, the velocity confidence is obtained in high dimensional space. The 2D UMAP is only used for visualization to illustrate the distribution of velocity confidence.

In addition, to avoid misleading, we have renamed "velocity confidence" to "velocity consistency" in our manuscript since it focuses on the consistency of velocities within neighboring cells, rather than the statistical definition of "confidence".

Re point 15: that is a practical issue, related mostly to cell/nuclei segmentation in FISH-based spatial transcriptomics data, this could be explained better.

Reply: We agree. Although it is possible to apply splicing-based velocity approaches to FISH-based datasets by considering the nuclei/cytoplasm RNA as unspliced/spliced mRNA [DOI: 10.1073/pnas.1912459116], cell/nuclei segmentation in FISH-based spatial transcriptomics images is still challenging [DOI: 10.1038/s41587-021-01044-w]. Most of the segmentation methods require manual tuning and corrections [DOI: 10.1002/cyto.a.23686]. Therefore, utilizing RNA velocity methods based on unspliced/spliced transcripts with nuclear/cytoplasmic data might introduce extra noise. This could worsen the inherent challenges of high noise and insufficient signal caused by RNA abundance split when employing spliced-based RNA velocity methods.

Re point 18: as a result, would TFvelo be applicable to the large-scale datasets we are generating nowadays? If not, that would be something to mention as a limitation in the discussion.

Reply: Due to the additional step of learning a representation of TFs in each iteration, the time efficiency of TFvelo is lower compared to that of scVelo. This could be a weakness for TFvelo when being applied to large-scale datasets. However, performing TFvelo with more CPUs in parallel could decrease the overall running time. We have mentioned this limitation in *Discussion*.

Re my original minor comments

Re point 27: I disagree, approaches like CellRank learn trajectory-specific weights per cell and include these in a GAM-fit to show trajectory-specific gene expression trends in heatmaps.

Reply: Sorry for our inaccurate statement. We agree that CellRank could show trajectory-specific gene expression trends in heatmaps. We clarify that these baseline RNA velocity approaches, which mainly employ the heatmap implemented by scVelo, suffers from the same issue that multiple cell types share the mixed pseudotime so they can not be well separated in the axis of pseudotime on this heatmap.

References

Cao, Junyue, Malte Spielmann, Xiaojie Qiu, Xingfan Huang, Daniel M. Ibrahim, Andrew J. Hill, Fan Zhang, et al. 2019. "The Single-Cell Transcriptional Landscape of Mammalian Organogenesis." *Nature* 566 (7745): 496-502.

Setty, Manu, Vaidotas Kisieliovas, Jacob Levine, Adam Gayoso, Linas Mazutis, and Dana Pe'er. 2019. "Characterization of Cell Fate Probabilities in Single-Cell Data with Palantir." *Nature Biotechnology* 37 (4): 451-60.

Van den Berge, Koen, Hector Roux de Béziers, Kelly Street, Wouter Saelens, Robrecht Cannoodt, Yvan Saeys, Sandrine Dudoit, and Lieven Clement. 2020. "Trajectory-Based Differential Expression Analysis for Single-Cell Sequencing Data." *Nature Communications* 11 (1): 1201.

Reviewer #3 (Remarks to the Author):

The authors properly answered all the points I raised.

Reviewer #4 (Remarks to the Author):

This revised manuscript by Li and colleagues describes TFvelo, a method that harnesses

the relationship between expression of transcription factors (TF) and their target genes (Target) to predict gene-specific temporal changes using single cell RNA sequencing (scRNA-seq) data. TFvelo is inspired by traditional RNA velocity, which uses the delay between unspliced (U) and spliced (S) entities to infer the rate of change in gene expression. Here, the authors use pre-defined relationships between TFs and their Target genes to model gene regulatory changes in a similar manner to the scvelo implementation of RNA velocity. The model is solved iteratively with expectation-maximization (EM) by learning the transcription factor weights, the phase portrait shape parameters, and cellular latent time.

With this revised manuscript, the authors have made several noteworthy improvements to both the manuscript and the code base that enhance the overall quality and reproducibility of the results. There are still some concerns that I have, outlined below, but if these are addressed, then I would be supportive of this study's publication in Nature Communications.

Some of the most important changes made to the manuscript based on the major comments from my previous review, with any remaining concerns, are stated below.

1. Direct performance comparisons to another RNA velocity-inspired method inferring velocities from gene regulatory information, MultiVelo. I appreciate the authors making a direct comparison between TFvelo and MultiVelo, which estimates gene-wise velocity using jointly-profiled chromatin accessibility (snATAC) and gene expression (snRNA). Since chromatin accessibility is influenced by the activity of transcription factor binding to DNA, there are some important parallels between MultiVelo and TFvelo that are not shared with other velocity approaches. I appreciate the author's effort to make this comparison, and I think it greatly strengthens the manuscript.

The authors claim velocities inferred with TFvelo are more accurate than those from MultiVelo due to the sparsity of ATAC measurements. Although I think is reasonable to suggest that TFvelo might, in some circumstances, recover more biologically-correct velocity vector fields, this could also be due to the more direct link between a set of TFs and their target, whereas gene activity or peaks obtained using snATAC are more challenging to directly link to a specific gene (i.e., an enhancer and other regulatory region may lie far from the gene it affects). I would qualify the statements in lines 392-396 with a bit more caution. Also, the reference to Figure 5c in lines 396-400 seem a bit of a contradiction to me: if the unspliced reads do not contain enough information in these data, why do they confirm the results obtained with TFvelo? It does not make much sense to be visualizing all three components (spliced, unspliced, and TF expression) on the same 'phase portrait' plot.

Furthermore, I would appreciate if the authors could make the code and analysis

notebook used to generate the results of this comparison available on their GitHub. As far as I can see, only the analysis using their tool on the relevant dataset is available, and not the results when comparing directly to MultiVelo.

Reply: Thank you for acknowledging our improvements. We explain your question raised in this section point by point below.

(1) *“I would qualify the statements in lines 392-396 with a bit more caution.”*

R: We agree with your ideas about the challenge in fitting ATAC data. We have updated our explanation and included the discussion that TFvelo can directly link multiple TFs into modeling a target gene, while peaks obtained by snATAC are more challenging to directly link to a specific gene, in the section of **“TFvelo can achieve competitive performance compared to Multivelo using only scRNA-seq data”**.

(2) *“If the unspliced reads do not contain enough information in these data, why do they confirm the results obtained with TFvelo? It does not make much sense to be visualizing all three components (spliced, unspliced, and TF expression) on the same phase portrait plot.”*

R: We apologize for the unclear statement. Following your suggestion, we have removed the 3D (TFs-u-s) plots from the manuscript.

The conclusion that TFvelo can achieve better fitting is drawn according to the 2D TFs-target phase portrait, cell type annotation and the UMAP distribution: The 2D TFs-target plots utilized in TFvelo can better separates cells from different types (the fourth column in **Fig. R9c**). The gene expression on UMAP (the last column in **Fig. R9c**) can help verify the learned dynamics of each gene. By comparison, the 3D and 2D phase portraits in Multivelo suffer from the high noise in data (the first three columns in **Fig. R9c**).

Figure R9. Results of TFvelo on 10x multi-omics embryonic mouse brain dataset. (a) Pseudo-time and trajectory inferred by Multivelo projected into a UMAP-based embedding. (b) Pseudo-time and trajectory inferred by TFvelo projected into a UMAP-based embedding. (c) The comparison between Multivelo and TFvelo on three example genes, which are AHI1, NTRK2 and GRIN2B, respectively. For Multivelo plot, c means the chromatin accessibility in ATAC-seq.

(3) “The code and analysis notebook.”

R: Now we have added more comprehensive codes, including those for reproducing of baseline approaches and quantitatively evaluating. Especially, we have provided two notebooks that generate all the results of Multivelo at: https://github.com/xiaoyeye/TFvelo/blob/main/baselines/MultiVelo_run.ipynb and

https://github.com/xiaoyeye/TFvelo/blob/main/baselines/Multiveloc_analysis.ipynb.

2. Incorporation of new metrics to evaluate performance across different velocity models. Several new metrics are applied to compare TFvelo to other velocity approaches. However, the presentation of these metrics is a bit confusing, as so many are introduced without much explanation as to why so many different metrics are necessary. Furthermore, the sections of the manuscript describing these metrics reads a bit more like a figure legend as opposed to a results section. I suggest the authors choose a few metrics to focus on in the main text, while moving the remaining ones to the new supplementary note. Personally, I found the intra/inter class distance on the phase portrait, the in-cluster coherence, and the cosine similarity (such as in Figure R26) metrics to be the most informative.

I am also a bit concerned there is a bias towards better performance for velocity methods that are specifically relying on a similarly formulated temporal model to TFvelo and UniTVelo. For instance, it is a bit unclear to me exactly why TFvelo and UniTVelo specifically perform much better on the in-cluster coherence and velocity confidence metrics (Figure 2h and 2i). The results of the metrics need to be explained in the text, not just stated.

Moreover, I am concerned about the use of the velocity confidence metric. None of the methods benchmarked against are probabilistic, and therefore confidence cannot be directly evaluated. Probabilistic models for RNA velocity do exist, but are not explored (VeloVAE, VeloVI, and Pyro-Velocity, among others). Given the use of multiple other metrics, I suggest the authors remove the velocity confidence metric and any related claims from their manuscript. To examine confidence of a velocity estimate, a probabilistic model in which velocity is implemented using variational inference (VI) should be employed instead.

Reply:

(1) Following your suggestions, we have updated the figures in main text, moved some metrics into supplementary information, and also added more explanations on why these metrics are selected. The explanation for each metric is provided below, while we only show the Intra/inter class distance on the phase portrait, the in-cluster coherence, and the cosine similarity in the main text.

a. Intra/inter class distance on the phase portrait: illustrate that the unspliced/spliced phase portrait is always noisy, that cells from multiple type are always mixed.

b. Phase portrait fitting error: support our observation that conventional approaches can not achieve high accurate phase portrait fitting, and our motivation of TFvelo on providing more accurate dynamics fitting for each gene in phase portrait.

c. In-cluster coherence and velocity consistency (which was named as velocity confidence): measure the local consistency of the inferred velocities.

d. Cross Boundary Direction Correctness assesses the correctness of transitions from

one cell type to the next, utilizing boundary cells with ground truth annotations.

e. Cosine similarity: reflect how the velocity inferred by TFvelo and scVelo correlated in each cell type.

(2) Velocity confidence measures the consistency of RNA velocities across neighboring cells, and in-cluster coherence measures the consistency of RNA velocities among cells of the same type. These two metrics have been employed to evaluate models' performance in multiple previous studies [DOI: 10.1073/pnas.2105859118, 10.1038/s41467-022-34188-7, 10.1038/s41592-023-01994-w]. The direct reason for the higher scores obtaining by TFvelo and UniTVelo is that their models get the smoother velocity streams across neighboring cells (**Fig. S5b,d** in **SI**). TFvelo and UniTVelo have similarity in model construction, both directly modeling gene abundance as smooth high-order differentiable functions with respect to time. In comparison, scVelo and Dynamo commence from ODEs and assume a switching point to separate the entire dynamics into two individual stages, breaking the model smoothness. This could be the theoretical reason why TFvelo and UniTVelo can get smoother streams on multiple datasets (**Fig. 2** and **Fig. 4** in main text). We have incorporated this analysis into the explanation paragraph of **Fig.2** in the manuscript.

(3) We totally agree with your perspective on the term "confidence". To avoid misleading, we rename it as "velocity consistency". Both velocity consistency and in-cluster coherence are used to reflect smoothness in velocity over the manifold. Following your suggestion, we move the velocity consistency metric into the **Supplementary Information**.

3. Application of TFvelo to a spatial transcriptomics dataset. I appreciate the authors re-running their analysis including the cells in M phase. However, I am still not fully convinced by the results on the MERFISH data, as I think they are difficult to interpret from the figures provided by the authors. From the stream plots alone (Figure 6e-f and Figure R27), it is unclear whether there is a strong enough velocity signal being detected to obtain a biologically-meaningful direction. In no figure panel is it clearly annotated which phases the various cells belong to, and therefore the counterclockwise vector field broadly indicated by the authors (Figure 6e) is difficult to biologically assess (do cells flow in the correct cell cycle direction). Likewise, NAV2, TTLL1, and KLF7 are indeed more expressed in different areas of the UMAP space, but it is unclear which cell cycle phases they are meant to represent. Although there is value in showing TFvelo's application to datasets lacking unspliced information, in which traditional RNA velocity cannot be performed, I suggest some reworking of these analyses in increase interpretability, if they are to be included in the study.

Reply: Given the facts mentioned by the reviewer, we agree that the results on the MERFISH data are not interpreted as clearly as those on the other datasets utilized in

this study. So we remove the MERFISH results from the manuscript, and use the human brain cancer dataset to show TFvelo's application to datasets lacking splicing information.

4. The code is now better documented and can be run. I would, however, request the specific code or notebooks utilized to generate the analyses and visualizations showed in the figures of the manuscript. At the moment, some, but not all, of these analyses are provided. In particular, the MERFISH analyses (Figure 6) is absent.

Reply: We have added more code on our GitHub page (<https://github.com/xiaoyeye/TFvelo>), including those for reproducing all baseline approaches and calculating all quantitative evaluation metrics presented in this study. The MERFISH analyses have been removed from the manuscripts.

Minor Comments

- I generally suggest reordering the bar plots in Figure 2 to sort the methods from best to worst performing (instead of keeping the methods in the same order in each panel), while leaving the colors the same across panels (as is currently the case).

Reply: Thanks for this suggestion. We have updated the figures accordingly.

- In Figures 2d and 2e, the x-axis labels should be 'TFvelo' and 'scvelo', rather than 'TFvelo' and 'Un/spliced', in order for better clarity on which RNA velocity implementation TFvelo is being benchmarked against.

Reply: Sorry for misleading. In fact, Figures 2d and 2e shows the data distribution on phase portrait, which is not related to the modeling methods. As a result, these approaches which tries to fit the unspliced-spliced phase portrait (including scVelo, UniTVelo, cellDancer etc.) share the same plot. These two plots show that the TFs-target phase portrait learned by TFvelo can provide more information than the original unspliced-spliced phase portrait, which can support our motivation about the challenge in phase portrait fitting for velocity methods. We have explained this more in the manuscript.

- In Figure 2g, I do not see any significant differences in cross boundary correctness, so I am not sure what I am supposed to take away from this evaluation.

Reply: Following your suggestion, we have move that into the supplementary information.

- In Figure 4d, the velocity is visualized differently for cellDancer (with discrete vector fields) compared to the other methods illustrated (with stream plots). Is there a particular reason for this? If not, I suggest using the same plotting style for all methods, to facilitate a fairer comparison of the velocity vector fields

Reply: we run each method according to its tutorial. The stream-plots in scVelo, UniTVelo, Dynamo, Multivelo and TFvelo are basically drawn by employing the

strategy provided by scvelo, which is implemented as `scvelo.tl.streamplot()`. But in the tutorial of cellDancer, after modeling the velocities on each cell and each gene, the arrow plot is drawn with a specific strategy different from `scvelo.tl.streamplot()`. We have also applied `scvelo.tl.streamplot()` to the velocities modeled by cellDancer, and get the result shown in **Fig. R10**. But we respect the independence and integrity of each method. As a result, the velocity is visualized differently for cellDancer. We have added clarification about this in our manuscripts.

Figure R10. The stream-plot obtained with `scvelo.tl.streamplot()` on the velocities modeled by cellDancer.

- The size of the plots for MultiVelo and TFvelo in Figure 5c are different from each other. Please make them more uniform in size. Additionally, the axes labels in the 3D plots (5c) and the cluster labels (5a) are difficult to read.

Reply: Thanks. We have updated the figures for better visualization.

- How does TFvelo define a particular TF weight as *significant*; versus *not significant*;

Reply: If the normalized weight of a TF is larger than 0.5, it is defined as a significant one. The details have been provided in our *Methods* section.

- I think the illustrated delay between TF and Target shown in Figure R23 a and b is pretty impressive, and should be included in the manuscript for the purpose of illustrating the additional signal present when aggregating TF-Targets together compared to using unspliced and spliced alone. Likewise, I appreciated the cosine similarity analysis in Figure R26 and think it is more valuable to be included in the main text rather than some of the other benchmarking metrics.

Reply: Thanks for acknowledging these additional analyses. Following your suggestions, we have re-organized our figures in the manuscript and presented these two figures in **Fig. 3f** and **Fig. 2g** respectively. Your valuable suggestions help us make the paper better organized.

Reviewer #2 (Remarks to the Author):

The authors addressed all my concerns; I agree with the publication in Nature Communications.

Reviewer #4 (Remarks to the Author):

The authors have satisfactorily answered all the points I have raised, and I agree with the work's publication to Nature Communication.

Reviewer #4 (Remarks on code availability):

I was able to install the TFvelo package and run the demo pipelines available on GitHub, using the author's README file instructions